



# Chromophoric dissolved organic matter dynamics revealed through the optimization of an optical-biogeochemical model in the NW Mediterranean Sea

Eva Álvarez[1], Gianpiero Cossarini[1], Anna Teruzzi[1], Jorn Bruggeman[2,3], Karsten Bolding[2,4], Stefano
Ciavatta[5], Vincenzo Vellucci[6], Fabrizio D'Ortenzio[7], David Antoine[8,7], Paolo Lazzari[1]

[1]National Institute of Oceanography and Applied Geophysics - OGS, Trieste, 34010, Italy
[2]Bolding & Bruggeman ApS, Asperup, 5466, Denmark
[3]Plymouth Marine Laboratory, Plymouth, PL1 3DH, UK
[4]Aarhus University, Department of Ecoscience, Silkeborg, 8600, Denmark
[5]Mercator Ocean International, 31400 Toulouse, France
[6]Sorbonne Université, CNRS, Institut de la Mer de Villefranche, IMEV, Villefranche-sur-Mer, F-06230, France
[7]Sorbonne Université, CNRS, Laboratoire d'Océanographie de Villefranche, LOV, Villefranche-sur-Mer, F-06230, France
[8]Remote Sensing and Satellite Research Group, School of Earth & Planetary Science, Curtin University, Perth, WA 6102,
Australia

*Correspondence to*: Eva Álvarez (ealvarez@ogs.it)





**Abstract.** Chromophoric dissolved organic matter (CDOM) significantly contributes to the non-water absorption budget in the Mediterranean Sea. The absorption coefficient of CDOM, $a_{CDOM}(\lambda)$, is measurable in situ and remotely from different platforms and can be used as an indicator of the concentration of other relevant biogeochemical variables, e.g., dissolved

organic carbon. However, our ability to model the biogeochemical processes that determine CDOM concentrations is still limited. Here we propose a novel parametrisation of the CDOM cycle that accounts for the interplay between the light- and nutrient-dependent dynamics of local CDOM production and degradation, as well as its vertical transport. The parameterization is included in a one-dimensional (1D) configuration of the Biogeochemical Flux Model (BFM), which is here coupled to the General Ocean Turbulence Model (GOTM) through the Framework for Aquatic Biogeochemical Models (FABM). Here BFM

is augmented with a bio-optical component that revolves spectrally the underwater light transmission. We did run this new GOTM-FABM-BFM configuration to simulate the seasonal $a_{CDOM}(\lambda)$ cycle at the deep-water site of the BOUSSOLE project in the North-Western Mediterranean Sea. Our results show that accounting for both nutrient and light dependence of CDOM production improves the simulation of the seasonal and vertical dynamics of $a_{CDOM}(\lambda)$, including a subsurface maximum that forms in spring and progressively intensifies in summer. Furthermore, the model consistently reproduces the higher-than-

average concentrations of CDOM per unit chlorophyll concentration observed at BOUSSOLE. The configuration, outputs and sensitivity analyses from this 1D model application will be instrumental for future applications of BFM to the entire Mediterranean Sea in a 3D configuration.

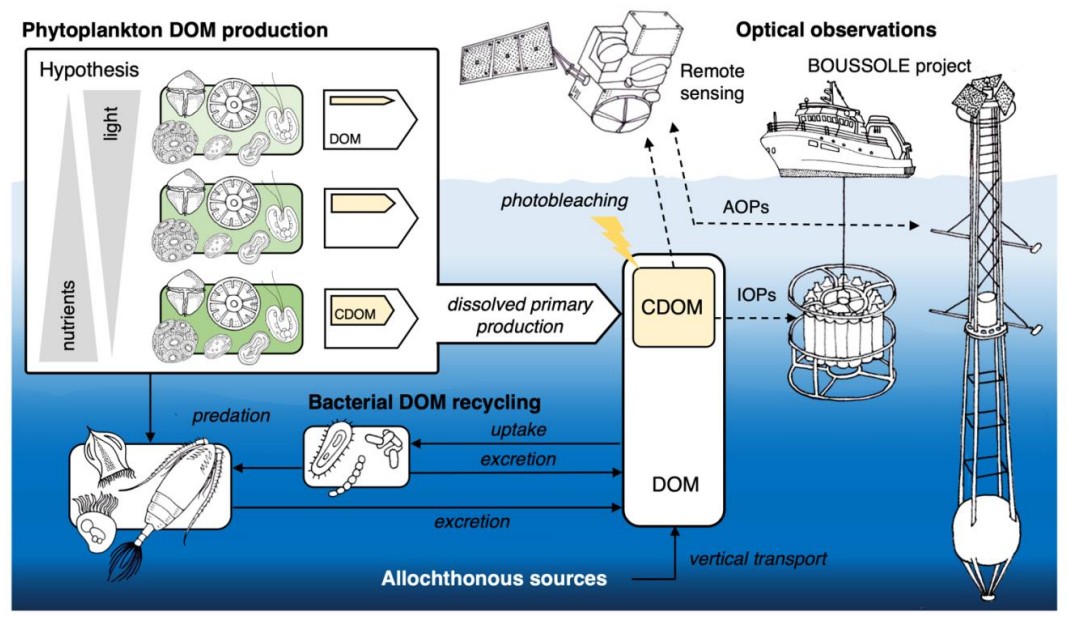



## 1 Introduction

The inventory and composition of dissolved organic matter (DOM) is of utmost importance to marine ecosystems, as it is the energy base and carbon source for most heterotrophic life in the ocean (Azam et al., 1983). Furthermore, the spatio-temporal

dynamics of the organic material dissolved in the oceans reflects the functioning of the carbon cycle (Legendre et al., 2015). Although most of the dissolved exudates that form the DOM are non-absorbing (Mühlenbruch et al., 2018), part of the pool absorbs light mainly in the ultraviolet (UV) and blue spectral range of the electromagnetic radiation. This portion of the DOM is referred to as chromophoric dissolved organic matter (CDOM, Jerlov, 1951; Sieburth and Jensen, 1968). CDOM absorbs light depending on both the CDOM concentration and its mass-specific, spectral absorption coefficient $a_{CDOM}(\lambda)$. This

coefficient is a measurable proxy for fundamental processes regulating CDOM dynamics. The high absorption of CDOM in the blue part of the electromagnetic spectrum affects the water leaving radiance, a radiometric quantity that can be retrieved by satellites, and interferes with estimates of chlorophyll-a concentration (Chl-a) from ocean-colour observations (Siegel et al., 2013). This is of particular importance in the Mediterranean Sea. The optical properties of the basin are peculiar because its surface waters (0-20 m) appear greener than the global ocean when the phytoplankton concentration is low (Chl-a < 0.2 mg

m$^{-3}$) (Morel and Gentili, 2009b; Claustre et al., 2002; Volpe et al., 2007). There are several possible factors determining this optical behaviour, including: the particular pigment ratios in the Mediterranean phytoplankton community (Organelli et al., 2011), the abundance of small coccolithophores (Gitelson et al., 1996), and the influence of Saharan dust (Claustre et al., 2002). However, the high contribution of CDOM was found to be a major factor, since the CDOM concentration in the basin is about twice than in the Atlantic at the same latitudes (Morel and Gentili, 2009b).

$a_{CDOM}(\lambda)$ can be measured both in situ at selected locations and on a global scale from remote-sensing platforms. The latter provide radiometric observations, which are used to compute a combined value for the absorption coefficient of CDOM and of non-algal particles, $a_{DG}(\lambda)$ (Werdell et al., 2018). On the other hand, spectrally resolved biogeochemical (BGC) models provide $a_{CDOM}(\lambda)$ estimates by simulating CDOM concentrations and prescribing the optical properties of the pool. Such BGC-optical models have the potential to advance the understanding of the dynamics that shape $a_{CDOM}(\lambda)$. However, they rely on

the accurate representation of both the dynamics of CDOM cycling and its optical properties. CDOM cycling is essentially controlled by in situ biological production (Romera-Castillo et al., 2010), terrestrial and atmospheric inputs, microbial consumption (Nelson and Gauglitz, 2016; Legendre et al., 2015; Stedmon and Markager, 2005), as well as deep ocean circulation and/or vertical mixing (Coble, 2007), and it is photoreactive and efficiently destroyed in the upper layers of the water column by solar radiation (Mopper and Kieber, 2000). Most marine ecosystem models that explicitly resolve CDOM

follow Bissett et al., (1999) (e.g., Xiu and Chai, 2014; Dutkiewicz et al., 2015), in which CDOM is assumed to have a source that is a fixed fraction of the dissolved primary production (*dpp*) of phytoplankton, is remineralized slowly and bleached under high light conditions. These formulations seem to work well in open waters (Dutkiewicz et al., 2015) but not in the North-Western (NW) Mediterranean Sea, where CDOM is not a constant proportion of *dpp*, even when covarying with phytoplankton Chl-a (Lazzari et al., 2021b). The aim of this work is to decipher the processes that influence CDOM dynamics in the NW

Mediterranean Sea, by advancing and using a BGC model in synergy with in situ observations.

A major challenge in modelling CDOM concentrations in the Mediterranean Sea concerns the origin of the CDOM pool in superficial waters, i.e. how large is the proportion of CDOM transported from allochthonous sources compared to CDOM produced locally by primary and secondary production (Santinelli et al., 2002; Copin-Montégut and Avril, 1993). In the study region, the higher-than-average CDOM concentrations seem to be sustained by allochthonous sources, as fluxes of DOM

depositions from the atmosphere are 2–5 times larger in the Mediterranean Sea than in the oceans, which explains the abundance of humic-like substances (Santinelli, 2015; Galletti et al., 2019). Therefore, accurate modelling of the balance between CDOM transport (atmospheric input, advection and/or vertical replenishment from deeper stocks) and destruction at the surface (photolysis and bacterial consumption) is crucial to accurately represent the role of CDOM of allochthonous origin in shaping the dynamics of $a_{CDOM}(\lambda)$ in the NW Mediterranean Sea.



As for locally produced CDOM, the challenge is that the mechanisms for CDOM production and destruction involve processes that are decoupled from phytoplankton *dpp*, such as photolysis, zooplankton messy feeding, and bacterial production and consumption. In the open waters of the NW Mediterranean Sea, far from terrestrial inputs, photobleaching and biological production and consumption are suggested to be important factors determining the seasonality of $a_{CDOM}(\lambda)$ (Pérez et al., 2016; Xing et al., 2014; Organelli et al., 2014). $a_{CDOM}(\lambda)$ and Chl-a concentration roughly covary at the surface, but the causative

mechanisms are different. While $a_{CDOM}(\lambda)$ at the surface is probably controlled by vertical transport of CDOM from depth and by photochemical destruction, Chl-a concentration at the surface is controlled by photoacclimation of phytoplankton and nutrient limitation. There is also a close coupling between subsurface $a_{CDOM}(\lambda)$ and phytoplankton Chl-a via microbial activities or interactions in the planktonic food web (Pérez et al., 2016; Xing et al., 2014; Organelli et al., 2014), although the relative contribution to CDOM production by phytoplankton and by bacteria is still controversial (Romera-Castillo et al.,

2010). Although it is generally assumed that phytoplankton in open oceans are not a direct source of CDOM, but rather a source of labile organic matter that is microbially transformed and subsequently produces CDOM (Nelson et al., 1998; DeGrandpre et al., 1996), observations in the NW Mediterranean Sea suggest direct CDOM production by phytoplankton (Oubelkheir et al., 2007, 2005; Xing et al., 2014). Therefore, to incorporate CDOM in an ecosystem model of the NW Mediterranean Sea, one must consider the composition of DOM produced by phytoplankton in addition to the cycling of

dissolved matter and the microbial loop in the area.

We approached this challenge by using the Biogeochemical Flux Model (BFM, Vichi et al. 2007) in a one-dimensional configuration. BFM was coupled to the General Ocean Turbulence Model (GOTM, Burchard et al., 1999), a 1D water column turbulent kinetic energy model, by using the Framework for Aquatic Biogeochemical Models (FABM, Bruggeman and Bolding, 2014). In such configuration, the model accounts for the vertically differentiated processes of photobleaching and

transport from deep inventories. Local CDOM production includes the excretion of phytoplankton, which depends on light and nutrient availability, the bacterial production and consumption, and zooplankton activities. The version of BFM used here resolves the spectral light transmission underwater (Terzić et al., 2021; Lazzari et al., 2021a) and simulates the inherent optical properties (IOPs) of CDOM, phytoplankton and organic detritus in the water column. Therefore, model output can be compared with a range of optical observations made in the field and remotely.

With this model configuration, we simulated a data-rich monitoring site in the NW Mediterranean. The *Bouée pour l'acquisition de Séries Optiques à Long Terme* (BOUSSOLE) observatory is located in the Ligurian Sea, about 32 nautical miles off the French coast (Antoine et al., 2006). In this area, the water depth varies between 2350 and 2500 m and the currents are extremely low. This peculiarity is due to its location near the center of the cyclonic circulation that characterizes the Ligurian Sea, and it makes reasonable the approximation of a 1D configuration. The BOUSSOLE observatory consists of a

mooring where a buoy, specifically designed for collecting radiometric and bio-optical quantities, collects optical data at high temporal resolution. Oceanographic cruises visit the site monthly to collect a complementary dataset of algal pigment concentrations and IOPs, which include light absorption coefficients of phytoplankton, non-algal particles and CDOM (Antoine et al., 2006). Numerous studies combining optics and biochemistry have exploited these two complementary sets of observations collected at the BOUSSOLE site, from the use of fluorescence to infer Chl-a (e.g., Bayle et al., 2015; Mignot et

al., 2011) and phytoplankton community composition (Sauzède et al., 2015), inferring phytoplankton community size structure from measured light spectra (Organelli et al., 2013) and estimating community production from particle backscattering coefficients (Barbieux et al., 2022; Barnes and Antoine, 2014). One-dimensional models developed or employed at the BOUSSOLE site have focused separately on physics and biochemistry (e.g., D'Ortenzio et al., 2008; Ulses et al., 2016) or optics (Blum et al., 2012), but to our knowledge no spectrally resolved BGC model has been previously used to simulate the

site.

Methods that merge models and observations can make significant progress in reducing model uncertainty (e.g., Teruzzi et al., 2021; Ciavatta et al., 2018) and data-intensive processes such as model calibration and validation are of utmost importance to



make model applications reliable (Arhonditsis and Brett, 2004). A major challenge when working with complex coupled hydrodynamic-BGC numerical models is the need to make an appropriate choice of model parameter values that ensure optimal

performance in terms of reproducing observational data. Genetic algorithms are commonly used by BGC modellers to solve nonlinear optimization problems such as parameter estimation (e.g., Kriest et al., 2017; Kuhn and Fennel, 2019; Falls et al., 2022; Wang et al., 2020). In this work, we calibrated the model by using the Parallel Sensitivity Analysis and Calibration utility (ParSAC, Bruggeman and Bolding, 2020), which implements the Differential Evolution (DE) algorithm (Storn and Price, 1997). The DE algorithm is a population-based stochastic parallel direct search method, which operates through genetic

operations, namely mutation, crossover, and selection to eventually guide the population to the most likely values of model parameters based on the agreement of model output with observations.

With these three elements, the 1D hydrodynamic-BGC-optical model (GOTM-(FABM)-BFM), the observations collected at BOUSSOLE and the optimization tool, we propose and optimize a novel formulation for the simulation of CDOM at the BOUSSOLE site. The specific objective of the present study is to improve the representation of the variability of $a_{CDOM}(\lambda)$

within a coupled BGC-optical model of the ocean by (i) linking biodegradable CDOM produced by phytoplankton to both nutrient availability and light intensity, and (ii) improving the representation of the sources of allochthonous CDOM. We performed an optimization of optical and BGC parameters to improve the representation of the observed variables at the BOUSSOLE site, and investigated the model simulation of the unobserved variables (e.g. DOC, bacterial concentration) and bio-optical relationships between $a_{CDOM}(\lambda)$ and Chla and DOC. Based on the optimized model configuration, we conducted

two experiments to gain new insights into: i) the BGC processes that determine the seasonal dynamics of CDOM at the surface and at the depth of the Chl-a maximum (DCM), and ii) the role of local production versus allochthonous inputs in the total pool of CDOM.





## 2 Methods

The model tested in the present work is a one-dimensional (depth) configuration of the BFM, extended here with a bio-optical
component, as described in Section **2.1**. The observational data, which are explained in detail in Section **2.2**, come from the
BOUSSOLE project. A test case of the BOUSSOLE site in the NW Mediterranean from surface to 2400 m is used, the setup
of the site is described in Section **2.3**. The optimization strategy is described in Section **2.4**, which also outlines the details of
the experiments.

### 2.1 Model description

The modelling framework consisted of four elements: i) a hydrodynamic model that simulates the vertical mixing of BGC
variables, ii) a coupling layer that connects the hydrodynamic model to the BGC model, iii) a BGC model that simulates the
sink and source terms of BGC variables, and iv) a bio-optical model that resolves the underwater field of spectral light, based
on the presence of optical constituents in the water column.

#### 2.1.1 The hydrodynamic model GOTM and the framework FABM

The General Ocean Turbulence Model (GOTM, Burchard et al., 1999) is a one-dimensional water column build around an
extensive library of turbulence closure models. GOTM is very configurable and simulates the vertical structure of the water
column – notably saline, thermal and turbulence dynamics. Being a 1D model, horizontal gradients at the application sites
need to be either negligible or parameterized. A key advantage of a 1D model is that it is feasible to make long simulations
with high vertical resolution as the computational resources required are much smaller compared to full 3D models. The open-
source, Fortran-based Framework for Aquatic Biogeochemical Models (FABM, Bruggeman and Bolding, 2014) provides a
coupling layer that enables the flexible coupling of ecosystem processes into GOTM. FABM enables complex BGC models
to be developed as sets of stand-alone, process-specific modules. These are combined at runtime through coupling links to
create a custom-made BGC/ecosystem model. At each simulation time step, the BGC equations are applied to each layer and
the rates of sink and source terms calculated at current time and space using local variables (e.g., light, temperature and
concentrations of state variables) provided by GOTM. The rates pass to the hydrodynamic model via FABM, that handles
numerical integration of the BGC processes and the transport of BGC substances (e.g., nutrients, dissolved and particulate
organic matter) between the layers. Updated states are then passed back to the BGC model via FABM.

The implementation of the BFM in FABM comprises 54 state variables. These include representations of dissolved inorganic
carbon ($O^{(3)}$), inorganic forms of nitrogen ($N^{(3)}$), ammonium ($N^{(4)}$) and phosphorus ($N^{(1)}$), four phytoplankton types ($P^{(1)}$ to
$P^{(4)}$), heterotrophic bacteria (B) and four grazers ($Z^{(3)}$ to $Z^{(6)}$), three pools of dissolved organic matter ($R^{(1)}$ to $R^{(3)}$) and CDOM
($X^{(1)}$ to $X^{(3)}$) differentiated by reactivity, and particulate organic matter ($R^{(6)}$). A subscript appended to each module indicates
the elemental constituents among carbon (C), nitrogen (N), phosphorus (P) and Chl-a. In the spectrally resolved version of
BFM used in this work, the transmission of light in the water column is resolved in 33 wavebands centred from 250 to 3700
nm, with 25 nm spectral resolution within the visible range (Terzić et al., 2021; Lazzari et al., 2021a). Vertically resolved
irradiance is absorbed and scattered by water, phytoplankton, CDOM and detritus. Although the code implementation involved
a redesign of the BFM code into a FABM-compliant modular structure, the core of the overall conceptual model of the
spectrally resolved BFM remains intact compared to previous applications (e.g., Lazzari et al., 2021b; Álvarez et al., 2022)
and described below. The sources for the code of GOTM, FABM and the spectrally resolved BFM adapted to work under the
FABM convention can be found in the Code Availability Section.

#### 2.1.2 The BGC model BFM and the bio-optical component


A complete record of the partial differential equations for each state variable that conform BFM can be found in several
publications (Vichi et al., 2007a, b) as well as the details of the spectrally resolved version (Terzić et al., 2021; Lazzari et al.,





2021a). The following sections describe the main dynamics of the cycling of phytoplankton functional types (PFTs), DOC and CDOM (Sect. 2.1.2.1), the treatment of OAC (PFTs, CDOM and detritus) (Sect. 2.1.2.2) and the solution for light transmission
(Sect. 2.1.2.3).

*Cycling of BGC variables: PFTs, DOC and CDOM*

Primary producers are divided into four types that roughly represent the functional spectrum of phytoplankton in marine systems: diatoms ($P^{(1)}$), nanoflagellates ($P^{(2)}$), picophytoplankton ($P^{(3)}$) and dinoflagellates ($P^{(4)}$). The variation in carbon concentration (mg C m$^{-3}$ d$^{-1}$), being i each of the PFTs, is calculated as follows:

$$dP_C^{(i)}/dt = \left.\frac{dP_C^{(i)}}{dt}\right|_{O^{(3)}}^{gpp} - \left.\frac{dP_C^{(i)}}{dt}\right|_{R_C^{(2)}}^{dpp} - \left.\frac{dP_C^{(i)}}{dt}\right|_{O^{(3)}}^{rsp} - \sum_{j=3,4,5,6}\left.\frac{dP_C^{(i)}}{dt}\right|_{Z_C^{(j)}}^{prd} ; \text{i = 1 to 4} \tag{1}$$

Here, we briefly describe spectrally resolved gross (*gpp*) and dissolved primary production (*dpp*), whereas the exact formulations for the respiration (*rsp*) and predation (*prd*) processes can be found in the BFM manual (Vichi et al., 2020). Gross primary production of each phytoplankton species (mg C m$^{-3}$ d$^{-1}$) is computed as follows:

$$\left.\frac{dP_C^{(i)}}{dt}\right|_{O^{(3)}}^{gpp} = r_{0_P}^{(i)} \times ft_P^{(i)} \times fn_P^{(i)} \times \left(1 - \exp\left(\frac{-\phi^{(i)}\times\theta^{(i)}\times\int_{387.5}^{800}a_{PS}^{*(i)}(\lambda)\times E_0(\lambda)d\lambda}{r_{0_P}^{(i)}}\right)\right) \times P_C^{(i)} \tag{2}$$

where $r_{0_P}$ is the maximum productivity rate (d$^{-1}$), regulated by temperature ($ft_P$, dimensionless), nutrients ($fn_P$, dimensionless) and light through an increasing, saturating function of the number of photons absorbed per Chl-a unit, the
photochemical efficiency ($\phi$, mg C µE$^{-1}$) and the Chl-a ($P_{Chl}^{(i)}$) to carbon ($P_C^{(i)}$) ratio (θ, mg Chl mg C$^{-1}$). Total absorbed irradiance is the integral of the Chl-a-specific absorption spectrum of photosynthetic pigments [$a_{PS}^{*}(\lambda)$] times $E_0(\lambda)$ (see section 2.1.2.3 and Eq. (26)) between 387.5 and 800 nm (µmol quanta mg Chla$^{-1}$ d$^{-1}$). The photoacclimation term used in the calculation of $P_{Chl}^{(i)}$ is resolved spectrally in the same way and the exact formulation can be found in Álvarez et al. (2022).

DOC in BFM is characterized by reactivity into labile ($R_C^{(1)}$), semi-labile ($R_C^{(2)}$) and semi-refractory ($R_C^{(3)}$), and, accordingly,
CDOM is represented by the same three reactivity levels ($X_C^{(1)}$ to $X_C^{(3)}$). CDOM is calculated as a fraction of the DOC produced by phytoplankton that only produces $R_C^{(2)}$, zooplankton that only produces $R_C^{(1)}$ and bacteria that only produces $R_C^{(3)}$ (green lines in **Fig. 1**, all in mg C m$^{-3}$ d$^{-1}$).

With respect to the semi-labile DOC produced by phytoplankton ($R_C^{(2)}$), the BFM considers two components: i) a constant percentage of carbon fixed by phytoplankton is lost from cells each day, along with dissolved organic phosphorus and nitrogen
(DOP and DON respectively) and ii) photosynthetic overflow under conditions of nutrient limitation leads to the loss of photosynthesized carbon, which cannot be assimilated into biomass due to the absence of other nutrients. Following Thornton's (2014) definitions, the first process is defined as the passive 'leakage' of molecules through the cell membrane and the second process of 'exudation' is defined as the loss of excess carbon due to changes in environmental conditions such as nutrient availability. Thus, the total amount of phytoplankton *dpp* in BFM is a dynamic term that changes in response to environmental
conditions and is calculated as follows:

$$\left.\frac{dP_C^{(i)}}{dt}\right|_{R_C^{(2)}}^{dpp} = \beta_P^{(i)} \times \left.\frac{dP_C^{(i)}}{dt}\right|_{O^{(3)}}^{gpp} + (G_P - G_P^{balance}) \tag{3}$$

where the first addend in the right-hand side of Eq. (3) represents the passive leakage of molecules across the cell membrane and is considered a fraction ($\beta_P$) of the *gpp*; and the second addend represents the difference between the carbon assimilated into biomass (G$_P$) and the total photosynthesized carbon in the case of intracellular nutrient deficiency ($G_P^{balance}$). G$_P$ is calculated as *gpp* minus *dpp* and *rsp*, and $G_P^{balance}$ is calculated as:





$$G_P^{balance} = \min\left(G_P, \frac{1}{p_P^{min}} \times \frac{dP_P}{dt}\bigg|_{N^{(1)}}^{upt}, \frac{1}{n_P^{min}} \times \sum_{k=3,4} \frac{dP_N^{(k)}}{dt}\bigg|_{N^{(k)}}^{upt}\right) \tag{4}$$

where $p_P^{min}$ and $n_P^{min}$ are the minimum phosphorous and nitrogen quota, respectively. The exact formulations for the processes of phosphorous and nitrogen compounds uptake (*upt*) can be consulted in Vichi et al. (2020).

To investigate what fraction of these two $R_C^{(2)}$ fluxes is chromophoric and whether this proportion varies with environmental conditions, we included light and nutrient dependencies on phytoplankton CDOM production. Assuming that modelled exudation is dominated by non-chromophoric carbohydrates, under decreasing conditions of nutrient availability where

exudation accounts for a larger proportion of *dpp*, the simulated proportion of chromophoric material in total *dpp* decreases. On the other hand, with a modelled leakage that contains more chromophoric material when cells have a higher intracellular pigment concentration, the simulated chromophoric fraction of the total *dpp* increases with decreasing light availability. Thus, whereas the DOC leaked constantly from phytoplankton cells is considered to have an amount that is chromophoric and proportional to cell pigmentation ($f_P^{X2} \times \theta/\theta_{chl}^0$), DOC exuded under intra-cellular nutrient shortage is considered non-

absorbing, and therefore it is not partitioned into CDOM. The coloured fraction of the total DOC of phytoplankton origin ($f_{R2}^{X2}$) depends on the relative proportions of DOC originated through leakage or exudation and on the intracellular concentration of Chl-a, as follows:

$$f_{R2}^{X2} = f_{P(i)}^{X2} \times \frac{\theta^{(i)}}{\theta_{chl}^{0(i)}} \times \frac{\beta_P^{(i)} \times \frac{dP_C^{(i)}}{dt}\bigg|_{O^{(3)}}^{gpp}}{\frac{dP_C^{(i)}}{dt}\bigg|_{R_C^{(2)}}^{dpp}} \tag{5}$$

The non-absorbing part of this flux ($1 - f_{R2}^{X2}$) is directed to the state variable $R_C^{(2)}$ and consumed by bacteria (*upt*):

$$\frac{dR_C^{(2)}}{dt} = \sum_{i=1,2,3,4}\left(\frac{dP_C^{(i)}}{dt}\bigg|_{R_C^{(2)}}^{dpp} \times (1 - f_{R2}^{X2})\right) - \frac{dB_C}{dt}\bigg|_{R_C^{(2)}}^{upt} \tag{6}$$

CDOM is directed to the state variable $X_C^{(2)}$, consumed by bacteria (*upt*) and photobleached (*deg*):

$$dX_C^{(2)}/dt = \sum_{i=1,2,3,4}\left(\frac{dP_C^{(i)}}{dt}\bigg|_{R_C^{(2)}}^{dpp} \times f_{R2}^{X2}\right) - \frac{dB_C}{dt}\bigg|_{R_C^{(2)}}^{upt} - \frac{dX_C^{(2)}}{dt}\bigg|_{R_C^{(3)}}^{deg} \tag{7}$$

The formulations for the processes of bacterial consumption (*upt*) can be consulted in Vichi et al. (2020).

Labile DOC ($R_C^{(1)}$) is produced by the excretion of $Z^{(5)}$ and $Z^{(6)}$ (*exc*), and therefore represents the sources of DOC associated to the zooplankton-mediated mortality of phytoplankton and bacteria, and it is consumed quickly by bacteria. Large zooplankton excretion, on the other hand, is composed by particulate only and directed to the state variable $R_C^{(6)}$.

$$dR_C^{(1)}/dt = \sum_{j=5,6}\left(\frac{dZ_C^{(j)}}{dt}\bigg|_{R_C^{(1)}}^{exc} \times (1 - f_{Z(j)}^{X1})\right) - \frac{dB_C}{dt}\bigg|_{R_C^{(1)}}^{upt} \tag{8}$$

Labile CDOM ($X_C^{(1)}$) is explicitly resolved as a constant fraction of *exc* determined by the parameter $f_{Z(5)}^{X1}$ for

microzooplankton and $f_{Z(6)}^{X1}$ for nanoheterotrophs:

$$dX_C^{(1)}/dt = \sum_{j=5,6}\left(\frac{dZ_C^{(j)}}{dt}\bigg|_{R_C^{(1)}}^{exc} \times f_{Z(j)}^{X1}\right) - \frac{dB_C}{dt}\bigg|_{R_C^{(1)}}^{upt} - \frac{dX_C^{(1)}}{dt}\bigg|_{R_C^{(3)}}^{deg} \tag{9}$$

The formulations for the processes of zooplankton excretion (*exc*) can be consulted in Vichi et al. (2020).



The production of DOC by phytoplankton and as a by-product of zooplankton activities is an important driver of secondary production by heterotrophic prokaryotes. Labile and semi-labile DOC, both non-absorbing ($R_C^{(1)}$ and $R_C^{(2)}$) and coloured ($X_C^{(1)}$ and $X_C^{(2)}$) are consumed (*upt*) by bacteria and excreted (*exc*) in the form of semi-refractory DOC ($R_C^{(3)}$) resulting in

bacterial DOC recycling that contributes to the sequestration of organic carbon:

$$dR_C^{(3)}/dt = \frac{dB_C}{dt}\Big|_{R_C^{(3)}}^{exc} \times (1 - f_B^{X3}) - \frac{dR_C^{(3)}}{dt}\Big|_{O^{(3)}}^{rem} \tag{10}$$

Part of this recycled DOC is coloured and therefore $X_C^{(3)}$ is explicitly resolved as a constant fraction of $R_C^{(3)}$ production determined by the parameter $f_B^{X3}$:

$$dX_C^{(3)}/dt = \frac{dB_C}{dt}\Big|_{R_C^{(3)}}^{exc} \times f_B^{X3} - \frac{dX_C^{(3)}}{dt}\Big|_{R_C^{(3)}}^{deg} - \frac{dX_C^{(3)}}{dt}\Big|_{O^{(3)}}^{rem} \tag{11}$$

Neither $R_C^{(3)}$ nor $X_C^{(3)}$ are explicitly consumed by bacteria, but both are remineralized (*rem*) at a very slow temperature-controlled ($Q_{10_R}$) rate ($r_R$, d$^{-1}$):

$$\frac{dR_C^{(3)}}{dt}\Big|_{O^{(3)}}^{rem} = Q_{10_R}^{\frac{T-10}{10}} \times r_R \times R_C^{(3)} \; ; \frac{dX_C^{(3)}}{dt}\Big|_{O^{(3)}}^{rem} = Q_{10_R}^{\frac{T-10}{10}} \times r_R \times X_C^{(3)} \tag{12}$$

Unlike DOC, all three pools of CDOM are photobleached (*deg*) at a rate that reaches a maximum defined by the parameter $b_{X(i)}$ (d$^{-1}$) when PAR is above the threshold defined by the parameter $I_{X(i)}$ (µmol quanta m$^{-2}$ d$^{-1}$) and decreases linearly at lower PAR values (Dutkiewicz et al., 2015):

$$\frac{dX_C^{(i)}}{dt}\Big|_{R_C^{(3)}}^{deg} = b_{X(i)} \times \min\left(PAR/I_{X(i)}, 1\right) \times X_C^{(i)}; \text{i = 1 to 3} \tag{13}$$

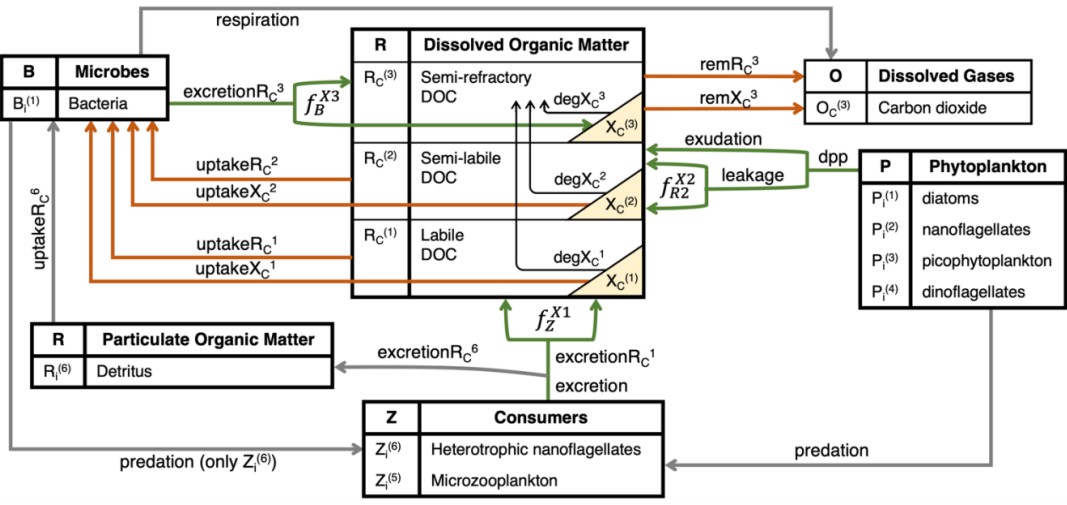


**Figure 1. Schematic representation of the hypothesized interactions regulating the concentrations of CDOM and DOC in the surface layers of the open waters of NW Mediterranean Sea. Boxes show BFM state variables and yellow triangles show the three state variables that represent three reactivities of CDOM. Green arrows indicate fluxes of DOC and CDOM production (excretionR$_C$$^1$,**
**excretionR$_C$$^3$, exudation and leakage; mg C m$^{-3}$ d$^{-1}$), orange arrows indicate fluxes of DOC and/or CDOM degradation through direct bacterial consumption (uptakeR$_C$$^1$, uptakeX$_C$$^1$, uptakeR$_C$$^2$, uptakeX$_C$$^2$; mg C m$^{-3}$ d$^{-1}$) or indirect remineralization (remR$_C$$^3$, remX$_C$$^3$; mg C m$^{-3}$ d$^{-1}$) and black arrows indicate photobleaching processes (degX$_C$$^1$, degX$_C$$^2$, degX$_C$$^3$; mg C m$^{-3}$ d$^{-1}$). The names in italics ($f_Z^{X1}$, $f_{R2}^{X2}$ and $f_B^{X3}$; dimensionless) represent the fractions that divide the flux between CDOM and non-coloured DOC. The subscript $i$ appended to each living component indicates the elemental constituents among carbon (C), nitrogen (N), phosphorus (P)**
**and Chl-a.**



*Bio-optical properties of CDOM, detritus and phytoplankton*

The optical constituents of the spectrally resolved version of BFM are: the four PFTs, detrital particles and the three forms of CDOM corresponding to their resistance to bacterial metabolic activity. CDOM absorbs light and has negligible contribution to scattering, therefore CDOM absorption at 450 nm, $a_{CDOM}(450)$ (m$^{-1}$), is first calculated as the product of CDOM biomass

($X_C^{(i)}$, mg C m$^{-3}$) and the mass-specific absorption coefficients at 450 nm ($a_{X(i)}^{450}$, m$^2$ mg C$^{-1}$). CDOM absorption as a function of wavelength ($\lambda$), $a_{CDOM}(\lambda)$, is modelled with an exponential function, decreasing with increasing $\lambda$:

$$a_{CDOM}(\lambda) = \sum_{i=1}^{3} a_{X(i)}^{450} \times exp[-S_{X(i)}^{350-500} \times (\lambda - 450)] \times X_C^{(i)} \tag{14}$$

where $S_{X(i)}^{350-500}$ is the spectral slope of the CDOM absorption coefficients between 350 and 500 nm.

Detritus absorbs and scatters light. Detritus absorption at 440 nm ($a_{NAP}(440)$, m$^{-1}$) is calculated as the product of the carbon component of detritus ($R_C^{(6)}$, mg C m$^{-3}$) and their mass-specific absorption coefficients at 440 nm ($a_R^{440}$, m$^2$ mg C$^{-1}$), then the

absorption spectrum of detritus is modelled as an exponential function which decrease with increasing $\lambda$:

$$a_{NAP}(\lambda) = a_R^{440} \times exp[-S_R^{350-500} \times (\lambda - 440)] \times R_C^{(6)} \tag{15}$$

where $S_R^{350-500}$ is the spectral slope of the detritus absorption coefficients, which we have set to 0.013 nm$^{-1}$ (**Table 1**). Detrital scattering at 550 nm ($b_{NAP}(550)$, m$^{-1}$) is calculated as the product of $R_C^{(6)}$ and the mass-specific scattering coefficient at 550 nm ($b_R^{550}$, m2 mg C$^{-1}$), then the scattering spectrum of detritus, $b_{NAP}(\lambda)$, is computed as an exponential function of $\lambda$:

$$b_{NAP}(\lambda) = b_R^{550} \times (550/\lambda)^{e_R} \times R_C^{(6)} \tag{16}$$

where $e_R$ is an exponent set to 0.5 (**Table 1**).

To describe the optical properties of phytoplankton, we used Chl-a-specific absorption spectra ($a_{PH}^*(\lambda)$) of different phytoplankton taxa growing in culture under different light, nutrient supply, and temperature conditions. We considered 177 spectra, digitized from literature, and provided as supplementary material in Álvarez et al. (2022). Each $a_{PH}^*(\lambda)$ was decomposed into the contribution of Chl-a, photosynthetic carotenoids, photoprotective carotenoids (PPC) and a fourth pigment group that was phycoerythrin for *Synechococcus* species, Chl-b for green algae taxa and *Prochlorococcus* species and

Chl-c for the other taxa. To obtain the relative pigment concentrations for each pigment group we fitted a multiple linear regression model to the weight specific absorption spectra for each pigment group obtained from Bidigare et al. (1990), in order to obtain the lowest sum of residuals between the predicted and the measured $a_{PH}^*(\lambda)$ (Hickman et al., 2010). The Chl-a-specific absorption spectra for photosynthetic pigments only ($a_{PS}^*(\lambda)$) were reconstructed using the obtained relative concentrations excluding PPC (Babin et al., 1996). From this collection of spectra, $a_{PH}^*(\lambda)$ and $a_{PS}^*(\lambda)$ were averaged for

picophytoplankton, nanophytoplankton, diatom and dinophyta taxa. The resulting four $a_{PH}^*(\lambda)$ were used to calculate total absorption by phytoplankton ($a_{PH}(\lambda)$, m$^{-1}$) as the sum of the products of PFTs Chl-a ($P_{Chl}^{(i)}$, mg Chl-a m$^{-3}$) and the Chl-a-specific absorption coefficients (m$^2$ mg Chl-a$^{-1}$):

$$a_{PH}(\lambda) = \sum_{i=1}^{4} a^{*(i)}_{PH}(\lambda) \times P_{Chl}^{(i)} \tag{17}$$

Scattering coefficients of phytoplankton were assumed to be functions of phytoplankton biomass ($P_C^{(i)}$, mg C m$^{-3}$) and the C-specific spectra taken from Dutkiewicz et al. (2015). Scattering coefficients by all phytoplankton ($b_{PH}(\lambda)$, m$^{-1}$) were calculated

as the sum of the product of phytoplankton biomass and the C-specific scattering spectrum for each of the four PFTs (m$^2$ mg C$^{-1}$):



$$b_{PH}(\lambda) = \sum_{i=1}^{4} b^{*(i)}_{PH}(\lambda) \times P_C^{(i)} \qquad (18)$$

Phytoplankton absorption cross-section for all pigments ($\bar{a}^*_{PH}$, m² mg Chl-a⁻¹), photosynthetic pigments ($\bar{a}^*_{PS}$, m² mg Chl-a⁻¹) and scattering cross-section ($\bar{b}^*_{PH}$, m² mg Chl-a⁻¹) were computed as the average of $a^*_{PH}(\lambda)$, $a^*_{PS}(\lambda)$ and $b^*_{PH}(\lambda)$ between 387.5 and 800 nm, respectively. These aggregated values were used to perturb the magnitude of their respective spectra without

modifying the spectral shape. The computation of $\bar{a}^*_{PS}$ and $\bar{b}^*_{PH}$ is equivalent to Eq. (19), substituting $a^*_{PH}(\lambda)$ by $a^*_{PS}(\lambda)$ and $b^*_{PH}(\lambda)$, respectively:

$$\bar{a}^*_{PH} = \int_{387.5}^{800} a^*_{PH}(\lambda) \; d\lambda / (800 - 387.5) \qquad (19)$$

Total absorption, scattering and backscattering, $a(\lambda)$, $b(\lambda)$ and $bb(\lambda)$ respectively, depend on the additive contribution of seawater and the BGC constituents along the water column. Total $a(\lambda)$ (m⁻¹) is calculated from the absorption by water ($a_W(\lambda)$, m⁻¹, Pope and Fry (1997)), CDOM, detritus and phytoplankton:

$$a(\lambda) = a_W(\lambda) + a_{CDOM}(\lambda) + a_{NAP}(\lambda) + a_{PH}(\lambda) \qquad (20)$$

Total $b(\lambda)$ (m⁻¹) is calculated as the sum of the scattering of water ($b_W(\lambda)$, m⁻¹, Smith and Baker (1981)), detritus and phytoplankton:

$$b(\lambda) = b_W(\lambda) + b_{NAP}(\lambda) + b_{PH}(\lambda) \qquad (21)$$

Total $bb(\lambda)$ (m⁻¹) is computed as the sum of each constituent's scattering multiplied by constant and $\lambda$ independent backscattering to scattering ratios ($bbr_W$, $bbr_R$ and $bbr_{PH}^{(i)}$), that were: 0.5 for water (Morel, 1974), 0.005 for detritus (Gallegos et al., 2011), 0.002 for P⁽¹⁾ (Dutkiewicz et al., 2015), 0.0071 for P⁽²⁾ (Gregg and Rousseaux, 2017), 0.0039 for P⁽³⁾ (Gregg and

Rousseaux, 2017; Dutkiewicz et al., 2015) and 0.003 for P⁽⁴⁾ (Dutkiewicz et al., 2015) (**Table 1**):

$$bb(\lambda) = b_W(\lambda) \times bbr_W + b_{NAP}(\lambda) \times bbr_R + \sum_{i=1}^{4} b^{*(i)}_{PH}(\lambda) \times P_C^{(i)} \times bbr_{PH}^{(i)} \qquad (22)$$

*Spectrally resolved light transmission and computation of $E_0(\lambda,z)$*

Incoming spectral irradiance at the top of the ocean was obtained from the OASIM model interfaced with the European Centre for Medium-Range Weather Forecast (ECMWF) atmospheric model (Lazzari et al., 2021a). Input data contained two downward streams just below the surface of the ocean: direct [$E_d(\lambda,0^-)$] and diffuse [$E_s(\lambda,0^-)$], both in W m⁻² and averaged at

33 wavebands. Irradiance along the depth of the water column (z) was described with three state variables, all in W m⁻²: downwelling direct [$E_d(\lambda,z)$] and diffuse [$E_s(\lambda,z)$] components and an upward diffuse component [$E_u(\lambda,z)$]. The vertically resolved three-stream propagation was resolved by the following system of differential equations (Aas, 1987; Ackleson et al., 1994; Gregg, 2002):

$$dE_d(\lambda,z)/dz = -\frac{a(\lambda,z) + b(\lambda,z)}{v_d} \times E_d(\lambda,z) \qquad (23)$$

$$dE_s(\lambda,z)/dz = -\frac{a(\lambda,z) + r_s \times bb(\lambda,z)}{v_s} \times E_s(\lambda,z) + \frac{r_u \times bb(\lambda,z)}{v_u} \times E_u(\lambda,z)$$
$$+ \frac{b(\lambda,z) - r_d \times bb(\lambda,z)}{v_d} \times E_d(\lambda,z) \qquad (24)$$

$$-dE_u(\lambda,z)/dz = -\frac{a(\lambda,z) + r_u \times bb(\lambda,z)}{v_u} \times E_u(\lambda,z) + \frac{r_s \times bb(\lambda,z)}{v_s} \times E_s(\lambda,z) + \frac{r_d \times bb(\lambda,z)}{v_d} \times E_d(\lambda,z) \qquad (25)$$



where $a(\lambda, z)$ is the total absorption (Eq. (20)), $b(\lambda, z)$ is the total scattering (Eq. (21)), and $bb(\lambda, z)$ is the total backward

scattering (Eq. (22)) at depth z, all in m$^{-1}$. r$_s$, r$_u$ and r$_d$ are the effective scattering coefficients, and $v_d$, $v_s$ and $v_u$ are the average

cosines of the three streams, which are constant for diffuse radiance but vary with solar zenith angle for direct radiance. For

details on the derivation of the solution see Terzić et al. (2021).

At the center of the depth layer, E$_d$(λ,z), E$_s$(λ,z) and E$_u$(λ,z) were averaged between the top and the bottom of the layer and

total scalar irradiance, E$_0$(λ,z), was computed by scaling the three streams by their inverse average direction cosines. E$_0$(λ,z)

was converted from irradiance values in units of W m$^{-2}$ to photon flux given in µmol quanta m$^{-2}$ d$^{-1}$ by multiplying by λ in

meters, dividing by the product of the Avogadro's constant (N$_A$), Planck's constant (h) and speed of light (c) and converting

mol quanta to µmol quanta and s$^{-1}$ to d$^{-1}$:

$$E_0(\lambda, z) = [E_d(\lambda, z)/\overline{v_d} + E_s(\lambda, z)/\overline{v_s} + E_u(\lambda, z)/\overline{v_u}] \times \frac{\lambda}{h \cdot c \cdot N_A} \times \frac{1m}{10^9 nm} \times \frac{10^6 \mu mol\ quanta}{1 mol\ quanta} \times \frac{86400s}{1\ day} \qquad (26)$$

E$_0$(λ,z) was the light available for phytoplankton growth (Eq. (2)) at depth z, and its integral value from 387.5 and 800 nm

constituted the photosynthetically available radiation (PAR) that was used as the light input for CDOM photobleaching (Sect.

2.1.2.1, Eq. (13)).



## 2.2 Observations

The Mediterranean Sea is a semi-enclosed marginal sea. The Ligurian sea, in the NW Mediterranean Sea, is characterized by oligotrophic to moderately eutrophic waters. The BOUSSOLE and the DYFAMED monitoring sites, both in the Ligurian sea, provided bio-optical and physical-chemical measurements, respectively (Sect. 2.2.1). In addition, we considered ocean-colour data collected by satellites (Sect. 2.2.2) (**Fig. 2**).

### 2.2.1 In situ data at BOUSSOLE and DYFAMED sites

The BOUSSOLE project includes a permanent optical mooring deployed at 7°54'E, 43°22'N, where the depth is 2440 m, collecting optical data at high temporal resolution, and oceanographic cruises visiting the site monthly to collect optical and BGC variables (Golbol et al., 2000). Only data from monthly cruises were used in the present work. Seawater is generally collected at 12 discrete depths within the 0-400 m water column (5, 10, 20, 30, 40, 50, 60, 70, 80, 150, 200, 400 m). Sampling is performed with a SeaBird SBE911+ CTD/rosette system equipped with pressure (Digiquartz Paroscientific), temperature (SBE3+) and conductivity (SBE4) sensors and 12-L Niskin bottles, from which water samples are collected for subsequent absorption measurements and high-performance liquid chromatography (HPLC) analyses. The CDOM absorption measurements are described in Organelli et al. (2014) and references therein. We only remind here that the spectral slope of $a_{\text{CDOM}}(\lambda)$ was determined between 350 to 500 nm ($S_{CDOM}^{350-500}$), although shorter UV $\lambda$ intervals have also been used to report spectral slopes in the Mediterranean Sea (275–295 nm, Catalá et al., 2018). Particulate light absorption spectra, $a_{\text{P}}(\lambda)$, were measured using the "quantitative filter pad technique (QFT)" (Mitchell, 1990; Mitchell et al., 2000), and the protocol is described in Antoine et al. (2006). $a_{\text{P}}(\lambda)$ was decomposed into phytoplankton and detrital absorption coefficients, $a_{\text{PH}}(\lambda)$ and $a_{\text{NAP}}(\lambda)$ respectively, using the numerical decomposition technique of Bricaud and Stramski (1990). The HPLC procedure used to measure phytoplankton pigment concentrations is fully described in Ras et al. (2008). The fractions of picophytoplankton, nanophytoplankton and microphytoplankton were calculated following the diagnostic pigments method (Uitz et al., 2006).

The *Dynamique des Flux Atmosphériques en Méditerranée* (DYFAMED) time-series station (Marty, 2002) is located at 7.54°E and 43.25°N, where data have been collected since 1991. Temperature (T) and salinity (S) data were collected within the 0-2400 m water column with the same CTD/rosette configuration used at BOUSSOLE with nutrients sampled at discrete depths. T and S data from 1994 to 2014 were downloaded from the OceanSITES Global Data Assembly Center and nitrate, nitrite and phosphate concentrations were downloaded from Coppola et al. (2021).

### 2.2.2 Data from satellites

Satellite retrieved IOPs and Chl-a were obtained from the OCEANCOLOUR_MED_BGC_L3_MY_009_143 product (E.U. Copernicus Marine Service Information, 2022). The product provides multi-sensor $R_{RS}(\lambda)$ spectra (SeaWiFS, MODIS-AQUA, NOAA20-VIIRS, NPP-VIIRS and Sentinel3A-OLCI) merged using the state-of-the-art multi-sensor merging algorithms and shifted to the standard SeaWiFS wavelengths (412, 443, 490, 555, and 670 nm). IOPs ($a_{\text{PH}}443$ and $a_{\text{DG}}443$) are derived from $R_{RS}(\lambda)$ via the Quasi-Analytical Algorithm version 6 (QAAv6) model (Lee et al., 2002). Chl-a concentration is determined using the Mediterranean regional algorithm: an updated version of MedOC4 (Volpe et al., 2019) and AD4 (Berthon and Zibordi, 2004), merged according to the water type present (Mélin and Vantrepotte, 2015; Volpe et al., 2019). For all variables, to obtain the time series from January 2011 to December 2014 at the study site, we extracted a 20x20km box around the BOUSSOLE site (from 7.77°E to 8.03°E in longitude and from 43.27°N to 43.47°N in latitude) and averaged it when available observations covered at least 20% of the area.



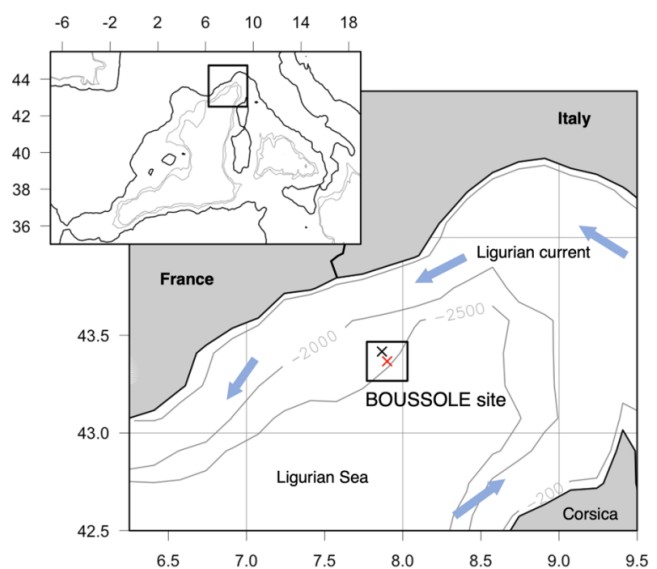


**Figure 2. Map showing the location of the BOUSSOLE mooring site (red cross) and the DYFAMED site (black cross) in the Ligurian Sea (NW Mediterranean Sea). The box surrounding the sites indicates the 20x20 km area considered for averaging data from the daily 1km CMEMS product to obtain the satellite time series at the study site.**



**2.3 Study region and set up of the 1D model for the BOUSSOLE site**

The configuration of the one-dimensional spectrally resolved version of BFM used to simulate the BOUSSOLE site has the same parameter values of the 3D configuration of BFM for the Mediterranean Sea, except for those reported in **Table 1**. Some updates comprised parameters related to the growth of nanophytoplankton ($P^{(2)}$), in particular those related to the uptake of phosphorous ($a_1$, $p_P^{min}$ and $p_P^{opt}$). This update was finalized at reproducing the high contribution of nanophytoplankton to the

total Chl-a content at the BOUSSOLE site (Antoine et al., 2006) (see **Fig. 4**). We also updated parameters for the production by phytoplankton and consumption by bacteria of semi-labile DOC ($\beta_P$ and $\nu_B^{R2}$) to simulate accurately the seasonality of DOC concentrations in superficial waters of the site (Avril, 2002) (see **Fig. 5**). The parameter values related to the partition between dissolved and particulate excretion in $Z^{(5)}$ and $Z^{(6)}$, and the mortality of $Z^{(3)}$ and $Z^{(4)}$, are assumed equal to the up-to-date values in Álvarez et al. (2022), together with all the optical parameters. All the other parameter values are equal to those

shown in Lazzari et al. (2012).

The 1D configuration of GOTM-(FABM)-BFM for the BOUSSOLE site consists of: i) a water column of 2448 m, subdivided in 196 vertical levels with higher resolution at surface and bottom, which were generated using the iGOTM toolweb (https://igotm.bolding-bruggeman.com/, accessed 12/10/2021); ii) atmospheric forcing that included: hourly 10 m winds, 2 m air temperature and humidity, sea level pressure, precipitation and cloud cover from ECMWF ERA5, atmospheric $pCO_2$ that

was set to a constant value of 400 ppm, and spectral plane downwelling irradiance at 15-minutes frequency from the OASIM model that has been validated for the BOUSSOLE site (Lazzari et al., 2021a); and iii) initial conditions for T, S and BGC variables.

Observed vertical profiles of T and S at the DYFAMED site from 2009 to 2014 were edited to match the GOTM profile-format and used as initial conditions and restored at monthly frequency, in order to reproduce the intensity and timing of the mixing

as closely as possible to observations. Monthly vertical profiles of BGC variables (nitrate, phosphate, silicate, $O_2$, DIC and alkalinity) were obtained from the Copernicus reanalysis (Cossarini et al., 2021), linearly interpolated onto the model vertical grid and used for initialization and restored at yearly frequency. All other variables were set to constant values throughout the water column. The initial values of $R_C^{(1)}$, $R_C^{(2)}$ and $R_C^{(3)}$ were 80, 400 and 480 mg C m$^{-3}$, respectively, consistent with DOC concentrations at depth in the West Mediterranean Sea (Catalá et al., 2018; Santinelli, 2015; Galletti et al., 2019). The initial

values of $X_C^{(1)}$, $X_C^{(2)}$ and $X_C^{(3)}$ were 1, 3 and 1.25 mg C m$^{-3}$, respectively. Given that below 200 m depth BFM simulates negligible concentrations of $X_C^{(1)}$ and $X_C^{(2)}$, the initial concentration of $X_C^{(3)}$ was chosen to match $a_{CDOM}(450)$ values observed at the site below 100 m (Organelli et al., 2014) and $a_{CDOM}(250)$ and $a_{CDOM}(325)$ values observed in the West Mediterranean Sea below 200 m (Catalá et al., 2018). All PFTs were initialized to 8 mg C m$^{-3}$ and 0.16 mg Chl-a m$^{-3}$. Bacteria and zooplankton were initialized to 1 mg C m$^{-3}$.

A 6-year hindcast (January 2009 to December 2014) was conducted, covering the period for which all the necessary forcing data were available. The first two years were considered as model spin-up and the next 4 years were considered for the optimization. The source for the setup can be found in the Code Availability Section.





**Table 1: Values of the BFM optical and BGC parameters that were not optimized. The parameters with source 'this study' were manually chosen in order to obtain a good approximation of model results to findings by** Avril (2002) **and** Antoine et al. (2006) **on the concentration of DOC at DYFAMED and PFTs biomass at BOUSSOLE, respectively.**

| Parameter | Description | Value | Units | Source |
|---|---|---|---|---|
| *CDOM* ($X^{(1)}$, $X^{(2)}$ and $X^{(3)}$) | | | | |
| $a_{X(1)}^{450}$ | $X^{(1)}$ mass-specific absorption at 450 nm | 0.015 | m² mg C⁻¹ | (Dutkiewicz et al., 2015) |
| $a_{X(2)}^{450}$ | $X^{(2)}$ mass-specific absorption at 450 nm | 0.015 | m² mg C⁻¹ | (Dutkiewicz et al., 2015) |
| $a_{X(3)}^{450}$ | $X^{(3)}$ mass-specific absorption at 450 nm | 0.015 | m² mg C⁻¹ | (Dutkiewicz et al., 2015) |
| $r_R$ | Remineralization rate of $R^{(3)}$ and $X^{(3)}$ | 3e-5 | d⁻¹ | (Legendre et al., 2015) |
| $Q_{10_R}$ | Characteristic $Q_{10}$ coefficient | 2.95 | - | (Lazzari et al., 2021b) |
| *Detritus* ($R^{(6)}$) | | | | |
| $S_R^{350-500}$ | $R^{(6)}$ spectral slope for absorption | 0.013 | nm⁻¹ | (Gallegos et al., 2011) |
| $e_R$ | Exponent for scattering of $R^{(6)}$ | 0.5 | - | (Gallegos et al., 2011) |
| $b_R^{550}$ | Detritus scattering at 550 nm | 0.0120 | m² mg C⁻¹ | this study |
| $bbr_R$ | Backscattering to scattering ratio of $R^{(6)}$ | 0.005 | - | (Gallegos et al., 2011) |
| $v_{R6}^{sed}$ | Settling velocity of particulate detritus | 4.25 | m d⁻¹ | this study |
| *Diatoms* ($P^{(1)}$) | | | | |
| $\bar{a}_{PS}^{*}$ | Absorption cross-section of photosynthetic pigments | 0.0130 | m² mg Chl-a⁻¹ | (Álvarez et al., 2022) |
| $\bar{b}_P^{*}$ | Scattering cross-section | 0.0140 | m² mg C⁻¹ | (Dutkiewicz et al., 2015) |
| $bbr_{PH}$ | Backscattering to scattering ratio $P^{(1)}$ | 0.0020 | - | (Dutkiewicz et al., 2015) |
| $\beta_P$ | Excreted fraction of primary production | 0.15 | - | this study |
| $b_P$ | Basal respiration rate at 10 °C | 0.076 | d⁻¹ | this study |
| $p_P^{min}$ | Minimum phosphorous quotum | 5.7e-4 | mmol P mg C⁻¹ | this study |
| *Flagellates* ($P^{(2)}$) | | | | |
| $\bar{a}_{PS}^{*}$ | Absorption cross-section of photosynthetic pigments | 0.0167 | m² mg Chl-a⁻¹ | (Álvarez et al., 2022) |
| $\bar{b}_P^{*}$ | Scattering cross-section | 0.0077 | m² mg C⁻¹ | (Dutkiewicz et al., 2015) |
| $bbr_{PH}$ | Backscattering to scattering ratio $P^{(2)}$ | 0.0071 | - | (Gregg and Rousseaux, 2017) |
| $\beta_P$ | Excreted fraction of primary production | 0.10 | - | this study |
| $b_P$ | Basal respiration rate at 10 °C | 0.09 | d⁻¹ | this study |
| $r_{0_P}$ | Maximum specific photosynthetic rate | 2.6 | d⁻¹ | this study |
| $Q_{10_P}$ | Characteristic $Q_{10}$ coefficient | 2.25 | - | this study |
| $p_P^{min}$ | Minimum phosphorous quotum | 3.5e-4 | mmol P mg C⁻¹ | this study |
| $p_P^{opt}$ | Optimal phosphorous quotum | 5.6e-4 | mmol P mg C⁻¹ | this study |
| $a_1$ | Membrane affinity for phosphorous | 0.0035 | m³ mg C⁻¹ d⁻¹ | this study |
| *Picophytoplankton* ($P^{(3)}$) | | | | |
| $\bar{a}_{PS}^{*}$ | Absorption cross-section of photosynthetic pigments | 0.0221 | m² mg Chl-a⁻¹ | (Álvarez et al., 2022) |
| $\bar{b}_P^{*}$ | Scattering cross-section | 0.0101 | m² mg C⁻¹ | (Dutkiewicz et al., 2015) |
| $bbr_{PH}$ | Backscattering to scattering ratio $P^{(3)}$ | 0.0039 | - | (Gregg and Rousseaux, 2017) |
| $\beta_P$ | Excreted fraction of primary production | 0.15 | - | this study |
| $p_P^{opt}$ | Optimal phosphorous quotum | 7.2e-4 | mmol P mg C⁻¹ | this study |
| *Dinoflagellates* ($P^{(4)}$) | | | | |
| $\bar{a}_{PS}^{*}$ | Absorption cross-section of photosynthetic pigments | 0.0163 | m² mg Chl-a⁻¹ | (Álvarez et al., 2022) |
| $\bar{b}_P^{*}$ | Scattering cross-section | 0.0112 | m² mg C⁻¹ | (Dutkiewicz et al., 2015) |
| $bbr_{PH}$ | Backscattering to scattering ratio $P^{(4)}$ | 0.0030 | - | (Dutkiewicz et al., 2015) |
| $\beta_P$ | Excreted fraction of primary production | 0.20 | - | this study |
| *Zooplankton* ($Z^{(5)}$ and $Z^{(6)}$) | | | | |
| $r_{0_{Z(5)}}$ | Potential specific growth rate of $Z^{(5)}$ | 2.71 | d⁻¹ | this study |
| $r_{0_{Z(6)}}$ | Potential specific growth rate of $Z^{(6)}$ | 3.88 | d⁻¹ | this study |
| *Bacteria* (B) | | | | |
| $\gamma_B^{a}$ | Activity respiration fraction | 0.76 | - | this study |
| $v_B^{R2}$ | Specific potential $R^{(2)}$ and $X^{(2)}$ uptake | 0.05 | d⁻¹ | this study |





**2.4 Optimization strategy and simulations for hypothesis testing**

The final step in completing the GOTM-(FABM)-BFM configuration at the BOUSSOLE site was to integrate model and observations to drive the simulated output as close as possible to the observations. Being computationally light, the one-dimensional BFM configuration has the advantage of allowing many simulations in a reasonable time which provided us a powerful tool for optimizing parameters (Sect. 2.4.1) and investigating processes based on the proposed new parameterizations (Sect. 2.4.2).

**2.4.1 Parameter optimization method**

To perform the estimation of model parameter values using observational data, we used the Parallel Sensitivity Analysis and Calibration utility (ParSAC, Bruggeman and Bolding, 2020). ParSAC applies the Differential Evolution (DE) algorithm (Storn and Price, 1997), which iteratively searches for optimal model parameters while minimizing the misfit between the model and the observations. Observations used for optimization were all collected at the BOUSSOLE site at monthly temporal resolution and roughly at the same number of discrete depths and included pico-, nano- and micro-phytoplankton Chl-a (pico-Chl-a, nano-Chl-a and micro-Chl-a, respectively), $a_{PH}(450)$, $a_{NAP}(450)$, $a_{CDOM}(450)$ and $S_{CDOM}^{350-500}$. The potential number of independent parameters included in the problem depended on the observations available. The only observations related to CDOM were $a_{CDOM}(\lambda)$ that depend both on the mass-concentration and on the mass-specific absorption coefficients across the electromagnetic spectrum. We assumed true the absorption coefficients of the three reactivities at 450 nm ($a_{X1}^{450}$, $a_{X2}^{450}$ and $a_{X3}^{450}$) and optimized parameters related to the production of CDOM ($f_{Z(5\,to\,6)}^{X1}$, $f_{P(1\,to\,4)}^{X2}$ and $f_{B}^{X3}$) and those related to the photobleaching of $X_C^{(3)}$ ($b_{X(3)}$ and $I_{X(3)}$). The spectral slope of the CDOM pool has been associated with aromaticity and average molecular weight of the CDOM compounds (Blough and Green, 1995) but appears to be less linked to CDOM concentration. Therefore, the spectral slopes between 350 and 500 nm for the three reactivities ($S_{X(1)}^{350-500}$, $S_{X(2)}^{350-500}$ and $S_{X(3)}^{350-500}$) were optimized to ensure the proper simulation of $a_{CDOM}(\lambda)$ in wavebands other than 450 nm. For phytoplankton, independent observations regarding mass-concentrations (Chl-a) and optical properties ($a_{PH}(\lambda)$) were available, therefore both types of parameters were optimized. All optimized parameters appear in some process independently: absorption cross-sections ($\bar{a}_{PH}^{*}$) in light attenuation (Eq. (17)) and the maximum Chl:C quota ($\theta_{chl}^{0}$) in photoacclimation (Eq. (19) in Álvarez et al., 2022) and CDOM production (Eq. (5)), with the exception of the absorption cross-sections of photosynthetic pigments ($\bar{a}_{PS}^{*}$) and the photochemical efficiencies ($\phi_c^0$) that both appear only in photosynthesis (Eq. (2)). We assumed true $\bar{a}_{PS}^{*}$ and optimized $\phi_c^0$, because these parameters are not well documented in literature being more difficult to measure. For detritus, we assumed true all BGC parameters that alter the mass-concentrations and optimized the reference absorption coefficients at 440 nm ($a_R^{440}$). Given that $a_{NAP}(\lambda)$ observations were available, optical parameters related to the spectral shape of absorption by detritus ($S_R^{350-500}$) could have been included in the optimization, but we decided to maintain this parameter constant given the small contribution of $a_{NAP}(\lambda)$ to the total non-water absorption as compared to CDOM and phytoplankton. All parameters related to scattering and backscattering both of phytoplankton ($\bar{b}_{PH}^{*}$ and $bbr_{PH}$) and detritus ($b_R^{550}$, $e_R$ and $bbr_R$) were assumed true. A total of 25 optical and BGC parameters were optimized (**Table A1** in Appendix A).

The DE algorithm is inspired by the rules of natural selection. Each individual in a given population (n=288) is a 25-dimensional target vector that represents a candidate solution to the problem. Different individuals have different fitness values in terms of how well they simulate observations. A higher fitness value indicates a combination of parameters that reproduce better observations and therefore must persist in the next generation. ParSAC formulates such fitness (i.e. probability that the candidate parameter values are the true parameter set representing reality) as a multi-objective function calculated as the log-transformed likelihood between the outcome of the model and the observations provided. For any set of observed data O =



$[O_1,\ldots, O_n, O_N]$ corresponding to the simulated data $P = [P_1,\ldots, P_n, P_N]$, the sum of the squares of the residuals (SSQ) is

computed as:

$$SSQ = \sum_{n=1,N} [P_n - O_n]^2 \qquad (27)$$

with N the number of pairs consisting of the simulation $P_n$ and the corresponding observation $O_n$. The residuals are assumed

to have a normal distribution with constant variance, and therefore the variance of the residuals is estimated as:

$$\sigma^2 = SSQ/(N-1) \qquad (28)$$

For M variables observed, all differences between model and observations are combined as:

$$lnLikelihood = \sum_{m=1,M} -N_m \times \ln \sigma_m - \frac{SSQ_m}{2\sigma_m^2} \qquad (29)$$

where the weight that each observation contributes to the lnLikelihood is the inverse of its variance to remove the effect of

different units and reduce the contribution of variables that the model cannot capture well.

The DE algorithm creates new generations of individuals by applying cycles of mutation, crossover and selection operators.

The target vectors in the first generation were chosen randomly across the parameter space. The minimum and maximum

values given to each parameter at the beginning of the optimization procedure are listed in **Table A1** (Appendix A). Once all

simulations in the first generation are completed, the DE algorithm creates the next generation of individuals by creating one

mutant vector for each target vector selecting randomly three individuals ($x_{r1}$, $x_{r2}$ and $x_{r3}$) and applying a mutation operator

that consisted on $x_{r1} + 0.5 \cdot (x_{r2} - x_{r3})$. All target vectors in the first generation are crossed with the mutant vector with a crossover

probability of 0.9 (the genetic characteristic of the mutant is retained with a 0.1 probability), generating one trial vector for

each target vector. The final phase is selection where the trial vector replaces the target if the lnLikelihood value obtained from

the trial is higher or equal than the lnLikelihood obtained from the target vector for a successive generation. After the simulation

of a new generation, the cycle of mutation, crossover and selection is carried out iteratively and eventually, the DE algorithm

provides an optimal set of 25-parameter values that minimizes the misfit between the model output and the observations (i.e.

maximizes the sum of lnLikelihoods in the population). After passing through each generation, ParSAC checks whether the

end condition is met, that in this work was a maximum number of 180 generations, after which the range of lnLikelihoods in

the population was less than 0.005. The optimal solution for the 25 parameter values was extracted as the mean of the parameter

distribution in generation 180. To determine the success achieved through optimisation, we calculated the correlation

coefficient (R) and root mean squared error (RMSE) for each observation compared to the model output at each generation.

### 2.4.2 Simulations and hypothesis testing experiments

Using the model configuration described in Section 2.3 and the optimised parameter values obtained as described in Section

2.4.1, we run a 6-year simulation with two years of spin-up and analysed the results of the last four years (January 2011 to

December 2014), which represents the *Optimized* model configuration. We conducted three more simulations in which we

reversed some of the assumptions of the optimized configuration. All simulations were run for the same time period as the

*Optimized* one and are summarized in **Table 2**. In the simulation *Nutrients*, we considered the proportion $X_C^{(2)}:R_C^{(2)}$ in *dpp* as

a constant fraction of leakage and therefore removed the dependence of $f_{R2}^{X2}$ on $\theta/\theta_{chl}^0$. In *Constant*, we considered the fraction

$X_C^{(2)}:R_C^{(2)}$ in *dpp* constant and therefore removed both dependencies from $f_{R2}^{X2}$, on $\theta/\theta_{chl}^0$ and on leakage/*dpp*. The comparison

of the results of *Optimized* with *Nutrients* and *Constant* constituted and experiment (EXP-X$^{(2)}$) where we assessed the extent

to which the nutrient availability and light dependencies of semi-labile CDOM production, respectively, contributed to the

generation of the observed $a_{CDOM}(\lambda)$ values. In *Bleaching,* we prescribed the same bleaching parameters for $X_C^{(3)}$ ($b_{X(3)}$ and

$I_{X(3)}$) as for the other two CDOM pools and quadrupled the CDOM:DOC fractions in all biotic sources ($f_{Z(j)}^{X1}$, $f_{P(i)}^{X2}$ and $f_B^{X3}$).

The comparison of *Optimized* with *Bleaching* constituted a second experiment (EXP-X$^{(3)}$) where we evaluated the extent to



which the *Optimized* simulation was able to represent the role of $X_C^{(3)}$, and thus of any source of allochthonous CDOM, in generating the observed $a_{CDOM}(\lambda)$ values.


**Table 2. List of simulations performed to test the findings provided by the optimized framework, indicating all formulations and/or parameters in which they differ from the optimized model configuration.**

| Question | Simulation | $X_C^{(2)}: R_C^{(2)}$ in leakage | $X_C^{(2)}: R_C^{(2)}$ in *dpp* | $f_B^{X3}$ | $f_{Z(5)}^{X1}$ | $f_{Z(6)}^{X1}$ | $b_{X(3)}$ | $I_{X(3)}$ |
|---|---|---|---|---|---|---|---|---|
| Optimized model configuration (EXP-X$^{(2)}$ & EXP-X$^{(3)}$) | *Optimized* | $f_{P(i)}^{X2}\cdot(\theta/\theta_{chl}^0)$ | $f_{P(i)}^{X2}\cdot(\theta/\theta_{chl}^0)\cdot$(leakage/*dpp*) | 0.018 | 0.018 | 0.030 | 0.00119 | 50 |
| Is $X_C^{(2)}$ production dependent on nutrient availability? (EXP-X$^{(2)}$) | *Nutrients* | 0.062 | 0.062$\cdot$(leakage/*dpp*) | 0.018 | 0.018 | 0.030 | 0.00119 | 50 |
| Is $X_C^{(2)}$ production dependent on light intensity? (EXP-X$^{(2)}$) | *Constant* | 0.018$\cdot$(*dpp*/leakage) | 0.018 | 0.018 | 0.018 | 0.030 | 0.00119 | 50 |
| Does $X_C^{(3)}$ sustain higher-than-average [CDOM]? (EXP-X$^{(3)}$) | *Bleaching* | 4$\cdot f_{P(i)}^{X2}\cdot(\theta/\theta_{chl}^0)$ | 4$\cdot f_{P(i)}^{X2}\cdot(\theta/\theta_{chl}^0)\cdot$(leakage/*dpp*) | 0.072 | 0.072 | 0.120 | 0.167 | 60 |




## 3 Results

### 3.1 Skill of the optimized simulation

In this section we evaluate how the optimized model configuration reproduces seasonal cycles and vertical gradients, in terms

of biological and optical variables. We compared the output of the *Optimized* simulation with in situ observed data of mixing and nutrients (Sect. 3.1.1), BGC variables (Sect. 3.1.2) and IOPs (Sect. 3.1.3) and with satellite-retrieved data not used in the optimization (Sect. 3.1.4). The optimized parameter values and the quantitative model-data skill metrics (R and RMSE) achieved throughout optimisation can be found in **Appendix A**.

### 3.1.1 Mixing and nutrients

The temporal evolution of a climatological year (2011-2014) of the temperature in the water column is presented for the observations (**Fig. 3a**) and for the *Optimized* model run (**Fig. 3b**). Overall, there is good agreement, in terms of both the timing of the stratification and the mixing events. The seasonal cycle of nitrate (**Fig. 3c** vs. **3d**) and phosphate (**Fig. 3e** vs. **3f**) are well captured as well as the depth of the nitracline that ranges from 45 m at the onset of the stratification to 60 m at the end of the year, with $NO_3$ and $PO_4$ underestimated by about 25% in the deeper layer.


**Figure 3: Vertical profiles as a function of time of (a-b) temperature, (c-d) nitrate and (e-f) phosphate, obtained from observations at the DYFAMED site (left panels) and from the *Optimized* simulation (right panels). The white line in the temperature panels represents the mixed layer depth (mld) obtained in (a) with a threshold of 0.2 °C on temperature** (de Boyer Montégut et al., 2004)
**and in (b) with a threshold of $1 \cdot 10^{-5}$ m$^2$ s$^{-2}$ on turbulent kinetic energy. The line on (c) and (d) represents the nitracline (nitrate concentration equals 2 µM).**





### 3.1.2 Chl-a, DOC and CDOM

Observations show maximum Chl-a concentrations of more than 1 mg m$^{-3}$ in the first 30m of the water column in March-April

and the formation of a DCM in April-May, that deepens down to around 50 m and gradually shoals after October (**Fig. 4a**). The model succeeds in capturing these important spatial features of the mean site total Chl-a (TChl-a) field such as the formation of a DCM in April-May that reaches 50m depth in September-October (**Fig.4e**). However, the simulated maximum Chl-a concentrations are about 45% smaller than the observations (**Fig. 4e** vs. **4a**), and therefore the optimized simulation generally underestimates the vertically integrated Chl-a. The subdivision of Chl-a into size fractions follows the main features

observed at BOUSSOLE, which include a larger contribution of phytoplankton larger than 2 µm to the superficial bloom in March-April (**Fig. 4g** and **4h**) and the noticeable contribution of nanophytoplankton to the TChl-a at the site (Antoine et al., 2006) (**Fig. 4c**).

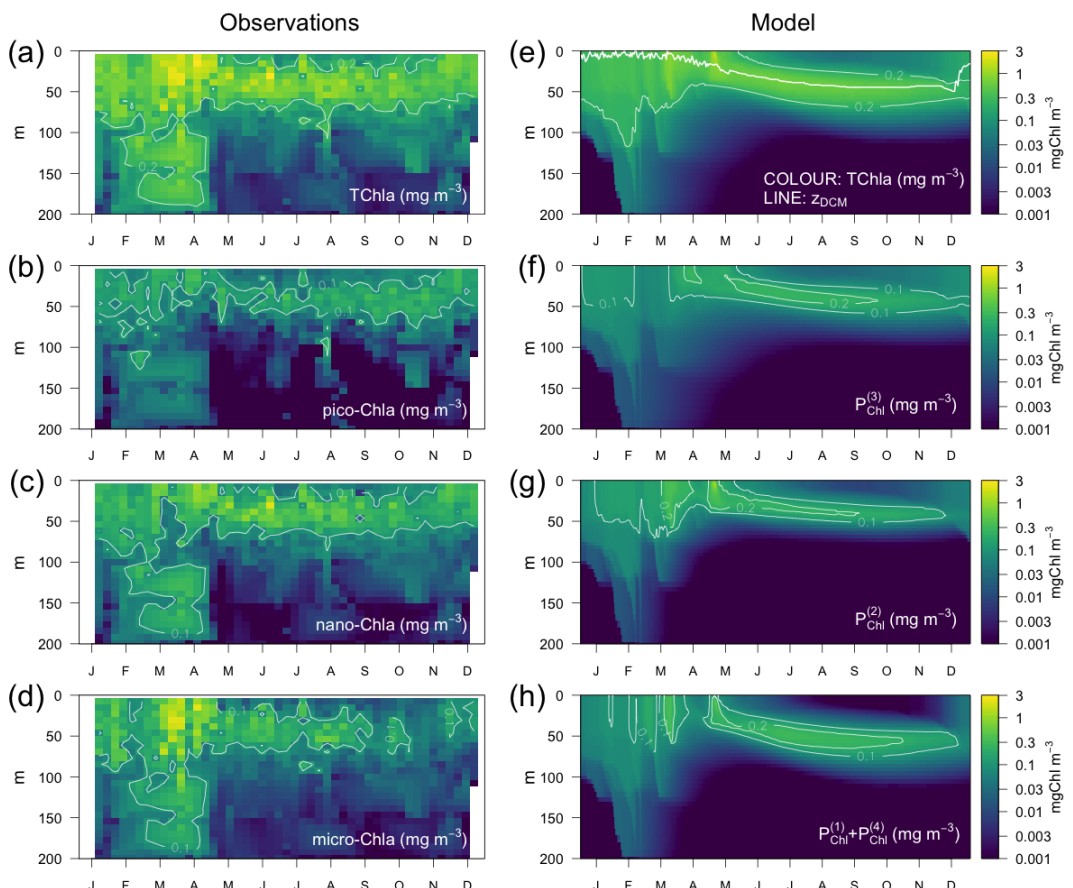

**Figure 4: Vertical profiles as a function of time of log-transformed (a) and (e) total, (b) and (f) picophytoplankton, (c) and (g)**
**nanophytoplankton, and (d) and (h) microphytoplankton Chl-a, obtained from observations at the BOUSSOLE site (left panels) and from the *Optimized* simulation (right panels). The thick line on (b) represents the depth of the DCM.**

Simulated DOC and CDOM showed clear and very different seasonal dynamics (**Fig. 5**). After a period of winter mixing,

when the concentrations of both variables were homogeneously distributed in the column, with DOC values between 38-40 µM (**Fig. 5a**) and CDOM concentrations at about 1.2 mg C m$^{-3}$ (**Fig. 5b**), superficial concentrations of both variables started to increase in March. While CDOM concentrations reached a maximum in May and decreased rapidly in surface afterwards,





DOC concentration had a maximum in June and gradually decreased until next winter. CDOM formed a sub-superficial maximum in May-June that deepened down to 50 m in October-November, this feature was absent in DOC.

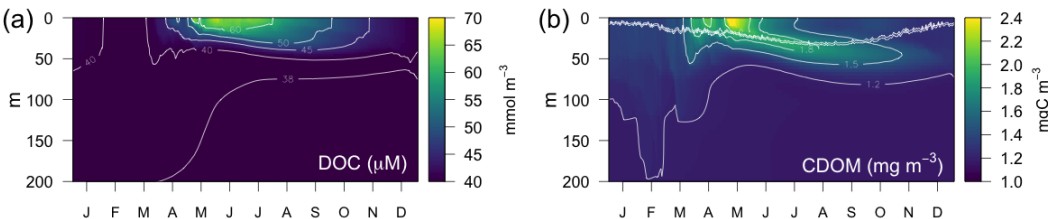


**Figure 5: Vertical profiles as a function of time of (a) total dissolved organic carbon (DOC) and (b) total CDOM in the *Optimized* simulation. The double line on (b) shows the position of the 50 and 60 µmol quanta m$^{-2}$ s$^{-1}$ of PAR that represent the PAR threshold for maximum bleaching of the biodegradable ($b_{X(1)}$ and $b_{X(2)}$) and the semi-refractory CDOM ($b_{X(3)}$), respectively.**

**3.1.3 Inherent optical properties (IOPs)**

The optimized model configuration captured important features of the mean site $a_{PH}(450)$ field, which mainly follow the distribution of TChl-a (**Fig.6a**). After general vertical homogeneity established in winter, the maximum values (0.05 m$^{-1}$) are found at surface in spring, and in summer the maximum locates around 40-50m and the lowest values (0.005 m$^{-1}$) are found at surface and below 75 m (**Fig.6b**). Observed $a_{NAP}(450)$ show very low values (0.01 m$^{-1}$) (**Fig. 6c**) comparable to those simulated

(**Fig. 6d**). Observed $a_{CDOM}(450)$ showed clear seasonal dynamics with rather vertically homogeneous values (0.018-0.02 m$^{-1}$) until April and a subsurface maximum from April to September at 30-50 m depth. After September, $a_{CDOM}(450)$ tended to be more homogeneous from surface to 130 m while still relatively high compared to winter (**Fig. 6e**). In the *Optimized* simulation, $a_{CDOM}(450)$ follows the distribution found for the observations and was therefore homogeneously distributed during winter (0.02 m$^{-1}$) and increased to values up to 0.04 m$^{-1}$ at surface in April. Thereafter, the maximum values were confined to the

depth of the yellow substance maximum (YSM) formed at 30 m depth in June and deepened throughout summer down to 50 m in December (**Fig. 6f**). Our depth-profile data showed that the simulated depth of the YSM (**Fig. 6f**) was only slightly shallower than that of the DCM (**Fig. 6e**), which is consistent with reports from Oubelkheir et al. (2007) and Xing et al. (2014) in the NW Mediterranean Sea.

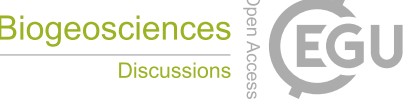



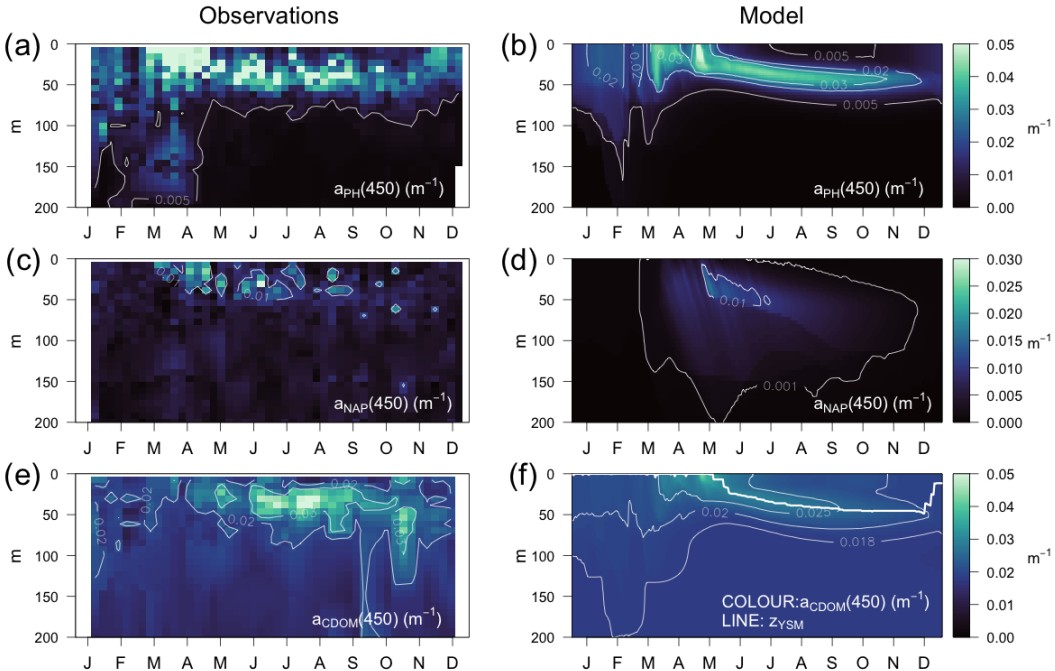

**Figure 6: Vertical profiles as a function of time of (a-b) $a_{PH}(450)$, (c-d) $a_{NAP}(450)$ and (e-f) $a_{CDOM}(450)$, obtained from observations at the BOUSSOLE site (left panels) and from the *Optimized* simulation (right panels). The thick line on (f) represents the depth of the yellow substance maximum ($z_{YSM}$) (depth of maximum $a_{CDOM}(400)$ above the 1028.88 isopycnal).**





### 3.1.4 Validation with satellite products

Satellite-retrieved Chl-a, $a_{PH}(443)$ and $a_{DG}(443)$ from January 2011 to December 2014 at the BOUSSOLE site are compared with simulated values, averaged over the first 9 m of the water column (**Fig. 7**). The simulated Chl-a values are slightly underestimated with respect to the satellite value (bias=-0.028 mg m$^{-3}$), especially in the period from June to October. If the mixing periods in February and March are excluded from the comparison, Pearson's correlation coefficient is 0.55. The simulated $a_{PH}(450)$, on the other hand, has a correlation coefficient with $a_{PH}(443)$ of 0.69 with no bias, suggesting that the

optimized values of $\bar{a}_{PH}^*$ compensate for the underestimation of Chl-a. Simulated $a_{DG}(450)$ has a correlation coefficient with $a_{DG}(443)$ of 0.40 with some degree of underestimation (bias=-0.0031 m$^{-1}$). The optimized simulation could not capture increments in $a_{DG}(443)$ during November-December, probably because these maxima were not recorded in the BOUSSOLE in situ measurements at monthly resolution.

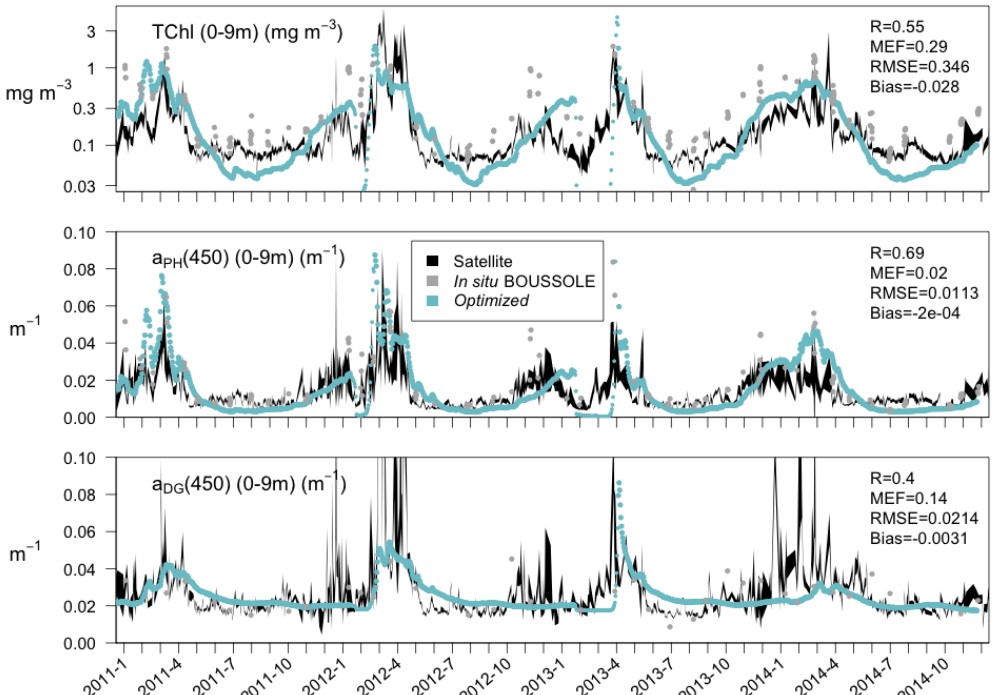

**Figure 7: Time series (January 2011 to December 2014) at the BOUSSOLE site of TChl-a (mg m$^{-3}$), $a_{PH}(450)$ and $a_{DG}(450)$ averaged over the first 0-9 m. The *Optimized* simulation (in blue) is compared to the same products retrieved from satellite at 443 nm (in black) reporting the correlation coefficient (R), the modelling efficiency (MEF), the root mean squared error (RMSE) and the average error (Bias), all calculated following** Stow et al. (2009)**. Comparable variables measured at the BOUSSOLE site are shown for reference (grey dots).**






### 3.2 Dynamics of CDOM in relation to other BGC products

In this section, we analyse the results of the *Optimized* simulation to investigate: i) the vertical and seasonal dynamics of $a_{CDOM}(\lambda)$ in relation to phytoplankton, bacteria and DOC (Sect. 3.2.1), ii) explore the mutual proportions of CDOM, phytoplankton and detritus in the absorption budget (Sect. 3.2.2) and iii) verify the ability to reproduce observed bio-optical relationships between $a_{CDOM}(\lambda)$ and other BGC variables (Sect. 3.2.3).

### 3.2.1 Seasonal/vertical dynamics of a$_{CDOM}$(450)

The average four-year depth profiles of TChl-a, $a_{CDOM}(450)$, bacterial carbon ($B_C$) and dissolved organic carbon (DOC) simulated for the BOUSSOLE site (depth <200 m) and their seasonal patterns are shown in **Figure 8** and **Figure 9**, respectively. During winter (January to March), when deep convective water mixing occurred, TChl-a was high (0.2-0.3 mg m³) and homogeneously distributed within the 0–75 m layer while it rapidly decreased to values close to 0.02 mg m⁻³ below 80 m. DOC and $a_{CDOM}(450)$ show vertical homogeneity in winter with values around 40 µM and 0.02 m⁻¹, respectively (**Fig. 8** Winter). A sub-superficial maximum of TChl-a formed in spring between 30-40 m, a $a_{CDOM}(450)$ maximum formed at 30 m depth, whereas the maximum values of DOC and $B_C$ remained within the first 10 m. DOC reached there the highest seasonal concentration with 66 µM (**Fig. 8** Spring). The sub-superficial maximum of TChl-a deepened in summer to 45 m. A reduction of surface $a_{CDOM}(450)$ values was observed from spring to summer, and the subsurface maximum was reinforced and deepened to 40 m. A $B_C$ subsurface maximum was also formed slightly above the $a_{CDOM}(450)$ and at least 10 m above the DCM (**Fig. 8** Summer). Chl-a-depleted surface waters and a weaker DCM persisted in fall. A maximum of $a_{CDOM}(450)$ was also observed in fall but in slightly deeper waters, closer to the DCM (**Fig. 8** Fall).

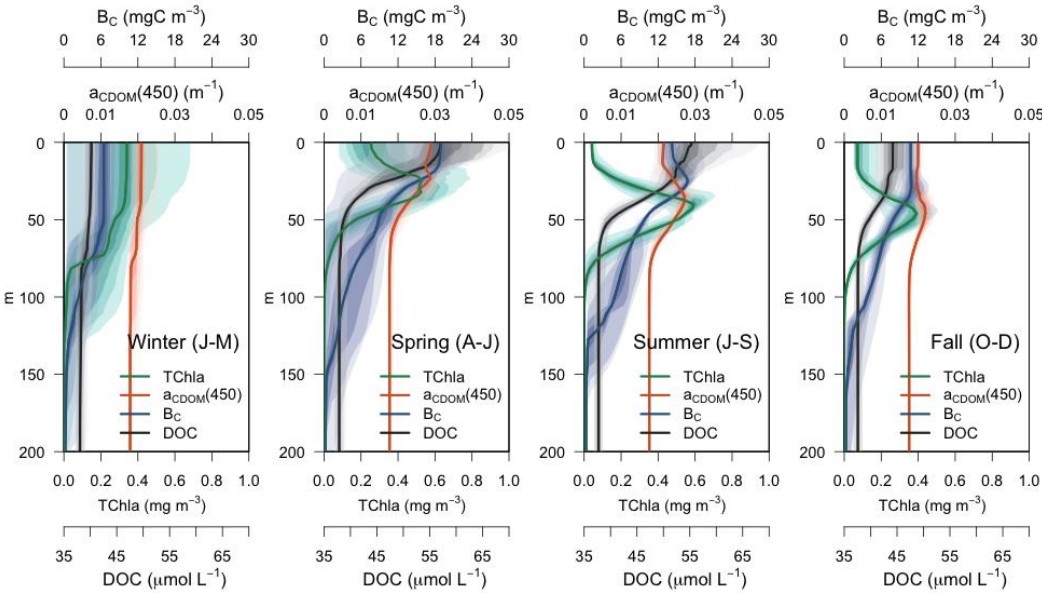

**Figure 8: Seasonal vertical profiles of phytoplankton TChl-a, $a_{CDOM}$(450), bacterial biomass and DOC. Solid lines show the median value for the season of four years (2011 to 2014) and the coloured areas show the percentiles in 10% steps from darker to lighter.**





In terms of temporal dynamics, TChl-a reached maximum concentrations (0.7 mg m$^{-3}$) at the end of the winter and gradually
       decreased to values of 0.05 mg m$^{-3}$ in summer, because of depletion of nutrients due to the strong stratification of the water
       column (**Fig. 9a**). At the same time, a DCM formed at approximately 40m depth and its TChl-a content increased throughout
       summer (**Fig. 9b**). The increase of $a_{CDOM}(450)$ started at the same time of the TChl-a increase but persisted much longer, when
       TChl-a was much lower and $B_C$ remained high (**Fig. 9a**). In the DCM, the maximum $a_{CDOM}(450)$ values were at a similar level
of maxima from March to June (**Fig. 9b**). While in the DCM, $a_{CDOM}(450)$ and DOC follow the same temporal pattern (**Fig.
       9b**), at surface the maximum of $a_{CDOM}(450)$ is reached two months earlier than DOC, that continues to accumulate during most
       of the spring, whereas $a_{CDOM}(450)$ decreases as a result of photobleaching (**Fig. 9a**). The similar temporal patterns of
       $a_{CDOM}(450)$ and $B_C$ at surface (**Fig. 9a**) and especially at the DCM (**Fig. 9b**) indicates a possible involvement of bacteria in
       CDOM production, although in our optimized simulation semi-refractory CDOM is mainly of allochthonous origin (see
Section 3.3.2 for a more detailed analysis).

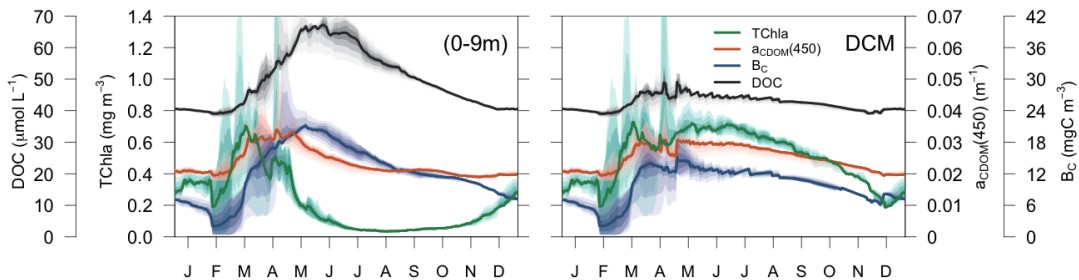

**Figure 9: Simulated annual evolution of phytoplankton TChl-a, $a_{CDOM}$(450), bacterial biomass ($B_C$) and DOC at a) surface (0-9 m)
and at b) the depth of DCM. The solid lines show the average year from 2011 to 2014 and the coloured areas show the interannual
variability.**

### 3.2.2 Light absorption budget

       The relative contributions of CDOM, phytoplankton and detritus to the total non-water light absorption at 450 nm are shown
for the *Optimized* simulation at the BOUSSOLE site in **Figure 10**. During the four-year simulation, the contribution of CDOM
       to the light absorption budget in the entire water column down to 100 m varied between 12 and 99 % with an average value of
       64 %. At surface (**Fig. 10a**), detritus was the minor contributor to the absorption budget (1 to 10%). Phytoplankton was the
       major absorbing substance only during the bloom in March and April, whereas $a_{CDOM}(450)$ dominated the rest of the year and
       contributed more than 50% to the light absorption. This dominance even held in summer and fall when $a_{CDOM}(450)$ was
minimal as a result of photobleaching (**Fig. 6f**). In accordance with the results reported by Organelli et al. (2014), the surface
       waters of BOUSSOLE can therefore be classified as "CDOM-type" for most of the year, except for the algal bloom in March
       and April. In the DCM (**Fig. 10b**), on the other hand, with the exception of a few short mixing periods, $a_{PH}(450)$ always
       contributed the most to absorption, but never exceeded 60%. Detritus were still a minor component (1 to 16%) and $a_{CDOM}(450)$
       varied between a minimum of 30% in May and a maximum of 60-70% in winter. Note that the depth of the YSM was
consistently about 5 m shallower than the DCM (**Fig. 8** in Spring, Summer and Fall). Even though the maximum values are
       not exactly at the DCM, $a_{CDOM}(450)$ is an important contributor to total absorption in this layer as the configuration of the
       optimized model is able to reproduce the formation of a CDOM subsurface maximum in spring that progressively intensifies
       throughout the summer, closely connected to the DCM.



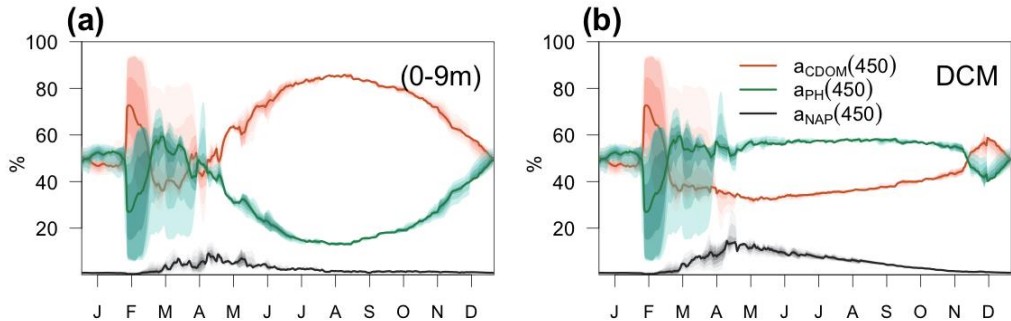


**Figure 10: Annual cycle of the relative contributions of CDOM, phytoplankton and detritus to total non-water light absorption at 450 nm (a) in the first 0-9 m and (b) at the DCM of the BOUSSOLE site. The solid lines show the annual average from 2011 to 2014 and the coloured areas show the inter-annual variability.**

### 3.2.3 Bio-optical relationships with $a_{CDOM}(\lambda)$

The overall relationship between TChl-a and $a_{CDOM}(\lambda)$, shown in **Fig. 11a**, reflects the covariation of these variables over vertical and seasonal scales. In fact, the *Optimized* simulation showed minimum $a_{CDOM}(450)$ values of 0.018 m$^{-1}$ (see also **Fig. 6f**) with very low TChl concentrations, in agreement with data observed at the BOUSSOLE site. $a_{CDOM}(450)$ also tended to increase with TChl-a concentrations, but the variability was large, with $a_{CDOM}(450)$ ranging from 0.025 to 0.07 m$^{-1}$ for TChl-

a values equal to 1 mg m$^{-3}$ (**Fig. 11a**). Furthermore, in agreement with Organelli et al. (2014), data shown in **Fig. 11a** were generally above the relationships proposed by Morel and Gentili (2009b) and Bricaud et al. (2010), derived from a global bio-optical model in the first case, and from in situ measurements collected in the Southeast Pacific Ocean in the second one. The main reason is the higher-than-average contribution of CDOM for a given Chl-a content in the Mediterranean Sea, in contrast to other oceans. Both TChl-a and $a_{CDOM}(450)$ were used as observations in the optimization, therefore the emergence of such

a relationship was expected.

The optimized model configuration also allowed the emergence of bio-optical relationships with unobserved variables, such as DOC. The simulated $a_{CDOM}(250)$ as a function of DOC concentration shows a fairly linear relationship (**Fig. 11b**). The same relationships found by Catalá et al. (2018) and Galletti et al. (2019) (grey areas and lines in **Fig. 11b**) were based on in situ measurements collected over the entire Mediterranean Sea in April-May of different years. Our modelled data for spring show

a good agreement with their samples taken in the upper part of the water column (with $a_{CDOM}(250)$ values larger than 0.9 m$^{-1}$) while our modelled data for fall-winter agree better with samples taken below 200 m ($a_{CDOM}(250)$ values smaller than 1 m$^{-1}$), a period when the water column was more homogeneous and the values of both DOC (**Fig. 5a**) and $a_{CDOM}(\lambda)$ (**Fig. 6f**) in the superficial part were more similar to those in the deep part of the water column. Note that Catalá et al. (2018) and Galletti et al. (2019) reported $a_{CDOM}(\lambda)$ values at 254 nm, while modelled data report $a_{CDOM}(\lambda)$ values in the 250 nm waveband that ranges

from 187.5 to 312.5 nm.



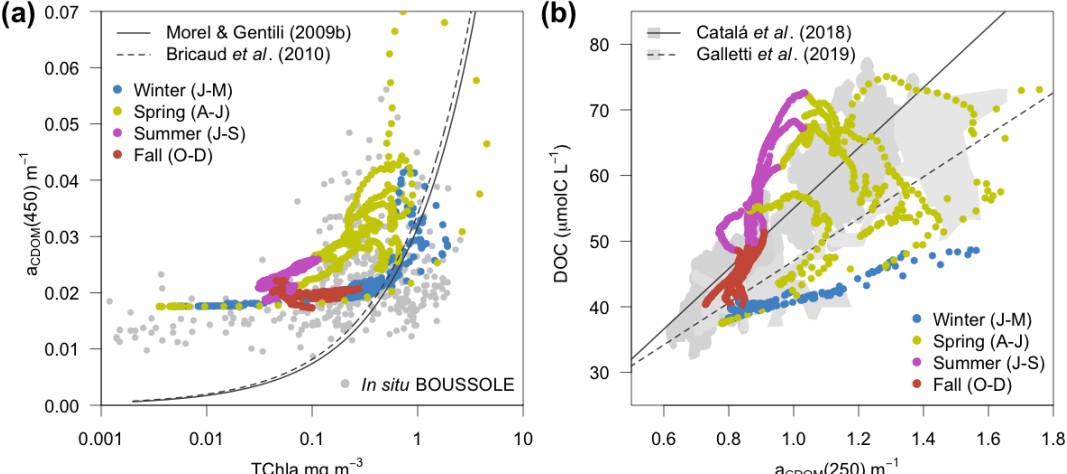

**Figure 11: Variations of $a_{CDOM}(\lambda)$ (a) at 450 nm ($a_{CDOM}(450)$) as a function of TChl-a and (b) at 250 nm ($a_{CDOM}(250)$) as a function of total DOC, all simulated at the surface (0-9m) of the BOUSSOLE site from 2011 to 2014. The in situ TChl-a and $a_{CDOM}(450)$ values collected in BOUSSOLE from 2011 to 2016 are displayed in the background of panel (a). The regression lines from** Catalá et al. (2018) **and** Galletti et al. (2019) **and the ranges of their reported values in the Mediterranean Sea are shown in the background of panel (b).**

### 3.3 Sources of CDOM

In this final section of results, using a series of hypothesis-testing experiments (**Table 2**), we investigated the mechanisms that determine the vertical and temporal patterns of CDOM distribution at the BOUSSOLE site. We step through two aspects: the biogenic in situ production of biodegradable CDOM (EXP-X$^{(2)}$, Sect. 3.3.1) and the role of semi-refractory CDOM in maintaining CDOM concentrations in the surface ocean (EXP-X$^{(3)}$, Sect. 3.3.2).

#### 3.3.1 In situ production of CDOM and synchronization with phytoplankton Chl-a

EXP-X$^{(2)}$ focuses on the physiological processes affecting CDOM production by phytoplankton. In the *Optimized* simulation, the parameters for the production of $X_C^{(2)}$ by phytoplankton ranged from 3.2 to 8.9 % ($f_{P(1)}^{X2}$=8 ± 0.9, $f_{P(2)}^{X2}$=4.3 ± 1.6, $f_{P(3)}^{X2}$=3.9 ± 0.7), except for P$^{(4)}$, whose value did not converge (**Appendix A**). These parameters, regulated by $\theta/\theta_{chl}^0$, represent which fraction of the leakage is coloured, whereas DOC from exudation is assumed to be non-coloured. Therefore, the fraction $X_C^{(2)}$: $R_C^{(2)}$ in the *Optimized* simulation was variable in time and depth for two reasons: 1) depends on the relative proportions of DOC from leakage or exudation and 2) depends on the $\theta$ to $\theta_{chl}^0$ ratio of phytoplankton. Two additional simulations considered an average value of 6.2% of CDOM in the leakage flux (*Nutrients*, **Table 2**) and an average value of 1.8% of CDOM in the total *dpp* flux (*Constant*, **Table 2**). **Figure 12a** shows the percentage that CDOM flux represents of total DOC flux ($f_{R2}^{X2}$) at surface (0-9m) and at the depth of the DCM in the three simulations of EXP-X$^{(2)}$.

In the *Nutrients* simulation, when the sub-surface Chl-a maximum develops in March-April, the proportions of CDOM production differ between the surface (1%) and the DCM (3%) as nutrient availability begins to decrease at the surface compared to the DCM. This difference remained fairly stable until the next mixing, except for an increase to 4 % in the DCM in the end of November. In the *Optimized* simulation, the fraction of CDOM at surface and at the DCM also differs from March onwards, but the percentages are lower in Spring and Summer (0.5 and 2 % respectively) and gradually increase in fall and winter as light intensity decreases and phytoplankton photoacclimate accordingly. In terms of $X_C^{(2)}$ production flux, this means



that whereas the *Constant* simulation is the one producing more $X_C^{(2)}$ at surface and less $X_C^{(2)}$ at the DCM (**Figure 12b** vs. **12c**),

the *Nutrients* simulation increases substantially CDOM production in the DCM compared to the other two (**Figure 12c**), and

the *Optimized* simulation decreases substantially CDOM production at the surface (**Figure 12b**). With respect to the observable

variable $a_{CDOM}(450)$, the *Optimized* formulation is able to reproduce the dynamics of $a_{CDOM}(450)$ at the surface during the

transition from the winter productive phase to the summer better than the other two (**Figure 12d**), while the *Nutrients*

simulation captures the dynamics of the DCM in summer quite similarly to the *Optimized* one, suggesting that nutrient

regulation predominantly determines the dynamics in the DCM (**Figure 12e**).

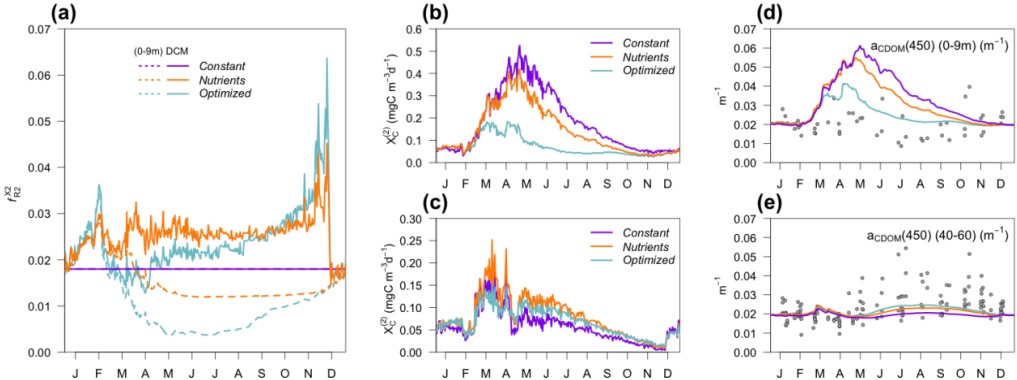

**Figure 12: EXP-X$^{(2)}$:** In situ production of CDOM by phytoplankton in the *Optimized* simulation (blue), in the simulation *Nutrients* where the fraction $f_{R2}^{X2}$ was a constant 6.2% of leakage (orange) and in the simulation *Constant* where the fraction $f_{R2}^{X2}$ was a
constant 1.8% of the total exudation (purple). (a) Annual cycle of $f_{R2}^{X2}$ at the surface (dashed lines) and at the depth of the DCM (solid lines) in the three simulations. Annual cycle of $X_C^{(2)}$ production flux (b) averaged over the first 0-9 m and (c) in the DCM. Annual cycle of $a_{CDOM}(450)$ averaged in (d) the first 0-9 m and (e) between 40-60 m; the dots show $a_{CDOM}(450)$ values measured at the BOUSSOLE site.

**3.3.2 The role of allochthonous CDOM and/or mediation by bacteria**

The results of EXP-X$^{(3)}$ are summarized in **Figure 13**. In the *Optimized* simulation, the deep inventory of $X_C^{(3)}$ replenishes the

surface during mixing and is slowly photobleached ($b_{X(3)}$=0.00119 d$^{-1}$ and $I_{X(3)}$=50 µE m$^{-2}$ d$^{-1}$, **Appendix A**). The optimized

value of $b_{X(3)}$ was much lower than the prescribed value for the labile and semi-labile pools (0.167 d$^{-1}$). This simulated a

moderate increase in $X_C^{(3)}$ bleaching rates towards the surface, compared to the simulation *Bleaching* where the bleaching

parameters for $X_C^{(3)}$ were set equal to the other two pools (**Fig. 13a**, blue line *vs.* magenta line). The lower $X_C^{(3)}$ degradation in

the *Optimized* simulation contributed to the minimum values of $a_{CDOM}(450)$ at the surface being above 0.018-0.02 m$^{-1}$

throughout the year, with $X_C^{(3)}$ contributing more than 80% of the total pool in fall-winter, and around 50% in spring (**Fig 13b**,

blue line). In *Bleaching*, on the other hand, where $X_C^{(3)}$ was bleached more intensively in the illuminated part of the water

column and biota-produced CDOM was quadrupled, the minimum values of Fall and Winter were equally maintained by $X_C^{(2)}$

and $X_C^{(3)}$, but during the productive season $X_C^{(1)}$ and $X_C^{(2)}$ were largely overestimated (**Fig 13b**, magenta line).

The *Bleaching* simulation reproduced consistent average values of $a_{CDOM}(450)$ though with stronger seasonal differences than

the *Optimized* simulation at depths close to the DCM (**Fig 13c**). Below the DCM, the biotically produced CDOM cannot

maintain the observed values of $a_{CDOM}(450)$ when the contribution of allochthonous $X_C^{(3)}$ decreases in *Bleaching* (**Fig 13d**).

These results indicate that in the *Optimized* simulation, vertical transport of CDOM from deep waters was the main process

that increased the concentration in the upper layer and kept $a_{CDOM}(450)$ larger than 0.02 m$^{-1}$ throughout the year. We cannot





rule out the possibility that this background $a_{CDOM}(450)$ is sustained in reality by bacterial production. Optimization decreased $b_{X(3)}$ instead of increasing $f_B^{X3}$ (2.3 ± 1.3 %), probably because increasing $f_B^{X3}$ cannot maintain the high $a_{CDOM}(450)$ values observed below the DCM, since bacteria are hardly present in this layer in our simulations (**Fig. 8**). In any case, in the *Optimized* simulation, the above-average $a_{CDOM}(450)$ values are maintained by $X_C^{(3)}$, either by allochthonous $X_C^{(3)}$ with a lower bleaching

rate than that produced by phytoplankton or $X_C^{(3)}$ produced by bacteria. Below 150 m, the only contributor to the total pool of CDOM is $X_C^{(3)}$ in both simulations. With an initialization value for $X_C^{(3)}$ of 1.25 mg m$^{-3}$ and a remineralization rate of 3e-5 d$^{-1}$ (Legendre et al., 2015), the values of $a_{CDOM}(450)$ were consistent with those observed in BOUSSOLE (**Fig. 13e**) and reported in Organelli et al. (2014).

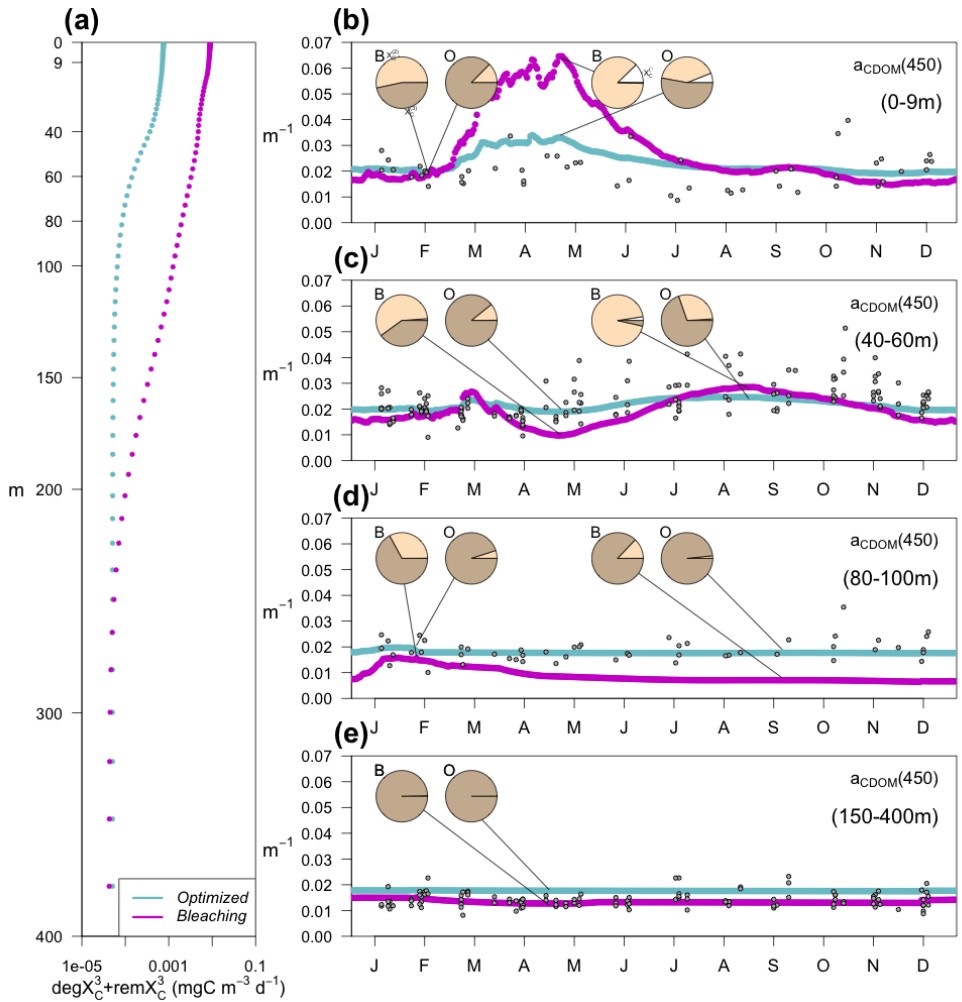


**Figure 13: EXP-X$^{(3)}$: Role of $X_C^{(3)}$ in the *Optimized* simulation (blue) and in the simulation *Bleaching* where the bleaching rate of $X_C^{(3)}$ was set to 0.167 d$^{-1}$ and $f_{Z(5)}^{X1}$, $f_{Z(6)}^{X1}$, $f_B^{X3}$ and $f_{P(i)}^{X2}$ were multiplied by 4 with respect to the optimized values (pink). (a) Degradation rate of $X_C^{(3)}$ (deg$_{X3}$ + rem$_{X3}$) as a function of depth and $a_{CDOM}(450)$ annual cycle averaged in depth layers between (b) 0-9 m, (c) 40-60 m, (d) 80-100 m and (e) 150-400 m. The dots in (b-e) show $a_{CDOM}(450)$ observations at the BOUSSOLE site. The inset pie charts**
**show the composition of the CDOM inventory in terms of reactivities ($X_C^{(3)}$ in brown, $X_C^{(2)}$ in orange and $X_C^{(1)}$ in white) at selected time points in the two simulations, *Bleaching* (B) and *Optimized* (O).**





## 4 Discussion

### 4.1 Optimization of a hydrodynamic-BGC model through a 1D fast system

Some of the uncertainties in coupled hydrodynamic-ecosystem models can be attributed to incomplete knowledge and
representation of fundamental processes being simulated (Frölicher et al., 2016) and poorly constrained parameter values
(Murphy et al., 2004), especially for the BGC processes. Search for optimal parameter values by manual trial-and-error
calibration or through brute-force approaches can be computationally unaffordable. Furthermore, testing parameters on an
individual basis can lead to important information being missed, given parameter interactions and non-linearity of complex
ecosystem models. Here we successfully applied an optimization procedure that integrates model output and observations to
constrain a relatively large number of parameters in a one-dimensional configuration of the BFM. The constrained parameters
contributed to improve the simulation of BGC (Chl-a) and optical variables (IOPs) at the data-rich BOUSSOLE site. The use
of the many observations available at this site led us to improve the simulation of bio-optical relationships that were either
observed ($a_{CDOM}(450)$ as a function of TChl-a) or not observed directly at the site (DOC as a function of $a_{CDOM}(250)$) but
reported by other studies in the Mediterranean Sea (Catalá et al., 2018; Galletti et al., 2019).

Our rigorous optimization had some limits, in particular it could not constrain all the parameters we had investigated. This was
the case with the parameters for dinoflagellates ($P^{(4)}$), which could not be constrained because this group represents only a
small part of the microphytoplankton size fraction compared to diatoms ($P^{(1)}$). This result indicates that the subset of
observations we used may not contain enough information, which is a well-known limitation of formal optimization (Ward et
al., 2010). It also suggests that the results of the optimization may vary if we use disaggregated observations for diatoms and
dinoflagellates. The choice of the parameter list is also crucial. A common strategy is to optimize the parameters to which
model output is more sensitive. Since we were interested in the processes that shape CDOM cycling we chose to include in
the optimization as many parameters implied in CDOM cycling as possible. Nevertheless, our results show that the
optimization trajectories were to some extent driven by the parameters to which the model output was more sensitive. This
was the case for the maximum Chl:C quota ($\theta_{chl}^0$) of $P^{(2)}$ and $P^{(3)}$ (**Fig. A2,** Appendix A), which drove the improvement in
nano- and pico-phytoplankton Chl-a (**Fig. A1,** Appendix A) rapidly and intensively. All parameters we investigated are
involved in different processes. But, interdependence and unidentified correlations between parameters can cause issues of
poor identifiability, i.e. different combinations of parameters might yield the same result. Moreover, the optimized parameter
values could compensate for processes that are either missing or poorly formulated in the model (e.g, the origin of $X_C^{(3)}$ in
surface waters, discussed below). These two aspects imply limitations of the values obtained in this work.

The data sets used in this work include fewer properties than the BGC model and observations themselves are subject to errors.
As a result, an excellent agreement between model and observation does not guarantee that the model's representation of
unobserved properties is correct (Fennel et al., 2022). In our results, the optimized model output matched relatively well the
observations provided (**Fig. 4** and **5**) and the same variables remotely retrieved (**Fig. 7**), while still being more consistent with
established knowledge of processes at the BOUSSOLE site (**Fig. 6** and **Fig. 11**) not considered in the optimization. As the
results of the optimization may vary depending on how many observation types are used (Wang et al., 2020), an increasing
breadth of observables will allow us to continually evaluate our model and predictions. High-frequency data collected by the
BOUSSOLE project (every 15 min during daylight, and at hourly intervals at night), such as Chl-a fluorescence, bb(λ) of total
particles or remote sensing reflectance above sea water, are valuable additional sources of information for further analysis. We
have not tested the influence of the sampling frequency on the optimization performance. Using monthly mean profiles has
been reported sufficient to contribute to the optimization of vertical Chl-a structure and optimize biological parameters (Shu
et al., 2022; Bisson et al., 2021). But assimilating high-frequency data in a 1D model has been proven beneficial for inferring
phytoplankton dynamics when compared to assimilating observations that are more limited in time (Kaufman et al., 2018).



CDOM cycling models are useful only if they are flexible enough (i.e. portable) to provide accurate predictions for a wide range of conditions occurring in marine ecosystems. In that case, they can be integrated into larger 3D models of BGC cycles.

The possibility of applying parameter optimization techniques in computationally efficient 1D models before using the resulting parameters in the 3D version seems advantageous. Furthermore, parameters optimized with surface data only (i.e. satellite retrieved) may present inconsistencies when simulating underwater biological fields (Wang et al., 2020). This is particularly problematic in oligotrophic regions where the deep Chl-a maximum (DCM) is relatively deep and poorly observed by the satellites (Cullen, 2015; Fennel and Boss, 2003). Optimizing parameter values at intensively sampled 1D sites can

overcome this problem and improve the simulation of both vertical and seasonal variability. Our application provided a new model formulation and sensitivity results that are critical for future implementation of the CDOM cycling in the 3D configuration of the BFM. Further analysis will be required to determine whether the optimal parameter sets from the 1D version should be applied directly in the 3D model. This decision should balance the risk of a lower agreement between model and data in 3D than in the 1D counterpart, with the opportunity to preserve of key features presented in this analysis. These

features include evident seasonal dynamics and vertical gradients, and the higher-than-average CDOM light absorption coefficients (Sections 3.1 and 3.2).

**4.2 Hypothesis testing in a 1D virtual laboratory: CDOM cycling numerical experiments**

Our results provided new insights on the in situ production of semi-labile CDOM by phytoplankton. We simulated a clear seasonal and vertical structure of $a_{CDOM}(\lambda)$. This is in better agreement with observations when the proportion of coloured

material in the *dpp* depends on nutrient and light constraints (**Fig. 12**). We have proposed a simplified parameterisations to account for the effects of light and nutrients on CDOM production. Nutrient limitation acts through the relative proportion between leakage and *dpp* and seems to be the main cause of the difference in CDOM production between surface and DCM. This could reflect the fact that when growth is limited due to low inorganic nutrient availability, the photosynthetic excess produces mainly non-coloured carbohydrates, as severe N and P limitation favours the liberation of carbohydrates that do not contain these elements (Myklestad, 1995) or simply that both higher CDOM production rates and nutrient availability occur

simultaneously in the DCM (Bushaw et al., 1996; Stedmon et al., 2007; Zhang et al., 2011a). Either way, the CDOM production dependency on nutrient availability helped to reproduce the observation that increased CDOM concentrations are found at the depth of the Chl-a maximum (Coble et al. 1998).

Light limitation acts through the elemental ratio of Chl:C relative to the maximum value and appears to be the main cause of

the seasonality of CDOM at the surface, although it contributes to amplifying the differences between surface and DCM in CDOM production. As with nutrients, this does not necessarily represent a causal mechanism, but a general pattern of high-light low-CDOM waters. Passive leakage by diffusion through the cell membrane is controlled by cell permeability, surface area to volume ratio of the cell, and the intracellular concentration of the compound in question (Bjørrisen, 1988). Thus, our results could be indeed representing a lower concentration of coloured material in the *dpp* of low pigmented cells. However,

given the inverse covariation between photosynthetic and photoprotective pigments, surface communities exposed to high light intensities that have a low θ may contain a higher proportion of photoprotective pigments. This, in principle, could contribute to the leakage of coloured molecules (e.g., carotenoids). More likely than lower CDOM production under high light, our results could be representing increased CDOM destruction and a larger or different impact of bleaching than assumed in our simulations. Photobleaching was likely the main cause of CDOM destruction in the surface layer from spring to summer

at the BOUSSOLE site (Organelli et al., 2014), as generally observed for other oceanic waters (Coble, 2007; Bracchini et al., 2010). However, it has been found that the photochemical lability of CDOM is not necessarily related to the highest solar irradiance but depends on the accumulated light exposure (Yamashita et al., 2013) and the photooxidative conditions in the water column (Swan et al., 2012). This may explain why the modulation of CDOM production by the photoacclimation of phytoplankton (through θ) helped to indirectly improve the effect of light intensity on CDOM concentrations.



Besides biological production in situ, our results show that vertical transport of CDOM from depth was the main process that increased CDOM concentration in the upper layer and kept $a_{\text{CDOM}}(450)$ above 0.018-0.02 m$^{-1}$ throughout the year. It cannot be excluded that this background value of $a_{\text{CDOM}}(450)$ is maintained by processes in the real system that are poorly represented in the model (bacterial production) or not represented at all (other sources of allochthonous CDOM such as atmospheric deposition or advection). Nevertheless, the higher-than-average $a_{\text{CDOM}}(450)$ values at the surface are supported by semi-

refractory CDOM (**Fig. 13b**), consistent with the role of atmospheric deposition in providing higher-than-average CDOM concentrations in the basin (Galletti et al., 2019). In the water column, we lack observations on the reactivity of the CDOM pool we simulated. However, our simulations mimic the results reported by Bracchini et al. (2010) that the refractory fraction of the CDOM dominates the water below the DCM (**Fig. 13d**). We cannot know whether such refractory CDOM is in fact 'newly' produced by bacteria or whether it is 'old' CDOM vertically transported from the deep reservoir.

**4.3 Conclusions and future developments**

Adding optical components to BGC models greatly improves the comparability of simulated and observed types of variables (IOCCG, 2020). In this work, we have shown how the integration of models and observations through an optimisation method provides insights into the dynamics of absorbing BGC products such as CDOM. Given its strong absorption of UV and blue light, the accurate simulation of CDOM cycling and main dynamics has major implications. The potential of using an

observable proxy, such as $a_{\text{CDOM}}(\lambda)$, for the retrieval of BGC products (e.g., DOC) results of significant interest. Furthermore, the accurate simulation of the CDOM absorption band in the blue part of the spectrum helps deciphering the mechanisms underlying the optical properties of the basin, especially in coastal and shelf waters and semi-enclosed seas such as the Mediterranean Sea. Future model developments may aim to simulate the fraction of CDOM that is able to fluoresce, i.e. fluorescent DOM, and therefore extend the points of convergence between observations and model results. This would be

advantaged by fitting DOM fluorescence sensors on-board BGC-Argo floats.

The results of the in situ production and photobleaching experiments contributed to our understanding of the main drivers of the cycling of CDOM compounds in the 1D configuration of the BFM model. These results highlight the potentiality of the 1D fast system to be used as a virtual laboratory for hypothesis testing regarding CDOM cycling, even when proposing relatively simple parameterizations to include the effect of light and nutrients in the production of chromophoric DOM. By

testing them in the 1D configuration, we have demonstrated that both help to represent the vertical and seasonal dynamics of CDOM at the site and can guide the development of further mechanistic models aimed at finding out what drives CDOM production by phytoplankton, even though observations on the relative importance of the different processes are still limited.

**Appendix A: Calibrated parameters and insights from optimization trajectories**

Twenty-five BGC and optical parameters, gathered in **Table A1**, were constrained (i.e., found optimal parameter estimates

with relatively small uncertainties) against observations of pico-, nano- and microphytoplankton Chl-a concentrations, $a_{\text{PH}}(450)$, $a_{\text{NAP}}(450)$, $a_{\text{CDOM}}(450)$ and $S_{CDOM}^{350-500}$ collected at the BOUSSOLE site from 2011 to 2014. The variation in correlation coefficients (R) between model and observation, as well as the root mean squared errors (RMSE) for all variables included in the multi-objective function (i.e., lnLikelihood) are shown in **Fig. A1**. All variables increased their R and decreased RMSE compared to the observations along the optimization. The difference in lnLikelihood relative to the initial one reached

0.01 at generation 115 and 0.005 at generation 144, so although the procedure was continued, no improvement was observed all R's stop improving (vary less than 0.02) after about 150 generations (nano-phytoplankton Chl-a was the last).

Interestingly, the improvements of the variables (RMSE reduction and R increase) and the convergences of the parameters (**Fig. A2**) were not homogeneous during the calibration process. Instead, the results show fast and late moving parameters. During the first 20 generations the main improvement was in the pico- and nano-Chl-a (**Fig. A1a,b**), that corresponded with





early constrained values of $\theta_{chl}^0$ in P[3] and P[2] (**Fig. A2h,m**). Not all independent parameters could be well constrained. Several parameters did not converge after 180 generations. First, the spectral slope of the labile CDOM ($S_{X(1)}^{350-500}$). This is explained by the fact that in our model configuration, labile CDOM is consumed as soon as it is produced, it does not accumulate and its spectral slope has in practice a very limited influence on the spectral slope of the whole CDOM pool ($S_{CDOM}^{350-500}$). Second, any of the parameters of dinoflagellates (P[4]) did converge. This is because we included an aggregate value of

microphytoplanktonic Chl-a in the multi-objective function (micro-Chl-a), and given the preponderance of diatoms (P[1]) in this size-class, varying P[1] parameters caused all changes in microphytoplankton concentration and P[4] parameters have very limited influence.


**Table A1: List of parameters considered for optimization with indication of the numerical range given at the beginning of the optimization and the mean value and range at the end (generation 180).**

| Parameter | Description | Min. | Max. | Optimized | | Unit |
|---|---|---|---|---|---|---|
| *CDOM* (X[1], X[2] and X[3]) | | | | | | |
| $b_{X(3)}$ | Bleaching rate of X[3] | 0.001 | 0.167 | 0.0019 | ± 4.3e-5 | d[-1] |
| $I_{X(3)}$ | PAR threshold for X[3] bleaching | 1 | 60 | 50 | ± 4 | µE m[-2] s[-1] |
| $S_{X(1)}^{350-500}$ | X[1] absorption spectral slope | 0.014 | 0.020 | 0.0175 | ± 15e-4 | nm[-1] |
| $S_{X(2)}^{350-500}$ | X[2] absorption spectral slope | 0.014 | 0.020 | 0.0162 | ± 6.6e-4 | nm[-1] |
| $S_{X(3)}^{350-500}$ | X[3] absorption spectral slope | 0.014 | 0.020 | 0.0180 | ± 1.3e-4 | nm[-1] |
| *Detritus* (R[6]) | | | | | | |
| $a_R^{440}$ | Detritus absorption at 440 nm | 0.0001 | 0.0010 | 0.00058 | ± 2e-5 | m[2] mg C[-1] |
| *Diatoms* (P[1]) | | | | | | |
| $\bar{a}_P^*$ | Absorption cross-section | 0.0100 | 0.0160 | 0.0152 | ± 3e-4 | m[2] mg Chl-a[-1] |
| $\phi_c^0$ | Photochemical efficiency | 0.700e-3 | 1.200e-3 | 0.750e-3 | ± 12e-6 | mg C[-1] µE[-1] |
| $\theta_{chl}^0$ | Maximum Chl:C quotum | 0.01 | 0.03 | 0.026 | ± 68e-5 | mg Chl-a mg C[-1] |
| $f_P^{X2}$ | Maximum fraction X[2] to R[2] | 0.01 | 0.10 | 0.080 | ± 9e-3 | - |
| *Flagellates* (P[2]) | | | | | | |
| $\bar{a}_P^*$ | Absorption cross-section | 0.0150 | 0.0400 | 0.0293 | ± 16e-4 | m[2] mg Chl-a[-1] |
| $\phi_c^0$ | Photochemical efficiency | 0.100e-3 | 0.350e-3 | 0.198e-3 | ± 3.9e-6 | mg C[-1] µE[-1] |
| $\theta_{chl}^0$ | Maximum Chl:C quotum | 0.01 | 0.045 | 0.045 | ± 6e-5 | mg Chl-a mg C[-1] |
| $f_P^{X2}$ | Maximum fraction X[2] to R[2] | 0.01 | 0.10 | 0.043 | ± 16e-3 | - |
| *Picophytoplankton* (P[3]) | | | | | | |
| $\bar{a}_P^*$ | Absorption cross-section | 0.0200 | 0.0450 | 0.0443 | ± 1.5e-4 | m[2] mg Chl-a[-1] |
| $\phi_c^0$ | Photochemical efficiency | 0.300e-3 | 1.000e-3 | 0.990e-3 | ± 2.3e-6 | mg C[-1] µE[-1] |
| $\theta_{chl}^0$ | Maximum Chl:C quotum | 0.01 | 0.045 | 0.014 | ± 27e-5 | mg Chl-a mg C[-1] |
| $f_P^{X2}$ | Maximum fraction X[2] to R[2] | 0.01 | 0.10 | 0.039 | ± 7e-3 | - |
| *Dinoflagellates* (P[4]) | | | | | | |
| $\bar{a}_P^*$ | Absorption cross-section | 0.0100 | 0.0250 | 0.0178 | ± 33e-4 | m[2] mg Chl-a[-1] |
| $\phi_c^0$ | Photochemical efficiency | 0.150e-3 | 0.450e-3 | 0.257e-3 | ± 64e-6 | mg C[-1] µE[-1] |
| $\theta_{chl}^0$ | Maximum Chl:C quotum | 0.01 | 0.03 | 0.019 | ± 545e-5 | mg Chl-a mg C[-1] |
| $f_P^{X2}$ | Maximum fraction X[2] to R[2] | 0.01 | 0.10 | 0.053 | ± 23e-3 | - |
| *Zooplankton* (Z[5] and Z[6]) | | | | | | |
| $f_{Z(5)}^{X1}$ | Fraction X[1] to R[1] in Z[5] excretion | 0.01 | 0.10 | 0.018 | ± 2e-3 | - |
| $f_{Z(6)}^{X1}$ | Fraction X[1] to R[1] in Z[6] excretion | 0.01 | 0.10 | 0.030 | ± 7e-3 | - |
| *Bacteria* (B) | | | | | | |
| $f_B^{X3}$ | Fraction X[3] to R[3] in B excretion | 0.01 | 0.10 | 0.018 | ± 2e-3 | - |





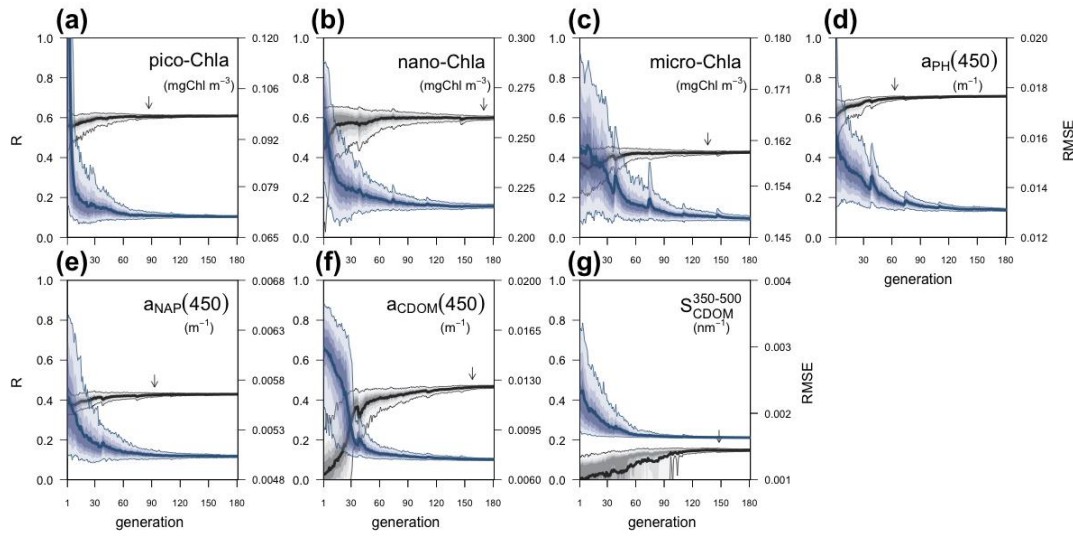


**Figure A1:** Optimization trajectories of the fit metrics: correlation coefficient (R, grey, left axis) and root mean square error (RMSE, blue, right axis). The thick lines show the median of the distribution of the metric and the coloured areas the percentiles up to 5-95%. The vertical arrows show the generation at which the range of model vs. observed R is less than 0.02.

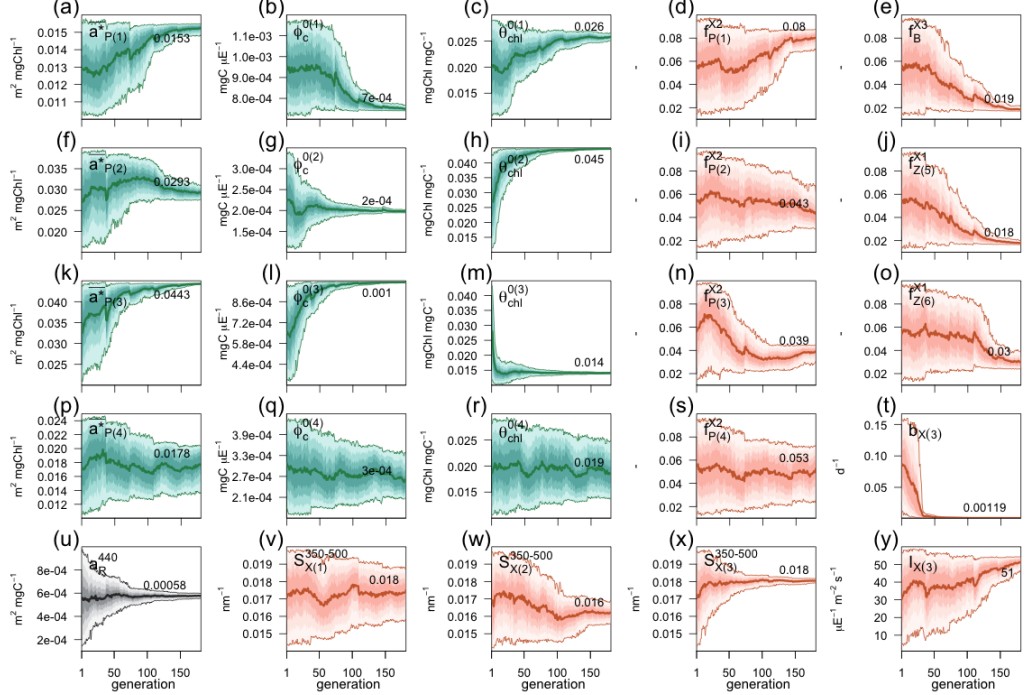

**Figure A2:** Optimization trajectories of parameter values: phytoplankton parameters are shown in shades of green, parameters related to CDOM are shown in shades of orange and the only detritus parameter is shown in grey. The thick lines show the median of the distribution of parameter values and the coloured areas the percentiles up to 5-95% (see Table A1 for the exact values and ranges at convergence).



**Code availability**

The GOTM code is available at https://www.github.com/gotm-model. FABM code is available at https://github.com/fabm-model/fabm.git. The BFM code adapted to work under the FABM convention is available at https://github.com/inogs/bfmforfabm.git (branch dev_1D, accessible on request) and the bio-optical model in https://github.com/BIOPTIMOD/Forward_Adjoint.git. ParSAC is available from PyPI, can be installed by executing the command: *pip install parsac <--user>*, and it requires Parallel Python. The 1D setup where the simulations presented can be

run and the configuration files for the optimization can be obtained from https://github.com/ealvarez-s/setup_BFM1D_CDOM.git.

**Data Availability**

The gridded product of temperature and salinity at DYFAMED was obtained from the OceanSITES Global Data Assembly Center via ftp (ftp://ftp.ifremer.fr/ifremer/oceansites, accessed 22/11/2021). Nitrate, nitrite and phosphate concentrations at

DYFAMED were obtained from https://doi.org/10.17882/43749 (Coppola et al., 2021). Chl-a and HPLC data at BOUSSOLE are available from http://www.obs-vlfr.fr/Boussole/html/boussole_data/other_useful_files.php (login as guest required). IOPs data at BOUSSOLE are available upon request to IMEV. Satellite data are available from the E.U. Copernicus Marine Service Information at https://doi.org/10.48670/moi-00299. The results of the numerical simulations can be found in https://doi.org/10.5281/zenodo.7628060 and scripts for data analysis and visualization are available through

https://github.com/ealvarez-s/results_BFM_CDOM.git. The analysis of model output has been performed with *R* 4.0.5 using the packages *RNetCDF* 2.5-2 and *RSQLite* 2.2.12 for importing data, *abind* 1.4-5, *akima* 0.6-2.1, *chron* 2.3-56, *lubridate* 1.8.0, *Hmisc* 4.5-0 and *stringr* 1.4.1 for data analysis, and *plot3D* 1.3, *unikn* 0.6.0 and *viridisLite* 0.4.1 for visualization (all publicly available in CRAN).

**Author contribution**

EÁ, GC and PL conceptualized the research; KB provided support for GOTM; JB provided support for FABM and ParSAC: EÁ, GC, AT, JB and PL completed the porting of BFM to FABM; VV and DA provided observational data; EÁ performed the optimization and simulations, did the analysis and wrote the original draft. GC, SC and PL provided funding through the SEAMLESS and NECCTON projects, and VV and DA provided funding through the BOUSSOLE project. All authors contributed to writing, reviewing and editing, and approved the final manuscript.

**Competing interests**

One of the co-authors (SC) is a member of the editorial board of Biogeosciences.

**Acknowledgements**

This work has received funding from the European Union's Horizon 2020 research and innovation programme under grant agreement No 101004032 (project SEAMLESS) and the Horizon Europe research and innovation action under grant agreement

No 101081273 (project NECCTON). All simulations were conducted on the GALILEO100 supercomputing facility at the CINECA consortium; we acknowledge CINECA for providing high-performance computing resources and support through the *IscrB_3DSBM* and the *tra21_seamless* projects. Satellite products were provided by the Copernicus Marine Environment Monitoring Service (CMEMS), implemented by Mercator Ocean in the framework of a delegation agreement with the European Union. The BOUSSOLE project was funded by the *Centre National d'Etudes Spatiales* (CNES) and the European

Space Agency (ESA) and supported by the French Oceanographic Fleet (FOF). We are grateful to Emilie Diamond and Melek



Golbol for conducting monthly cruises, to Annick Bricaud for CDOM analyses and to Joséphine Ras and Céline Dimier for the phytoplankton pigment analyses (SAPIGH analytical platform of the *Laboratoire d'Océanographie de Villefranche*, CNRS, France). The DYFAMED time series have been provided by the *Institut de la Mer de Villefranche* (L. Coppola) and the project funded by CNRS-INSU and ALLENVI through the MOOSE observing network. Data were collected and made
freely available by the international OceanSITES project and the national programs that contribute to it.

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
