# Peer review of "Chromophoric dissolved organic matter dynamics revealed through the optimization of an optical-biogeochemical model in the NW Mediterranean Sea"

_Biogeosciences, 2023_

## Author Comment (AC1)

**RC1: 'Comment on bg-2023-48', Anonymous Referee #1, 03 Apr 2023**
Citation: https://doi.org/10.5194/bg-2023-48-RC1

The paper titled "Chromophoric dissolved organic matter dynamics revealed through the optimization of an optical-biogeochemical model in the NW Mediterranean Sea" presents a coupled 1D physical-biogeochemical-optical modelling suite inbedded in a parameter optimization tool. The models are run at the Boussole site in the Ligurian Sea. The large observational database allows tuning the model parameters (except a few parameters who could not be sufficiently constrained by the observed variables).

The authors show a convincing case. The paper is very clear and well written and the figures are relevant. The fully coupled approach is novative. The parameter optimization is very relevant and seems to improve all model variables. The authors also present important possible limitations of the method and justify the choices made (e.g. in section 4.1).

We appreciate the constructive comments and suggestions from the Reviewer. We present our point-by-point responses to the Reviewer's comments below. The Reviewer's comments are in black, our responses follow each comment in blue. In each response, we detail the changes we propose to make to the manuscript in order to address these comments.

Therefor I have no major comments. I only would have been interested to see a couple of details explained:

(1) the authors state themselves that it remains a future objective to see how the optimized parameteres will behave in a 3D model. It is indeed well known that parameter optimization can be sensitive to particular configurations (see e.g. https://egusphere.copernicus.org/preprints/2023/egusphere-2023-363/#discussion for a very recent example). For example it can compensate poorly represented features by adapting other features as the authors also propose (line 570). Based on their experience, could the authors estimate if the model parameters would lead to realistic results in e.g. areas with more lateral contributions compared to the Boussole station ? The switch from one model to another (GOTM to NEMO) may also be discussed (if relevant). However if no trials have been realized yet, I do not suggest that the authors need to speculate.

Upscaling of the 1D results to the 3D Mediterranean domain is an ongoing activity. Thus, the comment raised by the reviewer is relevant. Even if we have not finalized trails to analyze these aspects yet, we will rewrite and expand the paragraph in the Discussion that covers the feasibility of applying the locally optimized parameters to 3D models, to cover those two aspects mentioned by the reviewer: the robustness of our results in areas with more lateral contributions compared to the BOUSSOLE station, and which can be the consequences of changing the transport model. The new paragraph will read as follows:

'CDOM cycling models are useful only if they are flexible enough (i.e. portable) to provide accurate predictions for a wide range of conditions occurring in marine ecosystems. In that case, they can be integrated into larger 3D models of BGC cycles. The possibility of applying

parameter optimization techniques in computationally efficient 1D models before using the resulting parameters in the 3D version seems advantageous. Furthermore, parameters optimized with surface data only (i.e. satellite retrieved) may present inconsistencies when simulating underwater biological fields (Wang et al., 2020). This is particularly problematic in oligotrophic regions where the deep Chl-a maximum (DCM) is relatively deep and poorly correlated with surface satellite observations (Cullen, 2015; Fennel and Boss, 2003). Optimizing parameter values at intensively sampled 1D sites can overcome this problem and improve the simulation of both vertical and seasonal variability. Further analysis will be required to determine whether the optimal parameter sets from the 1D version should be applied directly in the 3D model. This decision should balance the risk of a lower agreement between model and data in 3D than in the 1D counterpart, with the opportunity to preserve key features presented in this analysis. These features include seasonal dynamics and vertical gradients, and the higher-than-average CDOM light absorption coefficients (Sections 3.1 and 3.2). Since the approximation of a 1D configuration is reasonable at the BOUSSOLE site, we do not expect that our proposed parameter values are compensating for the absence of lateral transport. However, considering that biogeochemical optimization can be influenced by specific physical forcing and possible biases in circulation dynamics (Pasquier *et al*., 2023), further analysis is needed to upscale the present 1D results to the 3D Mediterranean domain, where lateral contributions and vertical mixing are different and can interact differently with the optimized biogeochemical processes.'

(2) can the authors explain if there is any limit imposed on parameters ranges during the creation of new values by the genetic algorithm ? Positivity, statistical distribution, inter-relations or consistency between pairs of parameters, ... ?

The limit imposed on parameters ranges is the range itself that is given as a minimum and maximum value (indicated in Table A1), being the minimum always positive. Therefore, this imposes positivity for all of them. The statistical distribution of parameter values is handled by ParSAC, the initial populations of parameters are distributed uniformly between minimum and maximum values. Regarding inter-relations between pairs of parameters, the unique constraint is that all parameters included in the optimization must appear independently of each other in some process. In case two parameters appear always in the same process, we chose one to be optimized. This does not exclude correlations between parameters, that we have acknowledged as a limitation of our results in the discussion (Section 4.1). We have not imposed any condition for the consistency between pairs of parameters, or any trade-off among them.

We will rewrite Section 2.4.1 'Parameters optimized' to clarify earlier the conditions that we imposed to the parameter list to be optimized and the parameter ranges. The new text will read as follows:

'The potential number of independent parameters included in the optimization problem is limited by the observations available and by the fact that optimized parameters must appear in some process independently. A total of 25 optical and BGC parameters were optimized (**Table A1** in Appendix A), all of them were given a positive range of values for initialization and evolution, without conditions or trade-offs for the consistency between pairs of parameters. The rest of the parameters not included in the optimization kept constant values (**Table 1**).'

(3) the model-satellite comparison is realized over the 9 upper meters. Can the authors justify this choice e.g. in relation to observed optical depth during the year ? Are the model variables simply averaged over the layers corresponding to these 9 meters ?

Yes, model variables are just averaged over nine meters throughout the whole seasonal cycle. In the 4-year time series considered, the first optical depth -FOD- (computed as $z_{EU}/4.6$) ranges from 4.8 to 28 m. However, 90% of the values (percentiles 0.05 to 0.95) range from 7.9 to 15.1 m. As a result, the time series of Figure 7 averaged over the first optical depth (dark blue in the figure below) are very similar to the time-series averaged over the first 9 m of the water column (light blue as the original Figure 7, almost indistinguishable). There is a slight difference in the summer periods when the FOD is slightly deeper than 9 m. We will include a new version of Figure 7 that averages the *Optimized* results over the first optical depth and will mention in the text the range of values the FOD takes throughout the year.

[Figure]

(4) there is a type line 800, "parameterisation" (without "s")

Thanks, this will change to: "We have proposed a simplified parameterisation to account for the effects of light and nutrients on CDOM production."

---

## Author Comment (AC2)

**RC2: 'Comment on bg-2023-48', Anonymous Referee #2, 05 Apr 2023**
Citation: https://doi.org/10.5194/bg-2023-48-RC2

The authors couple a BGC-optical model with a 1-dimensional physical ocean model for a site in the NW Mediterranean Sea, with the goal of better resolving dynamics of CDOM. It is nice to see an emphasis on modeling CDOM in a coupled physical-BGC model. However, some work is required to improve the manuscript and make it more accessible to readers.

We appreciate the constructive comments and suggestions from the Reviewer. We present our point-by-point responses to the Reviewer's comments below. The Reviewer's comments are in black, our responses follow each comment in green. In each response, we detail the changes we propose to make to the manuscript in order to address these comments, and include the proposed modified text and/or figure.

**general comments**

I have a couple of methodological questions concerning the optimization procedure. Firstly, the authors declare the estimated 25 parameters "optimal" without too much evidence to back it up. Interdependence between parameters (which the authors mention in the manuscript) and other factors can lead to complex cost functions with local minima. Based on the manuscript and appendix A, it appears like the one large optimization experiment was conducted, using a large number of iterations. I would encourage performing a few more experiments starting from different initial parameter values, in order to assess the new parameter estimates are similar to the initial ones. Here it could be enough to perform fewer iterations and examine the convergence. If this type of experiment was already performed, this information should be included in the manuscript so that readers (like me) are not left wondering if the reported results are based on parameter values from a local minimum.

Differential Evolution is a global optimization method that searches for the absolute extremum of the function over its entire definition domain. The risk of becoming trapped in a local minimum is comparatively lower compared to direct search approaches. As a parallel search technique, it has some built-in safeguards to forestall misconvergence. For example, by running several vectors simultaneously, superior parameter configurations can help other vectors escape local minima. In fact, DE shows consistent convergence to the global minimum in consecutive independent trials. A complete evaluation about the resilience of the DE algorithm to fail into local minima is given in Storm and Price 1997. We are also following the recommendations of this publication for the values of the metaparameters NP, F and CR.
We will add the following statement in Section 2.4.2 'Parameter optimization method':

'Genetic algorithms are particularly robust in overcoming local minima (Storn and Price, 1997). As a parallel search technique that runs several vectors simultaneously, DE sampling method can help vectors escape local minima and therefore avoid misconvergence.'

Nevertheless, we do agree with the reviewer that it would be interesting to see different trials of this optimization experiment. We disagree in the fact that only the iteration over a few generations will be necessary. From our point of view, this analysis would require to repeat the whole optimization experiment completely several times. Otherwise, we do not think the results will be meaningful. These new set of experiments would be time consuming and computationally expensive. We will consider doing these additional experiments provided the reviewer still considers them essential, and we are given the necessary time.

Secondly, the authors perform two experiments (named "Nutrients" and "Constant") in which certain parameters are held constant at some average values. Fig 12 shows some of the results, indicating that the constant values have a worse fit to data. However, one could argue that this is not an entirely fair comparison, as the constant values were not allowed to be optimized. That is, in how far would results and their interpretation change if the values of the constant parameters were adjusted, perhaps alongside all other parameters. I understand that the experiments were mainly conducted to examine the effect of underlying mechanisms that make these parameters non-constant, and I am not sure if this requires new experiments, but it would be nice to see at least some comment or discussion about the role of the constant value that was selected.

We do agree that Experiment 1 does not compare three possible optimized configurations, and as such, the two non-optimized ("Nutrients" and "Constant", names will change in the revised version) are in disadvantage with respect to *Optimized*. But the objective of the experiment is to highlight the vertical and temporal differences in CDOM production among formulations. Therefore, we will clarify this objective including the following statement in Section 3.3.1 In situ production of CDOM and synchronization with phytoplankton Chl-a: 'The constant % values in *Nutrients* and *Constant* were chosen to homogenise the values of $f_{R2}^{X2}$ at the beginning of the year between the formulations and to highlight the vertical and temporal differences in $f_{R2}^{X2}$ between the three formulations.'

Parts of the manuscript are well written and easy to follow, but in some parts I had to read sentences many times over to try to understand what is meant. Let me start with the notation: In the differential equations, many terms are abbreviated using a vertical line with a dx/dt part on the left and a 3-letter description and a variable on the right. This notation is never explained and, at first, I was under the impression the vertical line had a mathematical meaning (e.g., "evaluated at"). After going over equation 1 and accompanying text again, I noticed that this assumption was probably wrong. Instead, the variable on the bottom right of the vertical line may be the other variable this part of the equation is dependent on. I still don't exactly know, because this notation is never described in the manuscript. A couple of sentences would really help to make this notation accessible.

We will clarify notation by including new text in the first paragraph in Section 2.1.2 'The BGC model BFM and the bio-optical component': 'Following Vichi et al. (2007), the dynamical equations of this section are written in 'rate of change' form, where the right hand side contains the terms representing significant processes for each living or non-living component. For each process, the state variable subject to change is indicated before the vertical bar and the superscript and subscript after the bar are the 3-letter

acronym of the process represented and the state variable involved in the process, respectively.'

Similarly, the variable notation is not explained properly, and the reader is left guessing or is required to find more information in other parts of the manuscript. A small example: There are 4 zooplankton, which are (oddly) denoted Z^{(3)} to Z^{(6)}. Why is there no Z^{(1)}? What are the kind of zooplankton is represented by each variable? Why do only two of them appear in Eq 8 and 9? Some information can be obtained from Fig 1, but this information should be much more front and center, so that these questions do not even arise. Again, a few extra sentences would make the manuscript a lot more accessible. I mention a few more instances when I encountered text without enough explanation in my specific comments below.

We will modify the notation for the four zooplankton groups (line 168) from numbers 6, 5, 4 and 3 to acronyms: HETN (heterotrophic nanoflagellates), MICRO (microzooplankton), OMNI (omnivorous mesozooplankton) and CARNI (carnivorous mesozooplankton), respectively. This hopefully will avoid the question about the lack of 1 and 2, and will clarify the kind of zooplankton each variable represents.

Only two of them appear in Eq 8 and 9 because mesozooplankton (Z3=CARNI and Z4=OMNI) do not excrete dissolved carbon, only particulate. That's because the excretion of Z3 and Z4 is supposed to represent the production of fecal pellets and egestion (sloppy feeding). We will clarify as follows:

'Labile DOC ($R_C^{(1)}$) is produced by the excretion of microzooplankton ($Z^{(MICRO)}$) and nanoheterotrophs ($Z^{(HETEN)}$) (*exc*), and therefore represents the sources of DOC associated to the zooplankton-mediated mortality of phytoplankton and bacteria, and it is consumed quickly by bacteria. The released fraction of carbon by mesozooplankton excretion (both $Z^{(OMNI)}$ and $Z^{(CARNI)}$), on the other hand, is assumed to have no dissolved products and therefore directed to the state variable $R_C^{(DET)}$.'

Additionally, and in response to some of the specific comments, we will modify the superscripts of other state variables, as follows:

$P^{(1)}$ (diatoms) => $P^{(DIATOM)}$
$P^{(2)}$ (nano-flagellates) => $P^{(NANO)}$
$P^{(3)}$ (picophytoplankton) => $P^{(PICO)}$
$P^{(4)}$ (dinoflagellates) => $P^{(DINO)}$
$R^{(6)}$ (detritus) => $R^{(DET)}$
$O^{(3)}$ (dissolved inorganic carbon) => $O^{(DIC)}$
$N^{(1)}$ (phosphate) => $N^{(PO4)}$
$N^{(3)}$ (nitrate) => $N^{(NO3)}$
$N^{(4)}$ (ammonium) => $N^{(NH4)}$

The names of the simulations and experiments could be improved: The simulation names "Nutrients", "Constant" do not help the reader understand what is being modified or held constant. In comparison, The "Optimized" experiment has a better name. Then the

experiment names EXP-X^{(2)} and EXP-X^{(3)} are maybe less helpful than the simulation names: What is the significance of "X" in the name, and why start counting at 2? Just using "Experiment 1" and "Experiment 2" would raise fewer questions, but a short descriptive name would be even better.

Thanks for the suggestion, we will change the name of the simulations to give them short descriptive names. *Nutrients* and *Constant* will be changed to indicate what is being held constant.

Nutrients => Constant leakage.

Constant => Constant *dpp*.

And for consistency *Bleaching* will be also modified to indicate in which sense photobleaching rate is being modified:

Bleaching => Intense bleaching.

Regarding experiments EXP-X(2) and EXP-X(3), we will use short descriptive names and avoid numbering them:

EXP-X(2) (experiment 1) => EXP-Biology

EXP-X(3) (experiment 2) => EXP-Physics

There were a two instances showing a clear break in the train of thought, and I would suggest reordering the text: Firstly, Section 2.4.1 "Parameter optimization method" briefly introduces the parameter optimization method and then goes into detail (often difficult to follow, see specific comments) about which parameters were chosen to be optimized, before returning to the optimization method. Here, I would suggest moving the part about the parameters into its own subsection.

This Section 2.4.1 will be split into two: 2.4.1 'Parameters optimized' that describes which parameters have been chosen to be optimized, and 2.4.2 'Parameter optimization method' that describes the DE algorithm.

Secondly, Section 2.4.2 briefly introduces the hypothesis testing experiments without motivating these experiments well. Pages later, in Section 3.3.1, additional context is provided that is very useful to better understand the setup of the experiments. I suggest merging information from the two texts, so that the parameter changes performed are easier to understand and well motivated from the start.

Section 2.4.2 is now Section 2.4.3 'Simulations and hypothesis testing experiments'. We will reintroduce the hypothesis testing experiments:

'The model configuration described in Section 2.3 and the optimized parameter values obtained as described in Section 2.4.2 represents the *Optimized* model configuration. We used a series of hypothesis-testing experiments to investigate the mechanisms that determine the vertical and temporal patterns of CDOM distribution at the BOUSSOLE site. We conducted three more simulations in which we changed some of the assumptions of the *Optimized* configuration. All simulations are summarized in **Table 2**, all were run for 6-years with two years of spin-up and analyzed the results of the last four years (January 2011 to December 2014).'

Also, in response to the specific comment l 479 we will reformulate the description of each experiment.

I really like the figures, which look good and are informative. Here, I just have one general comment, I think it would be useful to consistently add description and units to their respective axis or the color bar. For example, in Fig 12, move the "a_CDOM(450)" and units in (d) and (e) to the y-axis, just like in panels (a-c). In Fig. 3 and similar, move property name and units to the color bar. This step would reduce text in the panels and avoid repeating units.

Thanks for the suggestions, we will make the following changes to labels and axis:
Figure 12: panels d and e, move label to y axis; panels b and c, add 'surface' and 'DCM'.
Figures 3, 4, 5 and 6: move variable name to color bar.
Figure 7: move variable name to y axis.
Figure 13: move variable name to y axis, keep depth in the panel.

**specific comments**

l 18: "The absorption coefficient of CDOM, [...], is measurable in situ and remotely": Can it indeed be measured reliably remotely and distinguished from other (living) light absorbing constituents?

No, remotely CDOM cannot be distinguished from detritus, not at least in current products. We will rewrite this part of the abstract: 'The absorption coefficient of CDOM, $a_{CDOM}(\lambda)$, is measurable in situ and can be retrieved remotely, although it is not distinguishable from the absorption of non-algal particles. These observations can be used as indicators for the concentration of other relevant biogeochemical variables in the ocean, e.g. dissolved organic carbon.'

Similarly, we will also rephrase the second paragraph of the introduction: '$a_{CDOM}(\lambda)$ can be measured in situ at selected locations and be retrieved on a global scale from remote-sensing platforms, although remotely it is not distinguishable from the absorption of non-algal particles.'

l 40: "the spatio-temporal dynamics [...] reflects the functioning of the carbon cycle": What exactly is meant here, I would suggest rephrasing.

We will rewrite this sentence to: 'Furthermore, as the largest pool of reduced carbon in the oceans, it plays an important role in the global carbon cycle (Legendre et al., 2015).'

l 41: "non-absorbing": I would suggest changing this to "not light absorbing".

Thanks for the suggestion, we will rewrite this sentence to: 'Although most of the dissolved exudates that form the DOM are non light absorbing (Mühlenbruch et al., 2018), a fraction of DOM absorbs light mainly in the ultraviolet (UV) and blue spectral range of the electromagnetic radiation.'

l 50: "There are several possible factors determining...": "Besides the presence of CDOM, there are other possible factors influencing..."

Thanks for the suggestion, we will rewrite this sentence as: 'Besides the presence of CDOM, there are other possible factors influencing this optical behavior, including: the particular pigment ratios in the Mediterranean phytoplankton community (Organelli et al., 2011), the abundance of small coccolithophores (Gitelson et al., 1996), and the influence of Saharan dust (Claustre et al., 2002).'

l 63: Since vertical transport is mentioned here, why not horizontal as well? Maybe change "terrestrial and atmospheric inputs" to something like "terrestrial and atmospheric CDOM sources and physical transport"?

Thanks for the suggestion, we will rewrite this sentence as: 'CDOM cycling is essentially controlled by in situ biological production (Romera-Castillo et al., 2010), terrestrial and atmospheric sources and physical transport, microbial consumption (Nelson and Gauglitz, 2016; Legendre et al., 2015; Stedmon and Markager, 2005), as well as deep ocean circulation and/or vertical mixing (Coble, 2007), and it is photoreactive and efficiently destroyed in the upper layers of the water column by solar radiation (Mopper and Kieber, 2000).'

l 73: Is the NW Mediterranean Sea the study region?

No, in this paragraph we were referring to the higher-than-average DOM deposition fluxes reported for the whole Mediterranean Sea, thanks for pointing out. We will rewrite this sentence as: 'In the basin, the higher-than-average CDOM concentrations seem to be sustained by allochthonous sources, ...'

l 75: "depositions from the atmosphere are 2–5 times larger in the Mediterranean Sea than in the oceans": higher than "in the oceans" is not very specific: higher than anywhere in the ocean, higher than some average?

We do agree, we will rewrite this sentence as: '... as fluxes of DOM depositions from the atmosphere are 2–5 times larger in the Mediterranean Sea than those estimated for the global ocean, which explains the abundance of humic-like substances (Santinelli, 2015; Galletti et al., 2019).'

l 85: What does "probably controlled" mean? Perhaps change to "mostly controlled" or "its dynamics are dominated by"?

and

l 86: One can maybe make the counterpoint that photoacclimation and nutrient limitation are also often controlled by vertical transport. Maybe distinguish between being controlled by and directly caused by, as the preceding sentence suggests?

We will rewrite this sentence as: 'While $a_{CDOM}(\lambda)$ dynamics at the surface is mainly explained by vertical transport of CDOM from depth and by photochemical degradation,

Chl-a concentration at the surface is mainly driven by photoacclimation of phytoplankton and nutrient limitation.'

l 92: "observations in the NW Mediterranean Sea suggest direct CDOM production by phytoplankton": Is this then equivalent to the way it is handled in other models, i.e. the fixed fraction of the dpp of phytoplankton?
This is equivalent to a fraction of the dpp of phytoplankton, not necessarily a fixed fraction.

l 169: Explain the symbols and superscripts used here ("O^{(3)}" etc.).
We will change the numbers on the superscripts for more self-explanatory names, as follows:
'The implementation of the BFM in FABM comprises 54 state variables. These include representations of dissolved inorganic carbon ($O^{(DIC)}$), inorganic forms of nitrogen ($N^{(NO3)}$), ammonium ($N^{(NH4)}$) and phosphorus ($N^{(PO4)}$), four phytoplankton types ($P^{(DIATOM)}$, $P^{(NANO)}$, $P^{(PICO)}$ and $P^{(DINO)}$), heterotrophic bacteria (B) and four grazers ($Z^{(HETN)}$, $Z^{(MICRO)}$, $Z^{(OMNI)}$ and $Z^{(CARNI)}$), three pools of dissolved organic matter ($R^{(1)}$ to $R^{(3)}$) and CDOM ($X^{(1)}$ to $X^{(3)}$) differentiated by reactivity, and particulate organic matter ($R^{(DET)}$).'

l 171: "A subscript appended to each module ...": This sentence requires more explanation. What are the modules? Is Chl-a considered an elemental constituent? I think I understand the rest of this paragraph, but this sentence is difficult to follow and seemingly unrelated to the preceding and following sentences.
We will clarify as: 'The subscript appended to each living and non-living component indicates the elemental constituents among carbon (C), nitrogen (N), phosphorus (P) and silica (S, only in $P^{(DIATOM)}$), and the content in Chl-a only in PFTs.'

l 184: What is "OAC"?
Generally, OAC accounts for 'optically active constituents'. In this work, we refer to those as optical constituents and therefore the use of OAC is a typo. We will rewrite this sentence as: '... the optical treatment of constituents (PFTs, CDOM and detritus) …'

Eq 1: "; i = 1 to 4" -> " for i = 1,2,3,4"
i = 1,2,3,4 will be included in Eq. 1 (phytoplankton) and i = 1,2,3 in Eq. 13 (cdom).

Eq 1: Why does zooplankton indexing start at 3?
We will modify the notation for the four zooplankton groups (line 168) from numbers 6, 5, 4 and 3 to acronyms: HETN (heterotrophic nanoflagellates), MICRO (microzooplankton), OMNI (omnivorous mesozooplankton) and CARNI (carnivorous mesozooplankton), respectively. The counter j will take values from 1 to 4, it will be modified in Equations 1, 8 and 9.

l 191: As an aside, I find the "prd" abbreviation of "predation" slightly confusing, as I always think of "production" first. I would suggest using "grz" for "grazing" instead.

'prd' is the 3-letter acronym that BFM uses conventionally to refer to predation. We will substitute 'prd' by 'grz' for clarity.

Eq 2: Underneath, it reads like both temperature and nutrients are dimensionless, when really these temperature- and nutrient-dependent factors are dimensionless. I would suggest changing "ft_P" to "ft_P(T)" where T is model temperature. The same applies to fn_P, which has the benefit of telling the reader which nutrients are entering this equation and control GPP. Currently, it might appear to a reader as if GPP is independent of other model variables.

We will clarify the formulation of the temperature- and nutrient-dependent factors as follows:

'Gross primary production of each phytoplankton species (mg C m$^{-3}$ d$^{-1}$) is computed as follows:

$$\frac{dP_C^{(i)}}{dt}\Big|_{O^{(3)}}^{gpp} = r_{0_P}^{(i)} \times ft_P^{(i)}(T) \times fn_P^{(i)}(nutrients)$$
$$\times \left( 1 - exp \left( \frac{-\phi^{(i)} \times \theta^{(i)} \times \int_{387.5}^{800} a_{PS}^{*(i)}(\lambda) \times E_0(\lambda)d\lambda}{r_{0_P}^{(i)}} \right) \right) \times P_C^{(i)}$$

where $r_{0_P}$ is the maximum productivity rate (d$^{-1}$), regulated by dimensionless temperature- ($ft_P(T)$) and nutrient-dependent ($fn_P(nutrients)$, nutrients are N$^{(PO4)}$, N$^{(NO3)}$ and N$^{(Si)}$ for P$^{(DIATOM)}$, and N$^{(PO4)}$ and N$^{(NO3)}$ for the rest of the phytoplankton types) factors.'

l 195, 197: Sometimes "mg Chl", sometimes "mg Chla" is used. To stay consistent with previous notation, I would suggest using "mg Chl-a" and "(mg Chl-a)^{-1}" when in the denominator.

Thanks for pointing this out. We will correct Chl to Chl-a throughout the text, including when it is mentioned in the Chl-a:C ratio.

Eq 4: I presume, P_P is the phosphorous quota (subscript P) of phytoplankton (regular P). Is the superscript denoting the phytoplankton type missing, or am I misunderstanding? Similarly, is the "P_N^{(k)}" meant to be "P_{N^{(k)}}"? That is, the superscript "(k)" appears to be attached to the P, but should it be attached to the subscript N instead?

Thanks for spotting this. The superscript (i) denoting the phytoplankton type was missing in the P quota and inserted incorrectly as (k) in the N quota. This Equation will be corrected as:

'$G_P^{balance}$ is calculated as:

$$G_P^{balance} = min \left( G_P, \frac{1}{p_P^{min}} \times \frac{dP_P^{(i)}}{dt}\Big|_{N^{(PO4)}}^{upt}, \frac{1}{n_P^{min}} \times \sum_{k=3,4} \frac{dP_N^{(i)}}{dt}\Big|_{N^{(k)}}^{upt} \right)$$

where $p_P^{min}$ and $n_P^{min}$ are the minimum phosphorous and nitrogen quota, respectively, and the two nitrogen sources are N$^{(NO3)}$ and N$^{(NH4)}$.

Eq 5: What is i set to here, is there perhaps a sum missing?
Thanks for spotting this, (i) is included because Eq. 5 indicates how $f_{R2}^{X2}$ is computed for each phytoplankton type, and not the coloured fraction of the total DOC of phytoplankton origin as the text says. We will keep referring to the coloured fraction of the total DOC of phytoplankton origin as $f_{R2}^{X2}$ (that is shown in Figure 1 and Figure 12), and will use $f_{P(i)}^{X2}$ to refer to the coloured fraction of the DOC produced by phytoplankton type i. Eq. 5 will therefore be rewritten to:

$$f_{P(i)}^{X2} = f_{P(i)}^{maxX2} \times \frac{\theta^{(i)}}{\theta_{chl}^{0(i)}} \times \frac{\beta_P^{(i)} \times \frac{dP_C^{(i)}}{dt}|\frac{gpp}{O^{(3)}}}{\frac{dP_C^{(i)}}{dt}|\frac{dpp}{R_C^{(2)}}}$$

and we will change equations 6 an 7 accordingly as:

$$\frac{dR_C^{(2)}}{dt} = \sum_{i=1,2,3,4} \left( \frac{dP_C^{(i)}}{dt}|\frac{dpp}{R_C^{(2)}} \times (1 - f_{P(i)}^{X2}) \right) - \frac{dB_C}{dt}|\frac{upt}{R_C^{(2)}}$$

$$dX_C^{(2)}/dt = \sum_{i=1,2,3,4} \left( \frac{dP_C^{(i)}}{dt}|\frac{dpp}{R_C^{(2)}} \times f_{P(i)}^{X2} \right) - \frac{dB_C}{dt}|\frac{upt}{R_C^{(2)}} - \frac{dX_C^{(2)}}{dt}|\frac{deg}{R_C^{(3)}}$$

The maximum fraction of coloured DOC in each phytoplankton type that is prescribed as a fixed parameter will be renamed as $f_{P(i)}^{maxX2}$, and Table 2, Table A1 and Figure A2 will be updated with the new name of the parameter.

l 231: Why do the other zooplankton not excrete labile DOC? Are they too small/large? This information is missing from the manuscript.
Line 232 states that 'Large zooplankton excretion … is composed by particulate only and directed to the state variable $R(6)$'. That's because the excretion of Z3 and Z4 is supposed to represent the production of fecal pellets and egestion (sloppy feeding).
We will clarify as follows: 'The released fraction of carbon by mesozooplankton is assumed to have no dissolved products and therefore directed to the state variable $R(6)$'.

Eq 14-16: Out of curiosity, why are different reference wavelength used in the 3 equations (450 nm, 440 nm and 550 nm)?
Because the reference wavelength is among the parameters obtained by fitting non-linear models to observed α$_{CDOM}$(λ), α$_{NAP}$(λ) and b$_{NAP}$(λ) spectra. In the basic formula for the specific absorption coefficient:
a=c2 x exp(-S(λ-λref))
c2, S and λref are all coefficients obtained when fitting the non-linear model to observed spectra (of both α$_{CDOM}$(λ) and α$_{NAP}$(λ)). For the absorption of CDOM, c2 scales the

absorption of CDOM at a reference wavelength ($\lambda$ref) of 450 nm (taken from Dutkiewicz et al. 2015, originally from the parameters fitted by Kitidis et al. 2006). For the absorption of NAP, c2 scales the absorption of NAP at a $\lambda$ref of 440 nm (taken from Gallegos et al. 2011, originally from the parameters fitted by Bowers and Binding 2006).

The same happens in the basic formula for the scattering coefficient:

b=c2 x ($\lambda$ref/$\lambda$)^e

where c2, e and $\lambda$ref are all found when fitting the non-linear model to observed spectra. In this case, c2 is the mass-specific scattering coefficient at a $\lambda$ref of 555 nm (originally from parameters fitted by Babin et al., 2003 to phytoplankton scattering spectra, Gallegos et al. 2011 used the same $\lambda$ref and adapted c2 to represent the scattering of several types of non-algal particles).

l 356: "()" are missing for a_PH and a_DG.

We will rewrite this part as: IOPs ($a$PH(443) and $a$DG(443)) are derived …'

Fig. 2: x- and y-axis labels are missing. Consider coloring land gray in the top left map.

Thanks for the suggestion. We will add '°E' and '°N' to the x- and y-axis respectively. We will color land grey in the small map of Figure 2.

l 378: "assumed equal": Why "assumed"?

'assumed' will be removed and the sentence left as follows: 'The parameter values related to the partition between dissolved and particulate excretion in $Z^{(MICRO)}$ and $Z^{(HETEN)}$, and the mortality of $Z^{(CARNI)}$ and $Z^{(OMNI)}$, are equal to the up-to-date values derived in Álvarez et al. (2022), together with all the optical parameters.'

l 389: "and restored at monthly frequency": Does this imply nudging/relaxation?

Yes, it does. We will rewrite this sentence as follows to clarify: 'Observed vertical profiles of T and S … were … used as initial conditions and constrained by a monthly nudging, in order to reproduce the intensity and timing of the mixing as closely as possible to observations.'

l 419: I would suggest removing the "while" here.

This 'while' will be removed as a result of the reorganization/reformulation of Section 2.4.1 'Parameters optimized' and Section 2.4.2 'Parameter optimization method' suggested in a previous comment.

l 425: What does "assumed true" mean, assumed that the true value is known and not included in the optimization? I would prefer "assumed known" or "assumed constant". Please make it explicit that this implies that the parameter in question is not included in the optimization. This needs to be made clear initially, the phrase is used a lot.

These parameters have not been included in the optimization. We will rephrase to clarify at the beginning of Section 2.4.1 'Parameters optimized' that all parameters not included in the optimization are assumed constant. We will also reformulate the whole section slightly in order not to repeat as much 'assumed constant'.

l 426: Subscript "(5 to 6)" and "(1 to 4)" are unconventional and could be confusing some readers. I suggest writing them out fully, i.e. (I am not including superscripts here): f_Z(5), f_Z(6), f_P(1), ..., f_P(4), and f_B.

This sentence will be changed to: 'We assumed constant the absorption coefficients of the three reactivities at 450 nm ($a_{X(1)}^{450}$, $a_{X(2)}^{450}$ and $a_{X(3)}^{450}$) and optimized parameters related to the production of CDOM ($f_{Z(MICRO)}^{X1}$, $f_{Z(HETEN)}^{X1}$, $f_{P(DIATOM)}^{X2}$, $f_{P(NANO)}^{X2}$, $f_{P(PICO)}^{X2}$, $f_{P(DINO)}^{X2}$ and $f_B^{X3}$) and those related to the photobleaching of $X_C^{(3)}$ ($b_{X(3)}$ and $I_{X(3)}$).'

l 432: "All optimized parameters appear in some process independently:": As a reader, I was now expecting a brief review of where the parameters mentioned in the previous sentences appear in equations. Instead, new parameters are mentioned and added to the list, of which two are not actually appearing in some process independently. Does the "all parameters" include the previously named ones? It would be helpful to break up this long sentence into two, don't use "all parameters" (one could use "the following" or similar).

Thanks for pointing out, we agree that the parameter appearing in an independent process is a condition that all parameters included in the optimization must meet, and in fact is met in our study. We will move this consideration to the beginning of the section, as it applies to all parameters. For phytoplankton, we will reformulate to avoid the use of "all parameters" and to clarify that we included in the optimization the two parameters that appeared in two different processes (the absorption cross section and the maximum Chl-a:C quotum), and chose one of the two that appear together in photosynthesis (the absorption cross-sections of photosynthetic pigments and the photochemical efficiency).

l 435: "We assumed true a_PS and optimized \phi_C, because these parameters are not well documented in literature being more difficult to measure.": Does this mean \phi_C was included in the optimization and a_PS was not, because they both appear in the same equation, and they may be dependent on each other?

Yes, the absorption cross-sections of photosynthetic pigments (a_PS) and the photochemical efficiency (phi_C) only appear multiplied in the photosynthesis equation. Their product is the equivalent to the so-called initial slope of the production-irradiace (PE) curve (alpha), used in most of the non-spectrally resolved formulations of the PE relationship.

l 441: "A total of 25 optical and BGC parameters were optimized": I think it would be helpful to mention this at the beginning of the paragraph.

We will move this sentence to the beginning of Section 2.4.1 'Parameters optimized'.

After the modifications stated in comments from l425 to l441, the rewritten Section 2.4.1 will read as follows:

'Observations used for optimization were all collected at the BOUSSOLE site at monthly temporal resolution and roughly at the same number of discrete depths and included pico-, nano- and micro-phytoplankton Chl-a (pico-Chl-a, nano-Chl-a and micro-Chl-a, respectively), $a_{PH}(450)$, $a_{NAP}(450)$, $a_{CDOM}(450)$ and $S_{CDOM}^{350-500}$. The potential number of independent parameters included in the optimization problem is limited by the observations available and by the fact that optimized parameters must appear in some process independently. A total of 25 optical and BGC parameters were optimized (**Table A1** in Appendix A), all of them were given a positive range of values for initialization and evolution, without conditions or trade-off for the consistency between pairs of parameters. The rest of the parameters not included in the optimization kept constant values (**Table 1**). The only observations related to CDOM were $a_{CDOM}(\lambda)$ that depend both on the mass-concentration and on the mass-specific absorption coefficients across the electromagnetic spectrum. We assumed constant the absorption coefficients of the three reactivities at 450 nm ($a_{X(1)}^{450}$, $a_{X(2)}^{450}$ and $a_{X(3)}^{450}$) and optimized parameters related to the production of CDOM ($f_{Z(MICRO)}^{X1}$, $f_{Z(HETEN)}^{X1}$, $f_{P(DIATOM)}^{X2}$, $f_{P(NANO)}^{X2}$, $f_{P(PICO)}^{X2}$, $f_{P(DINO)}^{X2}$ and $f_B^{X3}$) and those related to the photobleaching of $X_C^{(3)}$ ($b_{X(3)}$ and $I_{X(3)}$). The spectral slope of the CDOM pool has been associated with aromaticity and average molecular weight of the CDOM compounds (Blough and Green, 1995) but appears to be less linked to CDOM concentration. Therefore, the spectral slopes between 350 and 500 nm for the three reactivities ($S_{X(1)}^{350-500}$, $S_{X(2)}^{350-500}$ and $S_{X(3)}^{350-500}$) were optimized to ensure the proper simulation of $a_{CDOM}(\lambda)$ in wavebands other than 450 nm. For phytoplankton, independent observations regarding mass-concentrations (Chl-a) and optical properties ($a_{PH}(\lambda)$) were available, therefore both types of parameters were optimized. We optimized the absorption cross-sections ($\underline{a}_{PH}^*$) and the maximum Chl-a:C quota ($\theta_{chl}^0$) as they appear independently in light attenuation (Eq. (17)) and in photoacclimation (Eq. (19) in Álvarez et al., 2022) and in CDOM production (Eq. (5)), respectively. The absorption cross-sections of photosynthetic pigments ($\underline{a}_{PS}^*$) and the photochemical efficiencies ($\phi_c^0$) of the PFTs appear only in photosynthesis (Eq. (2)). We kept constant $\underline{a}_{PS}^*$ and optimized $\phi_c^0$, because these parameters are not well documented in literature being more difficult to measure. For detritus, we optimized the reference absorption coefficients at 440 nm ($a_R^{440}$) and kept constant all BGC parameters that alter the mass-concentrations. Given that $a_{NAP}(\lambda)$ observations were available, optical parameters related to the spectral shape of absorption by detritus ($S_R^{350-500}$) could have been included in the optimization, but we decided to maintain this parameter constant given the small contribution of $a_{NAP}(\lambda)$ to the total non-water absorption as compared to CDOM and phytoplankton. All parameters related to scattering and backscattering both of phytoplankton ($\underline{b}_{PH}^*$ and $bbr_{PH}$) and detritus ($b_R^{550}$, $e_R$ and $bbr_R$) were not included in the optimization.'

l 449: "O_1, ..., O_n, O_N": This notation is confusing, why include n and N here, is n=N-1? The same applies to P.

We will remove this notation. The sentence will read: 'For any variable observed m from a total of M, there will be N number of pairs consisting of the simulation $P_n$ and the corresponding observation $O_n$.'

Eq 27: Similar confusion, I assume n iterates from 1 to N here. So it should be "n = 1, ..., N" or (more conventional) "n=1" and the "N" is placed on top of the sum. The same applies to equation 29.
We will correct the Σ on both equation 27 and Eq. 29 to place "n=1" and "m=1", respectively, on the bottom of the sum and "N" and "M", respectively, on top of the sum.

Eq 29: So N_m is the number of observations for variable m? This needs to be made explicit, and I suggest introducing this notation right away in Eq 27, so that it is clear what SSQ_m is. I probably do not understand the terms here and what the subscript m indicates, it appears like SSQ_m/(2*\sigma_m) = SSQ_m/(2*SSQ_m/(N_m-1)) = (N_m-1)/2? This is very confusing.
We will clarify that m refers to each variable observed of a total of M, and n refers to each pair observation-prediction from a total of N. In the corrected notation, the number of observations of a given variable m is denoted as $N_m$ and the sum of the squares as $SSQ_m$. Regarding the formulation of the lnLikelihood, in the current version of the manuscript we gave the general formulation for the lnLikelihood as it is computed by ParSAC. This formulation can use a standard deviation for the observations that is manually provided by the user, or, if the sd is not specified (this work), ParSAC estimates it from the model-observation differences (Eq. 28). In the first case, the $SSQ/2sd^2$ terms (sd-dependent) in the lnLikelihood (Eq. 29) are not constant, and can therefore not be dropped. We agree with the reviewer that in our particular application, where we do not provide a known sd for each variable and compute them instead from the SSQ (second case), the last term in the lnLikelihood can be dropped. Therefore, for the sake of clarity, we will simplify Eq. 27, 28 and 29, to just two equations as follows:

'ParSAC formulates such fitness (i.e. probability that the candidate parameter values are the true parameter set representing reality) as a multi-objective function calculated as the log-transformed likelihood between the outcome of the model and the observations provided, assuming the observed values are log-normally distributed, with the median equal to the model prediction. For any variable observed m from a total of M, there will be N number of pairs consisting of the simulation $P_n$ and the corresponding observation $O_n$. The sum of the squares of the residuals for each variable observed m ($SSQ_m$) is computed as:

$$SSQ_m = \sum_{n=1}^{N_m} [P_n - O_n]^2$$

For M variables observed, all differences between model and observations are combined as:

$$lnLikelihood = \sum_{m=1}^{M} -N_m \times ln\,(SSQ_m)$$ '

l 460: The DE algorithm is not explained very well. For example, it is not clear from the text which of the symbols denotes the target vector and which is the mutant vector. What does "crossing" with the mutant vector mean? In "All target vectors in the first generation are crossed with the mutant vector", it sounds like the same mutant vector is applied to all target vectors. If I did not know how DE worked, I would not have a better idea after reading this section. This description needs to be rewritten.
We will rewrite this section as follows:

'Each individual in a given population of size S=288 is a 25-dimensional target vector that represents a candidate solution to the problem. The parameter values in the target vectors in the first generation are initialized randomly across the parameter space. The parameter values were sampled uniformly between minimum and maximum values, which are listed in **Table A1** (Appendix A). The DE algorithm creates new generations of individuals by applying cycles of mutation, crossover and selection operating on the target vectors. Mutation: once all simulations in a generation are completed, the DE algorithm creates a new set of S mutant vectors. Each mutant vector is created selecting randomly three targets ($x_{r1}$, $x_{r2}$ and $x_{r3}$) and applying a mutation operator that consisted on $x_{r1} + 0.5 \cdot (x_{r2} - x_{r3})$. Crossover: to increase the diversity of the parameter vectors, crossover is done to generate the final trial set of individuals. Crossover is performed between each target vector and its corresponding mutant vector. retaining the target vector in the population with a probability of 0.9, otherwise introducing the mutation only with a 0.1 probability. Selection: the final phase is selection where the lnLikelihood of the trial vector is compared to the lnLikelihood of the corresponding initial target. The trial vector replaces the target if the lnLikelihood value obtained from the trial is higher or equal than the lnLikelihood obtained from the target vector. After the selection procedure, the size of the population is again S. The cycle of mutation, crossover and selection is carried out iteratively and eventually, the DE algorithm provides an estimate of the optimal parameter values that minimizes the misfit between the model output and the observations (i.e. maximizes the sum of lnLikelihoods in the population). '

l 466: "optimal set of 25-parameter values": The DE algorithm aims to find optimal parameter values, but that does not mean that the resulting parameters are indeed optimal in that they maximize the likelihood. Hopefully, they are close to the global optimum. In other words, I would suggest using "estimate of the optimal parameter values".
We will take the reviewer suggestion and the sentence will read: 'After the simulation of a new generation, the cycle of mutation, crossover and selection is carried out iteratively and eventually, the DE algorithm provides an **estimate of the optimal parameter values** that minimizes the misfit between the model output and the observations (i.e. maximizes the sum of lnLikelihoods in the population).'

l 476: "reversed some of the assumptions of the optimized configuration": What does "reversed" mean here? I think "changed" might be more suitable.
We will use 'changed' instead of 'reversed'. Additionally, this first part of the Section 2.4.3 'Simulations and hypothesis testing experiments' will be rewritten in response to one of the general comments.

l 479: "The comparison of the results of Optimized with Nutrients and Constant constituted and experiment..." This sentence is difficult to follow, I suggest rephrasing. I had to read the previous sentences several times as well: one uses "removed the dependence >>of<< f on \theta/\theta_chl", the other "removed both dependencies >>from<< f, on \theta/\theta_chl", making it very difficult to understand what is dependent on what, and what is being removed. Here, I strongly suggest stating the aim of each experiment first and then go into details of the implementation.

Section 2.4.3 'Simulations and hypothesis testing experiments' will be rewritten in response to one of the general comments. The two experiments will be described separately as follows:

'In **EXP-Biology** we investigated the role of nutrient and light limitations on controlling the biogenic in situ production of biodegradable CDOM and therefore on generating the observed $a_{CDOM}(\lambda)$ values. This experiment consisted of comparing the results of *Optimized* with two additional simulations. In **Constant leakage**, we considered the proportion $X_C^{(2)}:R_C^{(2)}$ in *dpp* as a constant fraction of leakage and therefore removed the dependence of $f_{R2}^{X2}$ on $\theta/\theta_{chl}^0$. In **Constant dpp**, we considered the fraction $X_C^{(2)}:R_C^{(2)}$ in *dpp* constant and therefore removed both dependencies from $f_{R2}^{X2}$, on $\theta/\theta_{chl}^0$ and on leakage/*dpp*.

In **EXP-Physics** we investigated the role of vertically-transported CDOM in maintaining CDOM concentrations in the surface ocean. This experiment consisted of comparing the results of *Optimized* with an additional simulation, **Intense bleaching,** where we prescribed the same bleaching parameters for $X_C^{(3)}$ ($b_{X(3)}$ and $I_{X(3)}$) as for the other two CDOM pools and quadrupled the CDOM:DOC fractions in all biotic sources ($f_{Z(MICRO)}^{X1}$, $f_{Z(HETEN)}^{X1}$, $f_{P(DIATOM)}^{X2}$, $f_{P(NANO)}^{X2}$, $f_{P(PICO)}^{X2}$ $f_{P(DINO)}^{X2}$ and $f_B^{X3}$). By comparing *Optimized* with **Intense bleaching** we evaluated the extent to which the *Optimized* simulation was able to represent the role of $X_C^{(3)}$, and thus of any source of allochthonous CDOM, in generating the observed $a_{CDOM}(\lambda)$ values.'

Fig 3: Mention that these are climatological values in the caption.
The caption will be rewritten to:
'Figure 3. **Annually-averaged** vertical profiles as a function of time of (a-b) temperature, (c-d) nitrate and (e-f) phosphate, obtained from observations at the DYFAMED site (left panels) and from the *Optimized* simulation (right panels). The white line in the temperature panels represents the mixed layer depth (mld) obtained in (a) with a threshold of 0.2 °C on temperature (de Boyer Montégut et al., 2004) and in (b) with a threshold of $1\cdot10^{-5}$ m$^2$ s$^{-2}$ on turbulent kinetic energy. The line on (c) and (d) represents the nitracline (nitrate concentration equals 2 μM).'

Fig 3: Units in the color bar and in the panel are not identical (they are equivalent, but why create confusion).
We will remove the units from the panels and leave the units in the color bar, both in the nutrient panels of Figure 3 and the DOC panel of Figure 5.

l 568: Out of curiosity, what are the correlations between in situ and satellite observations?
Thanks for the question, we will evaluate the task of intercalibration exercise between observing systems in future works. For a detailed comparison between remotely and in-situ collected observations at the Boussole site, we suggest figures 40 (Chl-a, page 52) and 52 (CDOM, page 65) of the following report:
'Climatological characterization of Ocean Sites for OC-SVC-BOUSSOLE-D4'
https://www.eumetsat.int/media/49886

Fig 7: What happened in the model in early 2013 when there appears to be a drop of model chlorophyll to nearly zero?
It is the effect of the intensive winter mixing that homogenizes concentrations quite deep in the water column and takes chlorophyll almost to zero. It also happens in early 2012, although not so intensively. We have stated in the text that these periods have not been included in the metrics computation.

Fig 8 and 9: it might be helpful to change the order of legend entries to match the extra axes.
We will reorder the legend elements in order (top to bottom): Bc, aCDOM, TChl-a, DOC.

Fig 11b: Why are the ranges of reported values so patchy?
Because these are areas outlining clouds of points. We show below the original figures from Catalá et al. (2018) and Galletti et al (2019) with the ranges that we included in the background of Figure 11b (grey areas).

[Figure]

l 683: "1) depends": Missing "it".
We will correct this sentence as follows: 'Therefore, the fraction $X_C^{(2)} : R_C^{(2)}$ in the *Optimized* simulation was variable in time and depth for two reasons: 1) **it** depends on the relative

proportions of DOC from leakage or exudation and 2) **it** depends on the θ to $\theta_{chl}^{0}$ ratio of phytoplankton.'

l 684: "Two additional simulations considered an average value of 6.2% of CDOM in the leakage flux": writing "considered an average value" is not very precise, the reader needs to know exactly what is happening, for example: "Two additional simulations use a constant fraction of CDOM in the leakage flux, setting the CDOM fraction to 6.2%, which is the average value obtained from the Optimized simulation."
Thanks for the suggestion, this sentence will read: 'Two additional simulations use a constant fraction of CDOM in the leakage flux (*Constant leakage*, **Table 2**) and a constant fraction of CDOM in the total dpp flux (*Constant dpp*, **Table 2**), setting these CDOM fractions to 6.2% and 1.8%, respectively, which is the average values obtained from the *Optimized* simulation.'

Fig. 12: Panel (d) says "0-9m" but (e) is missing units: "40-60". It would also be useful to have this information in panels (b) and (c).
In Figure 12, we will add 0-9m to panel (b), DCM to panel (c) and 40-60m to panel (e).

l 799: "This is in better agreement with observations when the proportion of coloured material in the dpp depends on nutrient and light constraints": Add a reference to the model here, and change the "when" to "if", otherwise readers might think the "when" means "at times when" which would completely change the meaning of the sentence. Proposed new version: "This is in better agreement with observations if the proportion of coloured material in the dpp is modelled to depend on nutrient and light constraints"
Thanks for the suggestion, we will take it and use the proposed version.

l 809: Light limitation of what?
We will modify this sentence to: 'Light limitation to phytoplankton CDOM production acts through the elemental ratio of Chl:C relative to the maximum value…' For consistency we will also update the corresponding sentence for nutrient limitation in L801 to: 'Nutrient limitation to phytoplankton CDOM production acts through the relative proportion between leakage and *dpp*…'

l 836: "Adding optical components to BGC models greatly improves the comparability of simulated and observed types of variables": What exactly is meant here, it becomes easier to compare model and observations when the model has optical components?
No, we meant it increases the points of convergence 'simulated variable-observed variable'. We will rephrase as: 'Adding optical components to BGC models greatly increases the amount of data that can be used for model validation and calibration.'

---

## Author Response (AR1)

Dear Editor,

Thanks a lot for organizing the review of our manuscript. We have revised the manuscript according to the Reviewer's comments and our point-by-point responses to those comments.

Find below our point-by-point response to the reviews that includes a list of all relevant changes made in the manuscript. We would like to also state here a list of minor changes we have made to the manuscript that were not asked by the reviewers. These changes are marked in red font in the revised version of the manuscript, and the line numbers indicated refer to the revised version of the manuscript.

On behalf of all the authors,

Eva Álvarez

L12 We have updated the affiliation of one of the co-authors, V. Vellucci, and renumbered next institutions accordingly.

L378 We have updated the name of the Copernicus Marine Service to avoid the use of the acronym CMEMS, as indicated by the European Commission.

L417 Table 1, in the table header we have corrected the word 'biomass' for 'Chl-a', since PFTs where not evaluated in terms of biomass but in terms of Chl-a, both in this work and in the reference provided.

L528 We have changed the word 'about' for 'approximately'. The meaning of the sentence remains unaltered.

L594 We have changed the word 'in' for 'by'. The meaning of the sentence remains unaltered.

L690 Figure 11, we have clarified in the legend that in panel (a) the colors of the dots indicate seasons and the grey dots in the background are in situ observations.

L712 We have slightly reformulated the sentence: 'Figure 12a shows the percentage that CDOM flux represents of total DOC flux' to 'Figure 12a shows the percentage of CDOM flux with respect to total DOC flux'.

L869 We have added the word 'light' in '… insights into the dynamics of **light** absorbing BGC products such as CDOM.'

**RC1: 'Comment on bg-2023-48', Anonymous Referee #1, 03 Apr 2023**
Citation: https://doi.org/10.5194/bg-2023-48-RC1

The paper titled "Chromophoric dissolved organic matter dynamics revealed through the optimization of an optical-biogeochemical model in the NW Mediterranean Sea" presents a coupled 1D physical-biogeochemical-optical modelling suite inbedded in a parameter optimization tool. The models are run at the Boussole site in the Ligurian Sea. The large observational database allows tuning the model parameters (except a few parameters who could not be sufficiently constrained by the observed variables).

The authors show a convincing case. The paper is very clear and well written and the figures are relevant. The fully coupled approach is novative. The parameter optimization is very relevant and seems to improve all model variables. The authors also present important possible limitations of the method and justify the choices made (e.g. in section 4.1).

We have revised the manuscript according to our point-by-point responses to the Reviewer's comments. Following each comment, we detail the changes we have made to the manuscript which are marked in blue font in the revised version of the manuscript. The line numbers indicated in our responses refer to the revised version of the manuscript.

Therefor I have no major comments. I only would have been interested to see a couple of details explained:

(1) the authors state themselves that it remains a future objective to see how the optimized parameteres will behave in a 3D model. It is indeed well known that parameter optimization can be sensitive to particular configurations (see e.g. https://egusphere.copernicus.org/preprints/2023/egusphere-2023-363/#discussion for a very recent example). For example it can compensate poorly represented features by adapting other features as the authors also propose (line 570). Based on their experience, could the authors estimate if the model parameters would lead to realistic results in e.g. areas with more lateral contributions compared to the Boussole station ?  The switch from one model to another (GOTM to NEMO) may also be discussed (if relevant). However if no trials have been realized yet, I do not suggest that the authors need to speculate.

We have expanded the paragraph in the Discussion that covers the feasibility of applying the locally optimized parameters to 3D models, to cover those two aspects mentioned by the reviewer: the robustness of our results in areas with more lateral contributions compared to the BOUSSOLE station, and which can be the consequences of changing the transport model. The added text (**L822**) reads as follows:

'Since the approximation of a 1D configuration is reasonable at the BOUSSOLE site, we do not expect that our proposed parameter values are compensating for the absence of lateral transport. However, considering that biogeochemical optimization can be influenced by specific physical forcing and possible biases in circulation dynamics (Pasquier *et al*., 2023), further analysis is needed to upscale the present 1D results to the 3D Mediterranean

domain, where lateral contributions and vertical mixing are different and can interact differently with the optimized biogeochemical processes.'

(2) can the authors explain if there is any limit imposed on parameters ranges during the creation of new values by the genetic algorithm ? Positivity, statistical distribution, inter-relations or consistency between pairs of parameters, ... ?

We have rewritten Section 2.4.1 'Parameters optimized' to clarify earlier the conditions that we imposed to the parameter list to be optimized and the parameter ranges. The rewritten text (**L430**) reads as follows:

'The potential number of independent parameters included in the optimization problem is limited by the observations available and by the fact that optimized parameters must appear in some process independently. A total of 25 optical and BGC parameters were optimized (Table A1 in Appendix A), all of them were given a positive range of values for initialization and evolution, without conditions or trade-offs for the consistency between pairs of parameters. The rest of the parameters not included in the optimization kept constant values (Table 1).'

(3) the model-satellite comparison is realized over the 9 upper meters. Can the authors justify this choice e.g. in relation to observed optical depth during the year ? Are the model variables simply averaged over the layers corresponding to these 9 meters ?

We have included a new version of Figure 7 that averages the *Optimized* results (TChl-a, $a_{PH}(50)$ and $a_{DG}(450)$) over the first optical depth (**L597**) and have mentioned in the text the range of values the FOD takes throughout the year (**L586**).

(4) there is a type line 800, "parameterisation" (without "s")
Thanks, we have corrected the sentence (**L831**) to: "We have proposed a simplified parameterisation to account for the effects of light and nutrients on CDOM production."

**RC2: 'Comment on bg-2023-48', Anonymous Referee #2, 05 Apr 2023**
Citation: https://doi.org/10.5194/bg-2023-48-RC2

The authors couple a BGC-optical model with a 1-dimensional physical ocean model for a site in the NW Mediterranean Sea, with the goal of better resolving dynamics of CDOM. It is nice to see an emphasis on modeling CDOM in a coupled physical-BGC model. However, some work is required to improve the manuscript and make it more accessible to readers.

We have revised the manuscript according to our point-by-point responses to the Reviewer's comments. Following each comment, we detail the changes we have made to the manuscript which are marked in green font in the revised version of the manuscript. The line numbers indicated in our responses refer to the revised version of the manuscript.

**general comments**

I have a couple of methodological questions concerning the optimization procedure. Firstly, the authors declare the estimated 25 parameters "optimal" without too much evidence to back it up. Interdependence between parameters (which the authors mention in the manuscript) and other factors can lead to complex cost functions with local minima. Based on the manuscript and appendix A, it appears like the one large optimization experiment was conducted, using a large number of iterations. I would encourage performing a few more experiments starting from different initial parameter values, in order to assess the new parameter estimates are similar to the initial ones. Here it could be enough to perform fewer iterations and examine the convergence. If this type of experiment was already performed, this information should be included in the manuscript so that readers (like me) are not left wondering if the reported results are based on parameter values from a local minimum.

To support the resilience of the DE algorithm, in general, to fail into local minima, we added the following statement in Section 2.4.2 'Parameter optimization method' (**L479**):

'Genetic algorithms are particularly robust in overcoming local minima (Storn and Price, 1997). As a parallel search technique that runs several vectors simultaneously, DE sampling method can help vectors escape local minima and therefore avoid misconvergence.'

To support the robustness of our particular optimization experiment we have performed two additional replicates of the main optimization experiment. ParSAC creates the initial population from randomly-picked seeds and therefore each optimization replicate starts from a different distribution of parameter values. We have added an additional figure to the Appendix A (Figure A3, **L929**) where we show the results of the three replicates (the main one shown in the original version of the manuscript and the two replicates). We also added an additional paragraph to the text in Appendix A (**L904**) to explain these additional results that reads:

'Two replicates of the optimization experiment were run in order to examine the robustness of the optimization procedure. Each replicate use different randomly-generated seeds, and therefore starts from a different distribution of parameter values in the initial population. Despite slightly different optimization trajectories of the parameter values in the first 100 generations, the parameters that converged (excluding those of $P^{(DINO)}$ and $S_{X(1)}^{350-500}$) did it to values that were between 87-113% of the value reached in the main optimization experiment (**Fig. A3**)'

Secondly, the authors perform two experiments (named "Nutrients" and "Constant") in which certain parameters are held constant at some average values. Fig 12 shows some of the results, indicating that the constant values have a worse fit to data. However, one could argue that this is not an entirely fair comparison, as the constant values were not allowed to be optimized. That is, in how far would results and their interpretation change if the values of the constant parameters were adjusted, perhaps alongside all other parameters. I understand that the experiments were mainly conducted to examine the effect of underlying mechanisms that make these parameters non-constant, and I am not sure if this requires new experiments, but it would be nice to see at least some comment or discussion about the role of the constant value that was selected.

In order to clarify that the objective of the experiment is to highlight the vertical and temporal differences in CDOM production among formulations, we have included the following statement in Section 3.3.1 'In situ production of CDOM and synchronization with phytoplankton Chl-a' (**L709**):

'The constant % values in *Constant leakage* and *Constant dpp* were chosen to homogenise the values of $f_{R2}^{X2}$ at the beginning of the year between the formulations and to highlight the vertical and temporal differences in $f_{R2}^{X2}$ between the three formulations.'

Parts of the manuscript are well written and easy to follow, but in some parts I had to read sentences many times over to try to understand what is meant. Let me start with the notation: In the differential equations, many terms are abbreviated using a vertical line with a dx/dt part on the left and a 3-letter description and a variable on the right. This notation is never explained and, at first, I was under the impression the vertical line had a mathematical meaning (e.g., "evaluated at"). After going over equation 1 and accompanying text again, I noticed that this assumption was probably wrong. Instead, the variable on the bottom right of the vertical line may be the other variable this part of the equation is dependent on. I still don't exactly know, because this notation is never described in the manuscript. A couple of sentences would really help to make this notation accessible.

We have clarified the notation used by including new text in the first paragraph in Section 2.1.2 'The BGC model BFM and the bio-optical component' (**L186**):

'Following Vichi et al. (2007), the dynamical equations of this section are written in 'rate of change' form, where the right hand side contains the terms representing significant processes for each living or non-living component. For each process, the state variable subject to change is indicated before the vertical bar and the superscript and subscript after the bar are the 3-letter acronym of the process represented and the state variable involved in the process, respectively.'

Similarly, the variable notation is not explained properly, and the reader is left guessing or is required to find more information in other parts of the manuscript. A small example: There are 4 zooplankton, which are (oddly) denoted $Z^{(3)}$ to $Z^{(6)}$. Why is there no $Z^{(1)}$? What are the kind of zooplankton is represented by each variable? Why do only two of them appear in Eq 8 and 9? Some information can be obtained from Fig 1, but this information should be much more front and center, so that these questions do not even arise. Again, a few extra sentences would make the manuscript a lot more accessible. I mention a few more instances when I encountered text without enough explanation in my specific comments below.

We have modified the notation for the four zooplankton groups (**L170**) from numbers 6, 5, 4 and 3 to acronyms: HETN (heterotrophic nanoflagellates), MICRO (microzooplankton), OMNI (omnivorous mesozooplankton) and CARNI (carnivorous mesozooplankton), respectively. This hopefully clarify the kind of zooplankton each variable represents.

We have clarified that mesozooplankton (Z3=CARNI and Z4=OMNI) do not excrete dissolved carbon, only particulate, and that's why only microzooplankton groups a (Z5=MICRO and Z6=HETEN) appear in Eq. 8 and Eq. 9 (**L241**), as follows:

'Labile DOC ($R_C^{(1)}$) is produced by the excretion of microzooplankton ($Z^{(MICRO)}$) and nanoheterotrophs ($Z^{(HETEN)}$) (*exc*), and therefore represents the sources of DOC associated to the zooplankton-mediated mortality of phytoplankton and bacteria, and it is consumed quickly by bacteria. The released fraction of carbon by mesozooplankton excretion (both $Z^{(OMNI)}$ and $Z^{(CARNI)}$), on the other hand, is assumed to have no dissolved products and therefore directed to the state variable $R_C^{(DET)}$.'

Additionally, and in response to some of the specific comments, we have modified throughout the manuscript (**L169**) the superscripts of other state variables, as follows:
$P^{(1)}$ (diatoms) => $P^{(DIATOM)}$
$P^{(2)}$ (nano-flagellates) => $P^{(NANO)}$
$P^{(3)}$ (picophytoplankton) => $P^{(PICO)}$
$P^{(4)}$ (dinoflagellates) => $P^{(DINO)}$
$R^{(6)}$ (detritus) => $R^{(DET)}$
$O^{(3)}$ (dissolved inorganic carbon) => $O^{(DIC)}$
$N^{(1)}$ (phosphate) => $N^{(PO4)}$
$N^{(3)}$ (nitrate) => $N^{(NO3)}$
$N^{(4)}$ (ammonium) => $N^{(NH4)}$

The names of the simulations and experiments could be improved: The simulation names "Nutrients", "Constant" do not help the reader understand what is being modified or held constant. In comparison, The "Optimized" experiment has a better name. Then the experiment names EXP-X^{(2)} and EXP-X^{(3)} are maybe less helpful than the simulation names: What is the significance of "X" in the name, and why start counting at 2? Just using "Experiment 1" and "Experiment 2" would raise fewer questions, but a short descriptive name would be even better.

Thanks for the suggestion, we have changed the name of the simulations to give them short descriptive names. *Nutrients* and *Constant* are now *Constant leakage* and *Constant dpp*, respectively. For consistency *Bleaching* has been renamed *Intense bleaching* to indicate in which sense photobleaching rate is being modified:

Regarding experiments EXP-X(2) and EXP-X(3), we have renamed with shorter and descriptive names as EXP-Biology and EXP-Physics, respectively.

There were a two instances showing a clear break in the train of thought, and I would suggest reordering the text: Firstly, Section 2.4.1 "Parameter optimization method" briefly introduces the parameter optimization method and then goes into detail (often difficult to follow, see specific comments) about which parameters were chosen to be optimized, before returning to the optimization method. Here, I would suggest moving the part about the parameters into its own subsection.

This Section 2.4.1 has been split into two: 2.4.1 'Parameters optimized' (**L427**) that describes which parameters have been chosen to be optimized, and 2.4.2 'Parameter optimization method' (**L454**) that describes the DE algorithm.

Secondly, Section 2.4.2 briefly introduces the hypothesis testing experiments without motivating these experiments well. Pages later, in Section 3.3.1, additional context is provided that is very useful to better understand the setup of the experiments. I suggest merging information from the two texts, so that the parameter changes performed are easier to understand and well motivated from the start.

Section 2.4.2 is now Section 2.4.3 'Simulations and hypothesis testing experiments'. We have reintroduced the hypothesis testing experiments (**L489-L493**).

Also, in response to the specific comment l479 we have reformulated the description of each experiment, EXP-Biology in **L494** and EXP-Physics in **L500**.

I really like the figures, which look good and are informative. Here, I just have one general comment, I think it would be useful to consistently add description and units to their respective axis or the color bar. For example, in Fig 12, move the "a_CDOM(450)" and units in (d) and (e) to the y-axis, just like in panels (a-c). In Fig. 3 and similar, move property name and units to the color bar. This step would reduce text in the panels and avoid repeating units.

Thanks for the suggestions, we made the following changes to labels and axis:

Figures 3, 4, 5 and 6: We have added variable names to the color bar and removed repeated units. We hope these changes fulfill the requirement of the reviewer for clarifying

the figures. Also, for the sake of facilitating reading the figure, we have decided to leave the property name in the panel.

Figure 7: We moved the variable names to y axis.

Figure 12: In panels d and e, we moved labels to y axis; in panels b and c, we added '(0-9 m)' and 'DCM', respectively.

Figure 13: In panels b to e, we moved the variable name to y axis, and kept the depth intervals in the panels.

**specific comments**

l 18: "The absorption coefficient of CDOM, [...], is measurable in situ and remotely": Can it indeed be measured reliably remotely and distinguished from other (living) light absorbing constituents?
No, remotely CDOM cannot be distinguished from detritus, not at least in current products. We rewrote this part of the abstract (**L21**): 'The absorption coefficient of CDOM, $a_{CDOM}(\lambda)$, is measurable in situ and can be retrieved remotely, although ocean colour algorithms do not distinguish it from the absorption of detritus. These observations can be used as indicators for the concentration of other relevant biogeochemical variables in the ocean, e.g. dissolved organic carbon.'
Similarly, we also rephrased the second paragraph of the introduction (**L55**) to: '$a_{CDOM}(\lambda)$ can be measured in situ at selected locations and be retrieved on a global scale from remote-sensing platforms, although remotely it is not distinguishable from the absorption of detritus.'

l 40: "the spatio-temporal dynamics [...] reflects the functioning of the carbon cycle": What exactly is meant here, I would suggest rephrasing.
We rewrote this sentence (**L39**) to: 'Furthermore, as the largest pool of reduced carbon in the oceans, it plays an important role in the global carbon cycle (Legendre et al., 2015).'

l 41: "non-absorbing": I would suggest changing this to "not light absorbing".
Thanks for the suggestion, we have rewritten this sentence (**L40**) to: 'Although most of the dissolved exudates that form the DOM are non light absorbing (Mühlenbruch et al., 2018), a fraction of DOM absorbs light mainly in the ultraviolet (UV) and blue spectral range of the electromagnetic radiation.'

l 50: "There are several possible factors determining...": "Besides the presence of CDOM, there are other possible factors influencing..."
Thanks for the suggestion, we have rewritten this sentence (**L50**) as: 'Besides the presence of CDOM, there are other possible factors influencing this optical behavior, including: the particular pigment ratios in the Mediterranean phytoplankton community (Organelli et al.,

2011), the abundance of small coccolithophores (Gitelson et al., 1996), and the influence of Saharan dust (Claustre et al., 2002).'

l 63: Since vertical transport is mentioned here, why not horizontal as well? Maybe change "terrestrial and atmospheric inputs" to something like "terrestrial and atmospheric CDOM sources and physical transport"?
Thanks for the suggestion, we rewrote this sentence (**L62**) as: 'CDOM cycling is essentially controlled by in situ biological production (Romera-Castillo et al., 2010), **terrestrial and atmospheric sources and physical transport**, microbial consumption (Nelson and Gauglitz, 2016; Legendre et al., 2015; Stedmon and Markager, 2005), as well as deep ocean circulation and/or vertical mixing (Coble, 2007), and it is photoreactive and efficiently destroyed in the upper layers of the water column by solar radiation (Mopper and Kieber, 2000).'

l 73: Is the NW Mediterranean Sea the study region?
No, in this paragraph we were referring to the higher-than-average DOM deposition fluxes reported for the whole Mediterranean Sea, thanks for pointing out. We changed this sentence (**L74**) to: '**In the basin**, the higher-than-average CDOM concentrations seem to be sustained by allochthonous sources, ...'

l 75: "depositions from the atmosphere are 2–5 times larger in the Mediterranean Sea than in the oceans": higher than "in the oceans" is not very specific: higher than anywhere in the ocean, higher than some average?
We do agree, we have rewritten this sentence (**L76**) as: '... as fluxes of DOM depositions from the atmosphere are 2–5 times larger in the Mediterranean Sea than **those estimated for the global ocean**, which explains the abundance of humic-like substances (Santinelli, 2015; Galletti et al., 2019).'

l 85: What does "probably controlled" mean? Perhaps change to "mostly controlled" or "its dynamics are dominated by"?

and

l 86: One can maybe make the counterpoint that photoacclimation and nutrient limitation are also often controlled by vertical transport. Maybe distinguish between being controlled by and directly caused by, as the preceding sentence suggests?
We have reformulated this sentence (**L86**) as: 'While $a_{CDOM}(\lambda)$ dynamics at the surface is mainly explained  by vertical transport of CDOM from depth and by photochemical degradation, Chl-a concentration at the surface is mainly driven by photoacclimation of phytoplankton and nutrient limitation.'

l 92: "observations in the NW Mediterranean Sea suggest direct CDOM production by phytoplankton": Is this then equivalent to the way it is handled in other models, i.e. the fixed fraction of the dpp of phytoplankton?

This is equivalent to a fraction of the dpp of phytoplankton, not necessarily a fixed fraction. We haven't make any change related to this comment.

l 169: Explain the symbols and superscripts used here ("O^{(3)}" etc.).

We have changed the numbers on the superscripts for more self-explanatory names. The text (**L168**) now reads as follows:

'The implementation of the BFM in FABM comprises 54 state variables. These include representations of dissolved inorganic carbon ($O^{(DIC)}$), inorganic forms of nitrogen ($N^{(NO3)}$), ammonium ($N^{(NH4)}$) phosphorus ($N^{(PO4)}$) and silicate ($N^{(Si)}$), four phytoplankton types ($P^{(DIATOM)}$, $P^{(NANO)}$, $P^{(PICO)}$ and $P^{(DINO)}$), heterotrophic bacteria (B) and four grazers ($Z^{(HETN)}$, $Z^{(MICRO)}$, $Z^{(OMNI)}$ and $Z^{(CARNI)}$), three pools of dissolved organic matter ($R^{(1)}$ to $R^{(3)}$) and CDOM ($X^{(1)}$ to $X^{(3)}$) differentiated by reactivity, and particulate organic matter ($R^{(DET)}$).'

l 171: "A subscript appended to each module ...": This sentence requires more explanation. What are the modules? Is Chl-a considered an elemental constituent? I think I understand the rest of this paragraph, but this sentence is difficult to follow and seemingly unrelated to the preceding and following sentences.

We have clarified (**L172**) as: 'The subscript appended to each living and non-living component indicates the elemental constituents among carbon (C), nitrogen (N), phosphorus (P) and silica (Si, only in $P^{(DIATOM)}$), and the content in Chl-a only in PFTs.' This has been changed also in the caption for Figure 1 (**L267**), as: 'The subscript *i* appended to each living component and detritus indicates the elemental constituents among carbon (C), nitrogen (N) and phosphorus (P), Chl-a only for phytoplankton and silica (Si) only for diatoms'.

l 184: What is "OAC"?

Generally, OAC accounts for 'optically active constituents'. In this work, we refer to those as optical constituents and therefore the use of OAC was a typo. We rewrote this sentence (**L185**) as: '... the optical treatment of constituents (PFTs, CDOM and detritus) …'

Eq 1: "; i = 1 to 4" -> " for i = 1,2,3,4"

Phytoplankton types are now enumerated with their short name and therefore Eq. 1 shows i = DIATOM, NANO, PICO, DINO (**L194**). We have included i = 1,2,3 in Eq. 13 (**L257**).

Eq 1: Why does zooplankton indexing start at 3?

We have modified the notation for the four zooplankton groups (**L170**) from numbers 6, 5, 4 and 3 to acronyms: HETN (heterotrophic nanoflagellates), MICRO (microzooplankton), OMNI (omnivorous mesozooplankton) and CARNI (carnivorous mesozooplankton),

respectively. The counter j takes now those short names as values, that have been modified in Equations 1, 8 and 9.

l 191: As an aside, I find the "prd" abbreviation of "predation" slightly confusing, as I always think of "production" first. I would suggest using "grz" for "grazing" instead. 'prd' is the 3-letter acronym that BFM uses conventionally to refer to predation. We substituted 'prd' by 'grz' for clarity (Eq. 1 and **L195**).

Eq 2: Underneath, it reads like both temperature and nutrients are dimensionless, when really these temperature- and nutrient-dependent factors are dimensionless. I would suggest changing "ft_P" to "ft_P(T)" where T is model temperature. The same applies to fn_P, which has the benefit of telling the reader which nutrients are entering this equation and control GPP. Currently, it might appear to a reader as if GPP is independent of other model variables.
We have clarified the formulation of the temperature- and nutrient-dependent factors, including the dependencies on temperature and growth-limiting factors, respectively (Eq. 2, **L197**). We also have stated in the text which are the growth-limiting nutrients for each phytoplankton type included in the model (**L198**).

l 195, 197: Sometimes "mg Chl", sometimes "mg Chla" is used. To stay consistent with previous notation, I would suggest using "mg Chl-a" and "(mg Chl-a)^{-1}" when in the denominator.
Thanks for pointing this out. We have corrected Chl to Chl-a consistently throughout the text and figures, including when it is mentioned in the Chl-a:C ratio.

Eq 4: I presume, P_P is the phosphorous quota (subscript P) of phytoplankton (regular P). Is the superscript denoting the phytoplankton type missing, or am I misunderstanding? Similarly, is the "P_N^{(k)}" meant to be "P_{N^{(k)}}"? That is, the superscript "(k)" appears to be attached to the P, but should it be attached to the subscript N instead?
Thanks for spotting this. The superscript (i) denoting the phytoplankton type was missing in the P quota and inserted incorrectly as (k) in the N quota. This Equation has been corrected (**L221**) as:

$$G_P^{balance} = min\left( G_P, \frac{1}{p_P^{min}} \times \frac{dP_P^{(i)}}{dt}\Big|\frac{upt}{N^{(PO4)}}, \frac{1}{n_P^{min}} \times \sum_{k=NO3,NH4} \frac{dP_N^{(i)}}{dt}\Big|\frac{upt}{N^{(k)}} \right)$$

where $p_P^{min}$ and $n_P^{min}$ are the minimum phosphorous and nitrogen quota, respectively, and the two nitrogen sources are $N^{(NO3)}$ and $N^{(NH4)}$.

Eq 5: What is i set to here, is there perhaps a sum missing?
Thanks for spotting this, (i) is included because Eq. 5 was indicating how $f_{R2}^{X2}$ is computed for each phytoplankton type, and not the coloured fraction of the total DOC of

phytoplankton origin as the text said. In the revised version, we keep referring to the coloured fraction of the total DOC of phytoplankton origin as $f_{R2}^{X2}$ (that is shown in Figure 1 and Figure 12), and use instead $f_{P(i)}^{X2}$ to refer to the coloured fraction of the DOC produced by phytoplankton type I (**L232**). Eq. 5 has been rewritten accordingly:

$$f_{P(i)}^{X2} = f_{P(i)}^{maxX2} \times \frac{\theta^{(i)}}{\theta_{chl}^{0(i)}} \times \frac{\beta_P^{(i)} \times \frac{dP_C^{(i)}}{dt}|\frac{gpp}{O^{(DIC)}}}{\frac{dP_C^{(i)}}{dt}|\frac{dpp}{R_C^{(2)}}}$$

The maximum fraction of coloured DOC in each phytoplankton type that is prescribed as a fixed parameter has been renamed to $f_{P(i)}^{maxX2}$. Table 2, Table A1 and Figure A2 have been updated with the new name of the parameter.

Accordingly, equations 6 (**L236**) an 7 (**L237**) have change to use the fraction $f_{P(i)}^{X2}$ of each phytoplankton type inside the summation:

$$\frac{dR_C^{(2)}}{dt} = \sum_{i=1,2,3,4} \left( \frac{dP_C^{(i)}}{dt}|\frac{dpp}{R_C^{(2)}} \times \left(1 - f_{P(i)}^{X2}\right) \right) - \frac{dB_C}{dt}|\frac{upt}{R_C^{(2)}}$$

$$dX_C^{(2)}/dt = \sum_{i=1,2,3,4} \left( \frac{dP_C^{(i)}}{dt}|\frac{dpp}{R_C^{(2)}} \times f_{P(i)}^{X2} \right) - \frac{dB_C}{dt}|\frac{upt}{R_C^{(2)}} - \frac{dX_C^{(2)}}{dt}|\frac{deg}{R_C^{(3)}}$$

l 231: Why do the other zooplankton not excrete labile DOC? Are they too small/large? This information is missing from the manuscript.
Line 232 stated that 'Large zooplankton excretion … is composed by particulate only and directed to the state variable $R(6)$'. That's because the excretion of Z3 and Z4 is supposed to represent the production of fecal pellets and egestion (sloppy feeding).
In the revised manuscript, we have clarified (**L241**) as follows: 'The released fraction of carbon by mesozooplankton excretion (both $Z^{(OMNI)}$ and $Z^{(CARNI)}$), on the other hand, is assumed to have no dissolved products and therefore directed to the state variable $R_C^{(DET)}$.'

Eq 14-16: Out of curiosity, why are different reference wavelength used in the 3 equations (450 nm, 440 nm and 550 nm)?
Because the reference wavelength is among the parameters obtained by fitting non-linear models to observed $\alpha_{CDOM}(\lambda)$, $\alpha_{NAP}(\lambda)$ and $b_{NAP}(\lambda)$ spectra. In the basic formula for the specific absorption coefficient:
a=c2 x exp(-S(λ-λref))
c2, S and λref are all coefficients obtained when fitting the non-linear model to observed spectra (of both $\alpha_{CDOM}(\lambda)$ and $\alpha_{NAP}(\lambda)$). For the absorption of CDOM, c2 scales the absorption of CDOM at a reference wavelength (λref) of 450 nm (taken from Dutkiewicz et al. 2015, originally from the parameters fitted by Kitidis et al. 2006). For the absorption of NAP, c2 scales the absorption of NAP at a λref of 440 nm (taken from Gallegos et al. 2011, originally from the parameters fitted by Bowers and Binding 2006).
The same happens in the basic formula for the scattering coefficient:

b=c2 x (λref/λ)^e
where c2, e and λref are all found when fitting the non-linear model to observed spectra. In this case, c2 is the mass-specific scattering coefficient at a λref of 555 nm (originally from parameters fitted by Babin et al., 2003 to phytoplankton scattering spectra, Gallegos et al. 2011 used the same λref and adapted c2 to represent the scattering of several types of non-algal particles).
We did not make any modification to the manuscript in response to this comment.

l 356: "()" are missing for a_PH and a_DG.
We corrected this part (**L366**) as: IOPs ($a$PH(443) and $a$DG(443)) are derived …'

Fig. 2: x- and y-axis labels are missing. Consider coloring land gray in the top left map.
Thanks for the suggestion. We added '°E' and '°N' to the x- and y-axis respectively and filled land grey in the small map of Figure 2 (**L376**).

l 378: "assumed equal": Why "assumed"?
We have removed 'assumed' and left the sentence (**L387**) as follows: 'The parameter values related to the partition between dissolved and particulate excretion in $Z^{(MICRO)}$ and $Z^{(HETEN)}$, and the mortality of $Z^{(CARNI)}$ and $Z^{(OMNI)}$, are equal to the up-to-date values derived in Álvarez et al. (2022), together with all the optical parameters.'

l 389: "and restored at monthly frequency": Does this imply nudging/relaxation?
Yes, it does. We have modified this sentence (**L399**) as follows to clarify: 'Observed vertical profiles of T and S … were … used as initial conditions **and constrained by a monthly nudging**, in order to reproduce the intensity and timing of the mixing as closely as possible to observations.' For consistency, the relaxation to vertical profiles of nutrients has been clarified (**L402**) as: '… used for initialization and constrained by a yearly nudging.'

l 419: I would suggest removing the "while" here.
This 'while' has been removed as a result of the reorganization/reformulation of Section 2.4.1 'Parameters optimized' and Section 2.4.2 'Parameter optimization method' suggested in a previous comment.

l 425: What does "assumed true" mean, assumed that the true value is known and not included in the optimization? I would prefer "assumed known" or "assumed constant". Please make it explicit that this implies that the parameter in question is not included in the optimization. This needs to be made clear initially, the phrase is used a lot.
These parameters have not been included in the optimization. We have clarified at the beginning of Section 2.4.1 'Parameters optimized' that all parameters not included in the optimization are assumed constant (**L434**). We have also reformulated slightly this whole section in order not to repeat as much 'assumed constant'.

l 426: Subscript "(5 to 6)" and "(1 to 4)" are unconventional and could be confusing some readers. I suggest writing them out fully, i.e. (I am not including superscripts here): f_Z(5), f_Z(6), f_P(1), ..., f_P(4), and f_B.

This sentence has been changed (**L436**) to: 'We assumed constant the absorption coefficients of the three reactivities at 450 nm ($a_{X(1)}^{450}$, $a_{X(2)}^{450}$ and $a_{X(3)}^{450}$) and optimized parameters related to the production of CDOM ($f_{Z(MICRO)}^{X1}$, $f_{Z(HETEN)}^{X1}$, $f_{P(DIATOM)}^{X2}$, $f_{P(NANO)}^{X2}$, $f_{P(PICO)}^{X2}$, $f_{P(DINO)}^{X2}$ and $f_{B}^{X3}$) and those related to the photobleaching of $X_C^{(3)}$ ($b_{X(3)}$ and $I_{X(3)}$).'

l 432: "All optimized parameters appear in some process independently:": As a reader, I was now expecting a brief review of where the parameters mentioned in the previous sentences appear in equations. Instead, new parameters are mentioned and added to the list, of which two are not actually appearing in some process independently. Does the "all parameters" include the previously named ones? It would be helpful to break up this long sentence into two, don't use "all parameters" (one could use "the following" or similar).

Thanks for pointing out, we agree that the parameter appearing in an independent process is a condition that all parameters included in the optimization must meet, and in fact is met in our study. We clarified this consideration at the beginning of the section 2.4.1 'Parameters optimized' (**L430**), as it applies to all parameters. We have avoided the use of "all parameters" when referring to phytoplankton parameters.

l 435: "We assumed true a_PS and optimized \phi_C, because these parameters are not well documented in literature being more difficult to measure.": Does this mean \phi_C was included in the optimization and a_PS was not, because they both appear in the same equation, and they may be dependent on each other?

Yes, the absorption cross-sections of photosynthetic pigments (a_PS) and the photochemical efficiency (phi_C) only appear multiplied in the photosynthesis equation. Their product is the equivalent to the so-called initial slope of the production-irradiace (PE) curve (alpha), used in most of the non-spectrally resolved formulations of the PE relationship. Regarding phytoplankton parameters, we have clarified that we included in the optimization the two parameters that appeared in two different processes (the absorption cross section and the maximum Chl-a:C quotum), and chose one of the two that appear together in photosynthesis (the absorption cross-sections of photosynthetic pigments and the photochemical efficiency) (**L443**).

l 441: "A total of 25 optical and BGC parameters were optimized": I think it would be helpful to mention this at the beginning of the paragraph.

We have moved this sentence to the beginning of Section 2.4.1 'Parameters optimized' (**L432**).

After the modifications stated in comments from l425 to l441, the rewritten Section 2.4.1 (**L427**) reads as follows:

'Observations used for optimization were all collected at the BOUSSOLE site at monthly temporal resolution and roughly at the same number of discrete depths and included pico-, nano- and micro-phytoplankton Chl-a (pico Chl-a, nano Chl-a and micro Chl-a, respectively), $a_{PH}(450)$, $a_{NAP}(450)$, $a_{CDOM}(450)$ and $S_{CDOM}^{350-500}$. The potential number of independent parameters included in the optimization problem is limited by the observations available and by the fact that optimized parameters must appear in some process independently. A total of 25 optical and BGC parameters were optimized (**Table A1** in Appendix A), all of them were given a positive range of values for initialization and evolution, without conditions or trade-off for the consistency between pairs of parameters. The rest of the parameters not included in the optimization kept constant values (**Table 1**). The only observations related to CDOM were $a_{CDOM}(\lambda)$ that depend both on the mass-concentration and on the mass-specific absorption coefficients across the electromagnetic spectrum. We assumed constant the absorption coefficients of the three reactivities at 450 nm ($a_{X1}^{450}$, $a_{X2}^{450}$ and $a_{X3}^{450}$) and optimized parameters related to the production of CDOM ($f_{Z(MICRO)}^{X1}$, $f_{Z(HETEN)}^{X1}$, $f_{P(DIATOM)}^{X2}$, $f_{P(NANO)}^{X2}$, $f_{P(PICO)}^{X2}$, $f_{P(DINO)}^{X2}$ and $f_{B}^{X3}$) and those related to the photobleaching of $X_{C}^{(3)}$ ($b_{X(3)}$ and $I_{X(3)}$). The spectral slope of the CDOM pool has been associated with aromaticity and average molecular weight of the CDOM compounds (Blough and Green, 1995) but appears to be less linked to CDOM concentration. Therefore, the spectral slopes between 350 and 500 nm for the three reactivities ($S_{X(1)}^{350-500}$, $S_{X(2)}^{350-500}$ and $S_{X(3)}^{350-500}$) were optimized to ensure the proper simulation of $a_{CDOM}(\lambda)$ in wavebands other than 450 nm. For phytoplankton, independent observations regarding mass-concentrations (Chl-a) and optical properties ($a_{PH}(\lambda)$) were available, therefore both types of parameters were optimized. We optimized the absorption cross-sections ($\bar{a}_{PH}^{*}$) and the maximum Chl-a:C quota ($\theta_{chl}^{0}$) as they appear independently in light attenuation (Eq. (17)) and in photoacclimation (Eq. (19) in Álvarez et al., 2022) and in CDOM production (Eq. (5)), respectively. The absorption cross-sections of photosynthetic pigments ($\bar{a}_{PS}^{*}$) and the photochemical efficiencies ($\phi_{c}^{0}$) of the PFTs appear only in photosynthesis (Eq. (2)). We kept constant $\bar{a}_{PS}^{*}$ and optimized $\phi_{c}^{0}$, because these parameters are not well documented in literature being more difficult to measure. For detritus, we optimized the reference absorption coefficients at 440 nm ($a_{R}^{440}$) and kept constant all BGC parameters that alter the mass-concentrations. Given that $a_{NAP}(\lambda)$ observations were available, optical parameters related to the spectral shape of absorption by detritus ($S_{R}^{350-500}$) could have been included in the optimization, but we decided to maintain this parameter constant given the small contribution of $a_{NAP}(\lambda)$ to the total non-water absorption as compared to CDOM and phytoplankton. All parameters related to scattering and backscattering both of phytoplankton ($\bar{b}_{PH}^{*}$ and $bbr_{PH}$) and detritus ($b_{R}^{550}$, $e_{R}$ and $bbr_{R}$) were not included in the optimization.'

l 449: "O_1, ..., O_n, O_N": This notation is confusing, why include n and N here, is n=N-1? The same applies to P.
We reformulated this notation. The sentence (**L462**) now reads: 'For any variable observed m from a total of M, there will be N number of pairs consisting of the simulation $P_n$ and the corresponding observation $O_n$.'

Eq 27: Similar confusion, I assume n iterates from 1 to N here. So it should be "n = 1, ...,
N" or (more conventional) "n=1" and the "N" is placed on top of the sum. The same applies
to equation 29.
We have corrected the Σ on both equation 27 and Eq. 29 (now **Eq. 28**) to place "n=1" and
"m=1", respectively, on the bottom of the sum and "N" and "M", respectively, on top of the
sum.

Eq 29: So N_m is the number of observations for variable m? This needs to be made
explicit, and I suggest introducing this notation right away in Eq 27, so that it is clear what
SSQ_m is. I probably do not understand the terms here and what the subscript m
indicates, it appears like SSQ_m/(2*\sigma_m) = SSQ_m/(2*SSQ_m/(N_m-1)) = (N_m-1)/2?
This is very confusing.
We have clarified that m refers to each variable observed of a total of M, and n refers to
each pair observation-prediction from a total of N. In the corrected notation, the number of
observations of a given variable m is denoted as $N_m$ and the sum of the squares as $SSQ_m$.
Regarding the formulation of the lnLikelihood, in the previous version of the manuscript we
gave the general formulation for the lnLikelihood as it is computed by ParSAC. This
formulation can use a standard deviation for the observations that is manually provided by
the user, or, if the sd is not specified (this work), ParSAC estimates it from the model-
observation differences (Eq. 28). In the first case, the $SSQ/2sd^2$ terms (sd-dependent) in
the lnLikelihood (Eq. 29) are not constant, and can therefore not be dropped. We agree
with the reviewer that in our particular application, where we do not provide a known sd for
each variable and compute them instead from the SSQ (second case), the last term in the
lnLikelihood can be dropped. Therefore, for the sake of clarity, we have simplified Eq. 27,
28 and 29, to just two equations (**L459**) as follows:

'ParSAC formulates such fitness (i.e. probability that the candidate parameter values are the
true parameter set representing reality) as a multi-objective function calculated as the log-
transformed likelihood between the outcome of the model and the observations provided,
assuming the observed values are log-normally distributed, with the median equal to the
model prediction. For any variable observed m from a total of M, there will be N number of
pairs consisting of the simulation $P_n$ and the corresponding observation $O_n$. The sum of the
squares of the residuals for each variable observed m ($SSQ_m$) is computed as:

$$SSQ_m = \sum_{n=1}^{N_m} [P_n - O_n]^2$$

For M variables observed, all differences between model and observations are combined
as:

$$lnLikelihood = \sum_{m=1}^{M} -N_m \times ln(SSQ_m)$$

'

l 460: The DE algorithm is not explained very well. For example, it is not clear from the text
which of the symbols denotes the target vector and which is the mutant vector. What does
"crossing" with the mutant vector mean? In "All target vectors in the first generation are

crossed with the mutant vector", it sounds like the same mutant vector is applied to all target vectors. If I did not know how DE worked, I would not have a better idea after reading this section. This description needs to be rewritten.
We reformulated this section (**L466**) as follows:

'Each individual in a given population of size S=288 is a 25-dimensional target vector that represents a candidate solution to the problem. The parameter values in the target vectors in the first generation are initialized randomly across the parameter space. The parameter values were sampled uniformly between minimum and maximum values, which are listed in **Table A1** (Appendix A). The DE algorithm creates new generations of individuals by applying cycles of mutation, crossover and selection operating on the target vectors. Mutation: once all simulations in a generation are completed, the DE algorithm creates a new set of S mutant vectors. Each mutant vector is created selecting randomly three targets ($x_{r1}$, $x_{r2}$ and $x_{r3}$) and applying a mutation operator that consisted on $x_{r1} + 0.5 \cdot (x_{r2} - x_{r3})$. Crossover: to increase the diversity of the parameter vectors, crossover is done to generate the final trial set of individuals. Crossover is performed between each target vector and its corresponding mutant vector. retaining the target vector in the population with a probability of 0.9, otherwise introducing the mutation only with a 0.1 probability. Selection: the final phase is selection where the lnLikelihood of the trial vector is compared to the lnLikelihood of the corresponding initial target. The trial vector replaces the target if the lnLikelihood value obtained from the trial is higher or equal than the lnLikelihood obtained from the target vector. After the selection procedure, the size of the population is again S. The cycle of mutation, crossover and selection is carried out iteratively and eventually, the DE algorithm provides an estimate of the optimal parameter values that minimizes the misfit between the model output and the observations (i.e. maximizes the sum of lnLikelihoods in the population). '

l 466: "optimal set of 25-parameter values": The DE algorithm aims to find optimal parameter values, but that does not mean that the resulting parameters are indeed optimal in that they maximize the likelihood. Hopefully, they are close to the global optimum. In other words, I would suggest using "estimate of the optimal parameter values".
We have taken the reviewer suggestion and this sentence (**L478**) now reads: 'The cycle of mutation, crossover and selection is carried out iteratively and eventually, the DE algorithm provides **an estimate of the optimal parameter values** that minimizes the misfit between the model output and the observations (i.e. maximizes the sum of likelihoods in the population).'

l 476: "reversed some of the assumptions of the optimized configuration": What does "reversed" mean here? I think "changed" might be more suitable.
We used 'changed' instead of 'reversed' (**L492**). Additionally, this first part of the Section 2.4.3 'Simulations and hypothesis testing experiments' has been rewritten in response to one of the general comments.

l 479: "The comparison of the results of Optimized with Nutrients and Constant constituted and experiment..." This sentence is difficult to follow, I suggest rephrasing. I had to read the previous sentences several times as well: one uses "removed the dependence >>of<< f on \theta/\theta_chl", the other "removed both dependencies >>from<< f, on \theta/\theta_chl", making it very difficult to understand what is dependent on what, and what is being removed. Here, I strongly suggest stating the aim of each experiment first and then go into details of the implementation.

Section 2.4.3 'Simulations and hypothesis testing experiments' has been rewritten in response to one of the general comments. The two experiments will be described separately (**L494** and **L500**) as follows:

'In EXP-Biology, we investigated the role of nutrient and light limitations on controlling the biogenic in situ production of biodegradable CDOM and therefore on generating the observed $a_{CDOM}(\lambda)$ values. This experiment consisted of comparing the results of *Optimized* with two additional simulations. In *Constant leakage*, we considered the proportion $X_C^{(2)}: R_C^{(2)}$ in *dpp* as a constant fraction of leakage and therefore removed the dependence of $f_{R2}^{X2}$ on θ/chl0. In *Constant dpp*, we considered the fraction $X_C^{(2)}: R_C^{(2)}$ in *dpp* constant and therefore removed both dependencies from $f_{R2}^{X2}$, on θ/chl0 and on leakage/*dpp*.

In EXP-Physics, we investigated the role of allochthonous CDOM in maintaining CDOM concentrations in the surface ocean. This experiment consisted of comparing the results of *Optimized* with an additional simulation, *Intense bleaching*, where we prescribed the same photobleaching parameters for $X_C^{(3)}$ ($b_{X(3)}$ and $I_{X(3)}$) as for the other two CDOM pools and quadrupled the CDOM:DOC fractions in all biotic sources ($f_{Z(MICRO)}^{X1}$, $f_{Z(HETEN)}^{X1}$, $f_{P(DIATOM)}^{X2}$, $f_{P(NANO)}^{X2}$, $f_{P(PICO)}^{X2}$, $f_{P(DINO)}^{X2}$ and $f_B^{X3}$). By comparing *Optimized* with *Intense bleaching* we evaluated whether an increased locally produced CDOM could compensate a reduced allochthonous input and the extent to which the *Optimized* simulation was able to represent the role of $X_C^{(3)}$, and thus of any source of allochthonous CDOM, in generating the observed $a_{CDOM}(\lambda)$ values.'

Fig 3: Mention that these are climatological values in the caption.

The caption has been modified (**L531**) to:

'Figure 3. **Annually-averaged** vertical profiles as a function of time of (a-b) temperature, (c-d) nitrate and (e-f) phosphate, obtained from observations at the DYFAMED site (left panels) and from the *Optimized* simulation (right panels). The white line in the temperature panels represents the mixed layer depth (mld) obtained in (a) with a threshold of 0.2 °C on temperature (de Boyer Montégut et al., 2004) and in (b) with a threshold of $1 \cdot 10^{-5}$ m$^2$ s$^{-2}$ on turbulent kinetic energy. The line on (c) and (d) represents the nitracline (nitrate concentration equals 2 µM).'

The same applies to Figures 4 (**L548**), 5 (**L560**) and 6 (**L580**).

Fig 3: Units in the color bar and in the panel are not identical (they are equivalent, but why create confusion).

We removed the units from the panels and leave the units only in the color bar, both in the nutrient panels of Figure 3 (**L532**) and the DOC panel of Figure 5 (**L560**).

l 568: Out of curiosity, what are the correlations between in situ and satellite observations?
Thanks for the question, we will evaluate the task of intercalibration exercise between observing systems in future works. For a detailed comparison between remotely and in-situ collected observations at the Boussole site, we suggest figures 40 (Chl-a, page 52) and 52 (CDOM, page 65) of the following report:
'Climatological characterization of Ocean Sites for OC-SVC-BOUSSOLE-D4'
https://www.eumetsat.int/media/49886
We haven't done any change to the manuscript in response to this comment.

Fig 7: What happened in the model in early 2013 when there appears to be a drop of model chlorophyll to nearly zero?
It is the effect of the intensive winter mixing that homogenizes concentrations quite deep in the water column and takes chlorophyll almost to zero. It also happens in early 2012, although not so intensively. These periods have not been included in the metrics computation and we have stated in the modified text (**L589**) that TChl-a concentration drops almost to zero.

Fig 8 and 9: it might be helpful to change the order of legend entries to match the extra axes.
In Figure 8 (**L622**), we have reordered the legend elements in order (top to bottom): Bc, aCDOM, TChl-a, DOC. In Figure 9 (**L640**), the order of the legend elements matches the extra y-axis from left to right: DOC, TChl-a, aCDOM, Bc.

Fig 11b: Why are the ranges of reported values so patchy?
Because these are areas outlining clouds of points. We show below the original figures from Catalá et al. (2018) and Galletti et al (2019) with the ranges that we included in the background of Figure 11b (grey areas). We haven't make any modification to the manuscript in response to this comment.

[Figure]

l 683: "1) depends": Missing "it".

We have corrected this sentence (**L705**) as follows: 'Therefore, the fraction $X_C^{(2)} : R_C^{(2)}$ in the *Optimized* simulation was variable in time and depth for two reasons: 1) **it** depends on the relative proportions of DOC from leakage or exudation and 2) **it** depends on the θ to $\theta_{chl}^0$ ratio of phytoplankton.'

l 684: "Two additional simulations considered an average value of 6.2% of CDOM in the leakage flux": writing "considered an average value" is not very precise, the reader needs to know exactly what is happening, for example: "Two additional simulations use a constant fraction of CDOM in the leakage flux, setting the CDOM fraction to 6.2%, which is the average value obtained from the Optimized simulation."

Thanks for the suggestion, this sentence (**L707**) now reads:

'Two additional simulations use a constant fraction of CDOM in the leakage flux (*Constant leakage*, Table 2) and a constant fraction of CDOM in the total dpp flux (*Constant dpp*, Table 2), setting these CDOM fractions to 6.2% and 1.8%, respectively, which are the average values obtained from the *Optimized* simulation. The constant % values in *Constant leakage* and *Constant dpp* were chosen to homogenise the values of $f_{R2}^{X2}$ at the beginning of the year between the formulations and to highlight the vertical and temporal differences in $f_{R2}^{X2}$ between the three formulations.'

Fig. 12: Panel (d) says "0-9m" but (e) is missing units: "40-60". It would also be useful to have this information in panels (b) and (c).

In Figure 12 (**L729**), we have added 0-9m to panel (b), DCM to panel (c) and 40-60m to panel (e).

l 799: "This is in better agreement with observations when the proportion of coloured material in the dpp depends on nutrient and light constraints": Add a reference to the model here, and change the "when" to "if", otherwise readers might think the "when" means "at times when" which would completely change the meaning of the sentence. Proposed new version: "This is in better agreement with observations if the proportion of coloured material in the dpp is modelled to depend on nutrient and light constraints"

Thanks for the suggestion, we took it and now the sentence (**L840**) reads:

'This is in better agreement with observations if the proportion of coloured material in the dpp is modelled to depend on nutrient and light constraints (Fig. 12).'

l 809: Light limitation of what?

We have modified this sentence (**L840**) to: 'Light limitation to phytoplankton CDOM production acts through the elemental ratio of Chl:C relative to the maximum value…' For consistency, we have also updated the corresponding sentence for nutrient limitation (**L832**) to: 'Nutrient limitation to phytoplankton CDOM production acts through the relative proportion between leakage and *dpp*…'

l 836: "Adding optical components to BGC models greatly improves the comparability of simulated and observed types of variables": What exactly is meant here, it becomes easier to compare model and observations when the model has optical components?
No, we meant it increases the points of convergence 'simulated variable-observed variable'. We rephrased (**L867**) as: 'Adding optical components to BGC models greatly increases the amount of data that can be used for model validation and calibration.'

---

## Author Response (AR2)

Dear Eva Álvarez et al.,

We received one review on your revised manuscript. The other reviewer was not available, but I think you have addressed most of his/her previous comments and therefore I decide to proceed with the current evaluation from reviewer #2. Reviewer #2 thinks that the revised manuscript has addressed their comments and is much easier to follow. There are a few additional comments and suggestions, but it should be easy to address. I consider this a Minor revision.

Reviewer comments are appended below for your reference.

Looking forward to seeing your revisions.

Best sincerely,

Yuan Shen

Associate Editor

Dear Editor,

Thanks a lot for handling the review of our manuscript. We have revised the manuscript according to the Reviewer #2's comments. Find below our point-by-point response to the comments that includes a list of all changes made in the manuscript.

We would like to sincerely thank our two reviewers for taking the time and effort necessary to review the manuscript, and providing feedback in such a constructive and useful way. We have updated the Acknowledgements section to recognize their support.

Best regards,

On behalf of all the authors,

Eva Álvarez

**Review #2**:

The authors have addressed my comments and the updated manuscript is much easier to follow. It is good to see that the parameter estimates converge to similar values in replicate experiments. The figures are useful (they look great, too) and the results are convincing. At this stage, I have only a few specific comments, and two remarks about the updated naming which reflect my personal preferences and which the authors may choose to ignore.

We thank the reviewer for revising so carefully our work and for the really useful comments. We have revised the manuscript to improve readability following the specific comments of the reviewer, change the name of the two EXP-Biology simulations and update the y-axis label to "depth (m)" in some figures. Following each reviewer's comment, we detail the changes we have made to the manuscript which are marked in green font in the revised version of the manuscript. The line numbers indicated in our responses refer to the revised version of the manuscript.

The new variable names are much more intuitive, and a new reader is no longer in constant need to look up the meaning of the previously numbered variables. I am not certain why the naming was not extended to the DOM and CDOM variables ($X^{(labile)}$ or $X^{(l)}$ is more intuitive to me than $X^{(1)}$) but with a total of 6 numbered variables left ($X^{(1)}$, $X^{(2)}$, $X^{(3)}$, $R^{(1)}$, $R^{(2)}$, and $R^{(3)}$) it is not a big issue.

The names of R and X were not updated in order to keep short names, especially for the parameters associated to the dynamics of each state variable (e.g. $f^{maxX2}[P(i)]$, $b[X(3)]$, $f^{X1}[Z]$ …). We have clarified earlier the correspondence (labile, semi-labile, semi-refractory) with (1,2,3) as follows:

L 171 "three pools of dissolved organic matter differentiated by reactivity into labile ($R^{(1)}$), semi-labile ($R^{(2)}$) and semi-refractory ($R^{(3)}$), and CDOM differentiated by the same reactivities ($X^{(1)}$, $X^{(2)}$ and $X^{(3)}$)."

I welcome the new names for the experiments, EXP-Biology and EXP-Physics, which are more intuitive as well. The names of the simulations have also improved. A minor issue is that "Constant leakage" and "Constant dpp" experiments do not use a constant value for leakage or dpp, but a constant (CDOM/DOM) ratio in leakage or dpp. I would prefer "Constant leakage ratio" and "Constant dpp ratio" but these names are longer, and I leave it to the authors to decide.

Thanks for the suggestion, we have taken it and changed the name of the simulations from "*Constant leakage*" to "*Constant leakage ratio*" and from "*Constant dpp*" to "*Constant dpp ratio*" (L497, Table 2, L708 and Figure 12).

**specific comments (line numbers are based on the "tracked changes" version of the manuscript)**

l 40: "plays" → "DOM plays". Changed, thanks.

l 56: "The latter" is more difficult to interpret now that the previous sentence has changed, I suggest changing it to "Remote-sensing platforms". Modified as suggested.

l 86: "dynamics at the surface is" → "dynamics at the surface are". Corrected, thanks.

l 125: A different study testing and comparing different genetic algorithms, including DE, in the context of BGC models is: Mattern and Edwards (2017): Simple parameter estimation for complex models — Testing evolutionary techniques on 3-dimensional biogeochemical ocean models.

https://doi.org/10.1016/j.jmarsys.2016.10.012

Thanks for the suggestion, we included the reference in L127.

l 172: "The subscript appended to each living and non-living component indicates ..." No subscripts have been shown yet, I would suggest either giving an example or slightly rephrasing it to: "In the following, subscripts are appended to the symbols for the constituents, indicating..."

We have rephrased to: "In the following, subscripts are appended to the symbols for the components to indicate the elemental constituent for which the state variable stands, including carbon (C), nitrogen (N), phosphorus (P), Chl-a (only in phytoplankton) and silica (Si, only in $P^{(DIATOM)}$)."

l 188: "is indicated before the vertical bar" → "appears in front of a vertical bar". Changed.

Eq 4: I don't want to be too pedantic here, but it looks like $G\_P$ and $G\_P^{balance}$ are dependent on i (plankton-specific). I would suggest adding "(i)" to the subscript, like for the symbol $f\_p^{X2}$ in Eq (5).

Thanks for pointing this out. We refer now to $G[P]$ and $G[P]^{balance}$ of each phytoplankton type as $G[P(i)]$ and $G[P(i)]^{balance}$, both in the text (L220) and in Eq. 4.

Fig 1 caption: "only for" → "only used for". Changed.

l 432: This decision is up to the authors, but I would prefer having the table with the optimized parameters in the main text, and the table with the remaining parameters in the appendix.

We have decided to keep the order of the parameter tables as they were in the previous version. Although the alternative order could also be informative, we have chosen to combine all the results related to the optimization experiments in a single Appendix.

l 447: "We kept constant a and optimized \phi because these parameters are not well documented in literature...": This sentence still does not make it clear why a lack of documentation motivates the optimization of a parameter. Maybe add some more information, such as: "We kept constant a and optimized \phi, opting to optimize only one of these two interdependent parameters because they are not well documented in literature..."

We realize now that the original sentence suggested that both parameters (aPS and phi) were not well documented. We reformulated this part to clarify that the lack of documentation motivates the choice to optimize one parameter (phi, which is not very well documented) and not the other (aPS, which is much easily derived from observed aPH spectra).

l 457: I would suggest changing the introductory sentence to DE so that it says that the parameters represent individuals and not model simulations.

We removed: "(i.e., one model simulation)"

l 462: I like the updated text, but adding a bit more information removes ambiguity: "from a total of M" → "from a total number of M observed variables". Changed.

l 463: This N should be an N_m, or more description is needed.

Yes, thank you, we changed it.

l 463: Again, a small suggested change to improve readability: "consisting of the simulation P_n and the corresponding observation O_n" → "consisting of observation O_n and corresponding simulation-based estimates P_n (model results at the temporal and spatial observation locations)"

Thanks for the suggestion. We have reformulated following the reviewer's proposal.

l 468: In the description of the DE algorithm, make sure to point out that this represents one possible implementation. For example, the mutation scheme described here is one of many. I am not suggesting to name them all, but to keep the phrasing a bit more general, e.g.: "Each mutant vector is created selecting" → "In this implementation, each mutant vector is created selecting".

Thanks for pointing this out. We have accepted this suggestion of the reviewer (L474 & 477). The reference suggested above (Mattern & Edwards, 2017) was also very useful to read several possible ways the new samples are generated from an existing population.

l 473: "Crossover is performed between each target vector and its corresponding mutant vector, retaining the target vector in the population with a probability of 0.9, ..." In a typical implementation, this procedure is applied to each element (parameter) of the vectors, sometimes mutating one element for certain. In other words, would it be correct to say "Crossover is performed between each target vector and its corresponding mutant vector, retaining each element of the target vector in the population with a probability of 0.9, otherwise introducing the corresponding element from the mutant vector."?

Thanks for the suggestion. We took it and now the sentence reads: "Here, crossover is performed between each target vector and its corresponding mutant vector, retaining each element of the target vector in the population with a probability of 0.9, otherwise introducing the corresponding element from the mutant vector."

l 494: Again, I do not want to be pedantic but 4 of the 5 following sentences start with "In EXP-Biology", "In Constant leakage", "In Constant dpp", and "In EXP-Physics" and the sentences do not make it clear that the two hypothesis-testing experiments are "EXP-Biology" and "EXP-Physics", and that "Constant leakage" and "Constant dpp" are simulations/sub-experiments of "EXP-Biology". I would suggest modifying the text to add more information: "In the first experiment EXP-Biology, we investigated [...] This experiment consisted of comparing the results of Optimized with two additional simulations. These additional simulations are: (1) Constant leakage, in which we [...] (2) Constant dpp, in which we [...] In the second experiment, EXP-Physics, ..."

Thanks for the suggestions, we do agree the structure of the sentences was a bit redundant in the previous version. We took the reviewer's alternative and reformulated slightly the paragraph L493-512.

Fig. 3 - 6: I like the figures and the changes made to them, my only suggestion is to change the y-label from "m" to "depth (m)".

All y-axis in figures that show vertical profiles now display "depth (m)" (Fig. 3, 4, 5, 6, 8 and 13).

l 701: A reference to Eq (5) may be useful in this paragraph.

The first sentence introducing the results of EXP-Biology has been modified to: "EXP-Biology focuses on the physiological processes affecting CDOM production by phytoplankton examining which is the coloured fraction of the total DOC of phytoplankton origin (Eq. 5)."